# Adaptive Partitioning Schemes for Optimistic Optimization

**Raja Sunkara** [1]   **Ardhendu Tripathy** [2]

## Abstract

Applications such as engineering design often require us to optimize a black-box function, i.e., a system whose inner processing is not analytically known and whose gradients are not available. Practitioners often have a fixed budget for the number of function evaluations and the performance of an optimization algorithm is measured by its simple regret. In this paper, we study the class of "Optimistic Optimization" algorithms for black-box optimization that use a partitioning scheme for the domain. We develop algorithms that learn a good partitioning scheme and use flexible surrogate models such as neural networks in the optimization procedure. For multi-index functions on an $m$-dimensional subspace within $d$ dimensions, our algorithm attains $\tilde{O}(n^{-\beta/d})$ regret, where $\beta = 1 + \frac{d-m}{2m-1}$, as opposed to $\tilde{O}(n^{-1/d})$ for SequOOL, a state-of-the-art optimistic optimization algorithm. We use our approach to improve the quality of Activation-aware Weight Quantization (AWQ) of the OPT-1.3B model, achieving $\sim 10\%$ improvement in performance relative to the best possible unquantized model.

## 1. INTRODUCTION AND MOTIVATION

Optimization of black-box functions is often carried out by a class of algorithms called "Optimistic Optimization" algorithms (Munos, 2014). These algorithms are preferred due to their mild assumptions; however, they require a partition scheme to be provided for the search space. The quality of a partitioning scheme depends on the function being optimized (Grill et al., 2015). If information on the function is lacking, then a default partitioning scheme, i.e., axis-aligned rectangles, is used (Jones et al., 1993; Bartlett et al., 2019). But this default choice might not be a good choice for the

[1]Missouri University of Science & Technology, Rolla MO, US [2]OpsCanvas, Alexandria VA, US. Correspondence to: Raja Sunkara <rs5cq@mst.edu>, Ardhendu Tripathy <ardhendu@opscanvas.com>.

*Proceedings of the $42^{st}$ International Conference on Machine Learning*, Vancouver, Canada. PMLR 267, 2025. Copyright 2025 by the author(s).

function. Additionally, an axis-aligned rectangle scheme limits the application to low-dimensional search spaces (as the number of rectangles grows as $k^{dn}$, where $k$ is the number of splits at each height, $d$ is the dimension and $n$ is the number of splits needed in each dimension), or it may fail to exploit low-dimensional structures in high-dimensional spaces. Thus, there is a need to develop partitioning schemes that can adapt to the low dimensional structure that may be present in a black-box function.

Global optimization is impossible without restricting the class of functions: a degenerate function that takes the value 1 at particular location and zero everywhere else can only be optimized by sampling every point in the domain. The class of functions is typically restricted by assumptions on its "smoothness". One of the first global optimization algorithms with provable convergence was obtained for the class of Lipschitz functions. The DiRect algorithm (Jones et al., 1993; Jones & Martins, 2021) is a well-known algorithm that can optimize Lipschitz functions without knowing the Lipschitz constant. A different class of functions studied in the Bayesian Optimization literature is that of the Gaussian Process (GP) prior for the unknown function $f$. Combining the observed samples with the prior mean and covariance kernel, a posterior distribution for $f$ is obtained and used to guide the sampling strategy, see e.g. GP-UCB (Srinivas et al., 2012). Since its sampling strategy required maximizing the Upper Confidence Bound obtained from the posterior, its could feasibly be applied only on low-dimensional $\mathcal{X}$. Later works (Shekhar & Javidi, 2018; Salgia et al., 2021) incorporated domain partitioning ideas to reduce the computational cost. GP-UCB can also be applied if $f$ belongs to a Reproducing Kernel Hilbert Space corresponding to the covariance kernel. However, these methods require the kernel associated with $f$ as an input.

Unlike the above methods, which assume a global characteristic for $f$, the "Optimistic Optimization" class of algorithms (HOO (Bubeck et al., 2011), SOO (Munos, 2011), SequOOL (Bartlett et al., 2019)) just assume a local smoothness condition around the global maximizer. This local smoothness ensures that the function does not rapidly decrease around its maximum value. These algorithms require as input a hierarchical partitioning of $\mathcal{X}$ that is well-behaved with respect to a semi-metric on $\mathcal{X}$. Although the semi-metric is not needed to be known, the performance of these algorithms

can be heavily influenced by the choice of the partitioning scheme. Absent any additional information, the default partitioning is the axis-aligned rectangles from DiRect, leading to the shortcomings described in the beginning. We propose to develop an algorithm that adaptively builds a partitioning scheme as new samples are collected. Additionally, it refines partitions using SequOOL in the low dimensional subspace that accounts for most of the variation in the function.

We sumarize the main contributions of our paper below.

- We demonstrate the benefit of using a learned partitioning scheme for existing derivative-free optimization algorithms such as SequOOL.

- When the function is a low-dimensional multi-index function we theoretically prove improved regret bounds shown in Table 1.

- Empirically, we demonstrate the improvement in optimization error for several benchmark functions including Rastrigin (multi-modal), Branin (multiple minima), and Sharp Ridge (non-differentiable).

- We pose the quantization of Large Language Model (LLM) as a high-dimensional black-box optimization problem and obtain an improved perplexity value.

|            | SOO              | SequOOL                | Our Method              |
|------------|------------------|------------------------|-------------------------|
| $\eta = 0$ | $\rho^{\sqrt{n}}$ | $\rho^{\bar{\Omega}(n)}$ | $\rho^{\beta\bar{\Omega}(n)}$ |
| $\eta > 0$ | $\tilde{O}(n^{-1/\eta})$ | $\tilde{O}(n^{-1/\eta})$ | $\tilde{O}(n^{-\beta/\eta})$ |

*Table 1.* Regret bounds on a budget of $n$ evaluations for a $m$-dimensional multi-index function in $d$ dimensions. $\beta = 1 + \frac{d-m}{2m-1}$ and $\rho, \eta$ are parameters for the default partitioning scheme.

**Related works**  Here we summarize the methods we have compared in our experiments. Perhaps the closest related work is Random Embedding Simultaneous Optimistic Optimization (RESOO) (Qian & Yu, 2016), which scales SOO to high-dimensional optimization problems by applying SOO in a random low-dimensional search space. The simple regret of RESOO depends only on the effective dimension of the problem, rather than the full dimension of the solution space. REMBO (Random EMbedding Bayesian Optimization) (Wang et al., 2016) uses a random projection matrix to create a lower-dimensional embedding for high-dimensional optimization problems. It then applies Bayesian optimization on this low-dimensional space, allowing it to efficiently search for optima in the reduced space. HeSBO (Nayebi et al., 2019) uses hashing-enhanced embeding subspaces. ALEBO (Adaptive Linear Embedding Bayesian Optimization) (Letham et al., 2020) builds upon and improves the original REMBO by proposing a new linear-embedding method. However, these algorithms requires an lower-bound

to the low-dimensional subspace dimension, which is difficult to obtain in real-world problems. Additionally, the Bayesian Optimization algorithms can be computationally expensive for large budgets.

Latent Action Monte Carlo Tree Search (LA-MCTS) (Wang et al., 2020) recursively partitions the high-dimensional search space into regions with high/low function values using nonlinear decision boundaries. It serves as a meta-algorithm by using existing black-box optimizers (e.g., BO, TuRBO (Eriksson et al., 2019)) as its local model. LA-MCTS creates sub-regions by partitioning the variable ranges without reducing dimensionality. Each sub-region retains the full dimensionality of the original problem space. While LA-MCTS has shown good empirical performance, it encounters challenges in high-dimensional spaces. Voronoi Optimistic Optimization (VOO) (Kim et al., 2020) is a method for optimizing functions in high-dimensional spaces. Instead of using fixed partitions, VOO creates flexible sections called Voronoi cells. Each cell contains points closest to a specific evaluated point. As new points are evaluated, the cells automatically adjust based on the best results.

Evolutionary algorithms such as CMA-ES (Hansen, 2023) and simulated annealing (Xiang et al., 2013) are other popular approaches for black-box optimization. The CMA-ES technique performs well in finding the best solutions in high-dimensional optimization problems. However, these methods do not have convergence guarantees (Loshchilov & Hutter, 2016; Nomura et al., 2021).

## 2. PROBLEM SETUP & ADAPTIVE PARTITIONING SCHEMES

We consider the problem of optimizing a function $f \colon \mathcal{X} \mapsto \mathbb{R}$ using only its evaluations at appropriately chosen points in its domain $\mathcal{X}$, which is assumed to be a closed and compact set. Given a budget of $n$ evaluations, at each $t \in \{1, 2, \ldots, n\}$ the algorithm queries a point $\mathbf{x}_t \in \mathcal{X}$ and observes a real number $y_t = f(\mathbf{x}_t)$. After exhausting its budget, the algorithm returns the estimated maximizer $\hat{\mathbf{x}}(n)$. The optimization error is quantified by the simple regret $r_n$, defined as

$$r_n \triangleq f^* - f(\hat{\mathbf{x}}(n)), \text{ where } f^* \triangleq \sup_{\mathbf{x} \in \mathcal{X}} f(\mathbf{x}),$$

$$\text{and } \mathbf{x}^* \in \mathcal{X} \text{ such that } f^* = f(\mathbf{x}^*).$$

Our focus is on the class of *optimistic optimization* algorithms (Munos, 2011; Bubeck et al., 2011; Valko et al., 2013; Munos, 2014; Grill et al., 2015; Bartlett et al., 2019). These algorithms require as input a hierarchical partitioning of the domain $\mathcal{X}$ for their search procedure.

**Definition 2.1.** Partitioning scheme (Bartlett et al., 2019). Let $\mathcal{P}$ denote a tree representation of the domain $\mathcal{X}$. All the

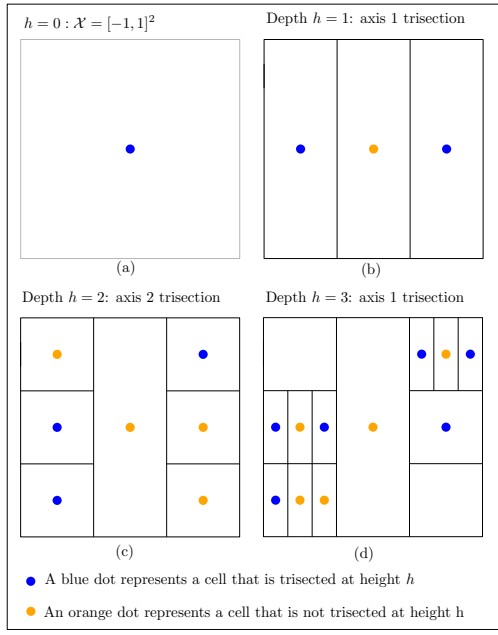

$h = 0 : \mathcal{X} = [-1, 1]^2$ (a)

Depth $h = 1$: axis 1 trisection (b)

Depth $h = 2$: axis 2 trisection (c)

Depth $h = 3$: axis 1 trisection (d)

● A blue dot represents a cell that is trisected at height $h$

● An orange dot represents a cell that is not trisected at height h

*Figure 1.* First four depths of the default partitioning $\mathcal{P}$ on $[-1, 1]^2$.

cells at depth $h$ form a partition of $\mathcal{X}$ and are denoted as $\mathcal{P}_h$. We index the cells at depth $h$ with an additional index $i$, i.e., $\mathcal{P}_{h,i}$ is a cell in the tree at depth $h$. A cell $\mathcal{P}_{h,i}$ is partitioned to obtain one or more cells of $\mathcal{P}_{h+1}$. $\mathcal{P}_h^*$ denotes a cell at depth $h$ containing a maximizer $\mathbf{x}^*$ of $f$ and $\mathbf{x}_{h,i}$ denotes a representative location within the $\mathcal{P}_{h,i}$ cell.

A common choice for the domain is obtained when we have interval constraints on each of its components. In this case, without loss of generality, we can consider $\mathcal{X} = [-1, 1]^d$. And a default choice for the partitioning scheme that is often used in practice is the axis-aligned trisection scheme (Jones et al., 1993). We use $\mathbb{H}_1^m$ to denote the unit hypercube in $m$-dimensional space. i.e., $[-1, 1]^m$ domain.

**Definition 2.2.** The default partitioning scheme $\mathcal{P} = \{\mathcal{P}_{h,i} : h, i \in \mathbb{N}_0\}$ of $\mathcal{X} = [-1, 1]^d$ uses an axis-aligned trisection method in a round-robin manner (see Figure 1). At depth $h = 0$, there is a single cell $\mathcal{P}_{0,1} = \mathcal{X}$. Each cell $\mathcal{P}_{h,i}$ at depth $h$ is split into three children cells $\mathcal{P}_{h+1,j}$ at depth $h+1$. The trisection occurs along the $(h \bmod d) + 1$ axis, i.e., the new cells are created by introducing $(d-1)$-dimensional hyperplanes orthogonal to the chosen axis. For all $h, i$ the representative $\mathbf{x}_{h,i}$ is the midpoint of the cell $\mathcal{P}_{h,i}$.

We consider partitioning schemes that are not aligned with the standard axes. We define such a rotated low-dimensional partitioning scheme using a matrix of orthonormal rows.

**Definition 2.3.** Given $\mathbf{A} \in \mathbb{R}^{m \times d}$ such that $\mathbf{A}\mathbf{A}^\top = \mathbf{I}_m$ and a scalar $\alpha > 0$, we establish a partitioning scheme denoted as $\mathcal{A}$. Let the default partitioning scheme (Definition 2.2) on $[-\alpha, \alpha]^m$ be denoted as $\mathcal{T}$. For any depth $h$ and

index $i$, the cell $\mathcal{A}_{h,i} \triangleq \{\mathbf{A}^\top \mathbf{t} : \mathbf{t} \in \mathcal{T}_{h,i}\}$ is a cell in the partitioning on the $m$-dimensional projection of $\mathcal{X}$ onto the subspace spanned by rows of $\mathbf{A}$.

Since the projection of $\mathcal{X}$ onto $\mathbf{A}$ can result in points outside $\mathcal{X}$, the value of $\alpha$ is chosen to ensure that the projection is covered by the $\mathcal{A}$ partitioning scheme. An appropriate value of $\alpha$ is in the following definition.

**Definition 2.4.** Let $\mathbf{c}_j \in \{-1, 1\}^d$ denote the $2^d$ corners, indexed by $j$, of the default axis-aligned $\mathcal{P}$ partitioning scheme. Given a matrix $\mathbf{A}$ with $m$ orthonormal rows denoted as $\mathbf{a}_1, \mathbf{a}_2, \ldots, \mathbf{a}_m$, we define

$$\boldsymbol{\alpha}_{\max} \triangleq \left[ \max_{1 \leq j \leq 2^d} \mathbf{a}_1^\top \mathbf{c}_j, \max_{1 \leq j \leq 2^d} \mathbf{a}_2^\top \mathbf{c}_j, \cdots \max_{1 \leq j \leq 2^d} \mathbf{a}_m^\top \mathbf{c}_j \right]^\top$$

as the extent of the $\mathcal{A}$ partitioning scheme. We choose the largest component of the extent as $\alpha \triangleq \max_{i,j} \mathbf{a}_i^\top \mathbf{c}_j$.

In this paper, we will provably demonstrate the advantage of our method when there is a low-dimensional ridge structure present in the black-box function. We consider the class of multi-index functions (Fornasier et al., 2012), i.e., there is a matrix $\mathbf{A} \in \mathbb{R}^{m \times d}$ with orthonormal rows and a Lipschitz function $g$ such that

$$f(\mathbf{x}) = g(\mathbf{A}\mathbf{x}). \tag{1}$$

Then optimizing over the low-dimensional subspace is sufficient to recover the maximizer. All proofs and omitted details can be found in the supplementary.

**Proposition 2.5.** *Suppose* $f : \mathbb{R}^d \mapsto \mathbb{R}$ *is a multi-index function with* $f(\mathbf{x}) = g(\mathbf{A}\mathbf{x})$. *If* $\mathbf{x}^* \in \mathcal{X} = [-1, 1]^d$ *is an optimizer and* $\alpha$ *is chosen as per definition 2.4, then there is a* $\mathbf{z}^* \in \mathbb{R}^m$ *such that* $f(\mathbf{A}^\top \mathbf{z}^*) = f(\mathbf{x}^*)$ *and* $\|\mathbf{z}^*\|_\infty \leq \alpha$.

*Proof Sketch.* For an optimizer $\mathbf{x}^* \in [-1, 1]^d$ let $\mathbf{z}^* = \mathbf{A}\mathbf{x}^*$. Then $f(\mathbf{A}^\top \mathbf{z}^*) = f(\mathbf{A}^\top \mathbf{A}\mathbf{x}^*) = f(\mathbf{x}^*)$ since $\mathbf{A}^\top \mathbf{A}$ is identity. We then use that $\mathbf{x}^*$ lies in a convex hull of cube corners and apply the definition of $\alpha$ to show that $\|\mathbf{z}^*\|_\infty \leq \alpha$, see details in Appendix 11.4. □

The above proposition implies that if we have access to the true subspace matrix $\mathbf{A}$, we can compute $\alpha$ and restrict our optimization to the $m$ dimensional space $\alpha\mathbb{H}_1^m$ using the $\mathcal{A}$ partitioning scheme. We theoretically characterize the benefit of using the partitioning scheme $\mathcal{A}$ for the class of multi-index functions, i.e., if $f$ satisfies (1) then using the partitioning scheme $\mathcal{A}$ can decrease $r_n$ at a faster rate with $n$. To show this, we use the partitioning scheme assumption made by Grill et al. (2015) and Bartlett et al. (2019) that states that the suboptimality of any point in the $\mathcal{P}_h^*$ cell improves as depth $h$ increases. The rate of this improvement is characterized by a parameter $\rho \in (0, 1)$.

**Assumption 2.6.** (Bartlett et al., 2019) For any global optimum $\mathbf{x}^\star$, there is a $\nu > 0$ and $\rho \in (0, 1)$ such that for all $h \in \mathbb{N}_0$ and all $\mathbf{x} \in \mathcal{P}_h^*$, we have that $f(\mathbf{x}) \geq f(\mathbf{x}^\star) - \nu\rho^h$.

Bartlett et al. (2019) define the *near-optimality dimension* which characterizes the difficulty of optimizing a black-box function using a partitioning scheme.

**Definition 2.7.** (Bartlett et al., 2019). Consider a partitioning $\mathcal{P}$ that satisfies Assumption 2.6 for some $\nu, \rho$. For any $C > 1$, the near-optimality dimension $\eta_{\mathcal{P}}(\nu, \rho, C)$ of $f$ with respect to the partitioning $\mathcal{P}$ is defined as

$$\eta_{\mathcal{P}}(\nu, \rho, C) \triangleq \inf\{\eta \geq 0 : |\mathcal{N}_{\mathcal{P}}(3\nu\rho^h)| \leq C\rho^{-\eta h} \,\forall h \geq 0\},$$

where $\mathcal{N}_{\mathcal{P}}(3\nu\rho^h)$ is the set of near-optimal cells $\mathcal{P}_{h,i}$ at depth $h$ for which $\sup_{\mathbf{x} \in \mathcal{P}_{h,i}} f(\mathbf{x}) \geq f(\mathbf{x}^*) - 3\nu\rho^h$.

Intuitively, a larger $\rho$ implies that the function is only improving slowly near the maximizer, and a larger $\eta$ implies that there are many near-optimal cells which must be ruled out to get the true maximizer. In both cases, we need a larger budget of evaluations to converge. The following example shows a partitioning scheme $\mathcal{A}$ with a lower near-optimality dimension than the default partitioning scheme.

**Example 2.8.** Consider the function $f(x_1, x_2) = g(\mathbf{A}\mathbf{x}) = 1 - |x_1|$ with $\mathbf{A} = [1, 0]$ and $g(z) = 1 - |z|$. Let $\eta_{\mathcal{P}}, \eta_{\mathcal{A}}$ be the near-optimality dimensions for the partitioning schemes $\mathcal{P}, \mathcal{A}$ as defined in Definitions 2.2 and 2.3, respectively. Then we have that $\eta_{\mathcal{P}} = 1$ and $\eta_{\mathcal{A}} = 0$.

In the next example, the subspace defined by $\mathbf{A}$ is not aligned with the canonical axes and we see that $\rho_{\mathcal{A}}$ is smaller than $\rho$ of the default partitioning scheme.

**Example 2.9.** Consider the function $f(x_1, x_2) = g(\mathbf{A}\mathbf{x}) = 1 - |x_1 + x_2|$ with $\mathbf{A} = [1, 1]$ and $g(k) = 1 - |k|$. Let $\eta_{\mathcal{P}}, \eta_{\mathcal{A}}$ be the near-optimality dimensions for the partitioning schemes $\mathcal{P}, \mathcal{A}$ associated with $(\nu, \rho), (\nu_{\mathcal{A}}, \rho_{\mathcal{A}})$ respectively. Then $\rho = 1/\sqrt{3}$ and $\rho_{\mathcal{A}} = 1/3$ and $\eta_{\mathcal{P}} = \eta_{\mathcal{A}} = 0$.

In addition to identifying the important subspace spanned by $m$ orthonormal directions, an adaptive partitioning scheme can choose which direction to split at each depth.

**Definition 2.10.** Direction selection strategy. For a partitioning $\mathcal{A}$ in Definition 2.3, a direction selection strategy $\tau_h : \mathcal{H} \to [1 : m]$ defined for each height $h$ takes the history $\mathcal{H}$ of all past function evaluations till depth $h - 1$ and outputs the index of the direction to be split at depth $h$.

In the following example, a direction selection strategy that splits the $x_1$ axis twice as often as the $x_2$ axis results in a lower near-optimality dimension than that of the default partition $\mathcal{P}$ which splits both the axes in equal proportion.

**Example 2.11.** The near-optimality dimension of the default partitioning scheme for the function $f(x_1, x_2) =$

---

**Algorithm 1** Obtaining directions for an adaptive partitioning scheme

---
**Require:** $T$, oracle for $f$ which is a multi-index function defined using $\mathbf{A}$ (see (1))

1: Sample $f$ at $T$ points chosen as $x^{(i)} \overset{iid}{\sim} \mathcal{N}(\mathbf{0}, \mathbf{I}_d)$ and fit a single hidden-layer neural network $\hat{y}$
2: $\hat{\mathbf{A}} \leftarrow$ top right singular vectors of the hidden layer weight matrix preserving 95% variance
3: $u \leftarrow$ Upper bound to $\mathrm{dist}(\mathbf{A}, \hat{\mathbf{A}})$ obtained in Lemma 9.11 or Theorem 9.12
4: **return** $\hat{\mathbf{A}}$ and $\hat{\alpha}$ (see Equation 4) used to specify the partitioning scheme $\mathcal{A}$ in Definition 2.3

---

$1 - |x_1| - x_2^2$ is $\eta_{\mathcal{P}} = 0.5$. For the partitioning scheme $\mathcal{A}$ from Definition 2.3 with $\mathbf{A} = I_2, \alpha = 1$ and direction selection strategy $\tau_h = 1$ if $h \mod 3 \neq 0$ and $\tau_h = 2$ otherwise, its near-optimality dimension $\eta_{\mathcal{A}} = 0$.

Empirically, as we increase the optimization budget, we observe that SequOOL on $\mathcal{A}$ decreases regret at a faster rate than SequOOL on $\mathcal{P}$ for the above example functions.

## 3. PROPOSED ALGORITHMS FOR BLACK-BOX OPTIMIZATION

To utilize an adaptive partitioning scheme, we develop two kinds of algorithms: 1. a two-stage algorithm where the first stage learns an adaptive partitioning scheme and the second stage uses it for optimization, and 2. an interleaved algorithm where learning and optimization happen iteratively.

**Two-stage algorithm.** In the first stage, we use a learning algorithm to obtain $\hat{\mathbf{A}}$, i.e., the directions used to define the adaptive partitioning scheme $\mathcal{A}$. If $f$ is a multi-index function satisfying (1), the quality of this estimate is measured by the subspace distance between $\hat{\mathbf{A}}$ and the true $\mathbf{A}$. The value of $\hat{\alpha}$ chosen in Lemma 4.5 is such that $f(\hat{\mathbf{A}}^\top \mathbf{t}) = f(\mathbf{x}^\star)$ for some $\mathbf{t} \in [-\hat{\alpha}, \hat{\alpha}]^m$ and the optimization can find the maximizer in the low-dimensional subspace.

Algorithm 1 learns $\hat{\mathbf{A}}$ by fitting a single hidden-layer neural network to the function evaluations at random locations in its domain. A single hidden-layer neural network can model the fact that only a subspace of the domain explains all the variation in the function values (see Proposition 8.1). In practice, we can choose the value of $m$ to explain a desired percentage (such as 95%) of the total variation in the SVD step calculating $\hat{\mathbf{A}}$. Another approach to learn $\hat{\mathbf{A}}$ is from Fornasier et al. (2012, Algorithm 2) which estimates the gradient of the function using finite differences. The second stage applies SequOOL to the partitioning scheme $\mathcal{A}$ returned by Algorithm 1.

**Algorithm 2** SequOOL on an adaptive partitioning scheme with a direction selection strategy

---

**Require:** Total number of openings $n$, number of samples $T$ for updating $\hat{f}$, integer $c$ stating how often $\hat{f}$ is updated, number of dimensions $m$, oracle for $f$, direction selection strategy $\tau_h$.

Setup a partitioning scheme $\hat{\mathcal{A}}$ with $\hat{\mathbf{A}}$ and $\hat{\alpha}$ obtained by (Fornasier et al., 2012, Algorithm 2)

Obtain $\hat{f}$ by fitting a neural network on available samples, set $h_{\max} \leftarrow \lfloor n^2 / (n\overline{\log}n + \frac{Tn}{3c}) \rfloor$

**for** $h \leftarrow 1$ **to** $h_{\max}$ **do**

  $\mathcal{T}_h \leftarrow$ the cell with the largest function sample value at height $h$

  **if** $h \mod c = 0$ **then**

    Obtain evaluations of $f$ at $T$ uniform random locations in $\mathcal{T}_h^*$ and update $\hat{f}$

  **end if**

  Open $\lfloor h_{\max}/h \rfloor$ cells at depth $h$ and trisect them along the direction returned by $\tau_h(\hat{f})$

**end for**

**return** $\hat{\mathbf{x}}(n) \leftarrow \arg\max_{\text{all } h,i} f(\mathbf{x}_{h,i})$

---

**Interleaved learning and optimization.** Instead of separating the learning and optimization in two distinct stages, an interleaved algorithm updates the neural network fit at regular intervals. The updated approximant is used to specify the direction selection strategy in Definition 2.10.

Algorithm 2 uses a parameter $\tau_h$ that takes the updated approximation $\hat{f}$ as input. Our proposed method for $\tau_h$ is the lookahead direction selection strategy, which is detailed in Algorithm 3 in Appendix section 3.

A different value of $h_{\max}$ is used in Algorithm 2, compared to $h_{\max} = \left\lfloor \frac{n^2}{n\overline{\log}n} \right\rfloor$ used in SequOOL, to account for the additional evaluations required to update the surrogate model $\hat{f}$ at regular intervals. This adjustment ensures that the total number of evaluations, including both SequOOL openings and model updates, remains within the overall budget. The following proposition guarantees that the chosen $h_{\max}$ ensures the total number of function evaluations does not exceed $3n$.

**Proposition 3.1.** *Let* $n, T, c, h_{\max}$ *be the parameters defined in Algorithm 2 with* $\overline{\log}n \triangleq \sum_{t=1}^{n} \frac{1}{t}$. *Then the total number of function evaluations taken by the algorithm will not exceed* $3n$.

## 4. THEORETICAL ANALYSIS

We show that for the class of multi-index functions (1), the partitioning $\hat{\mathcal{A}}$ obtained using the learned $\hat{\mathbf{A}}$ can have a lower $\eta$ and a lower $\rho$ compared to that of the default partitioning $\mathcal{P}$. Our proof proceeds by analyzing the rela-

tionships between three partitioning schemes: the default scheme $\mathcal{P}$, the scheme $\mathcal{A}$ based on the true subspace $\mathbf{A}$, and the scheme $\hat{\mathcal{A}}$ based on the estimated subspace $\hat{\mathbf{A}}$. Our analysis proceeds in two main stages: We first relate the parameters of the default partitioning scheme $\mathcal{P}$ to those of the scheme $\mathcal{A}$. This includes comparing their SequOOL parameters $(\nu, \rho)$, characterizing their star cells, and bounding the number of near-optimal cells. We then extend this analysis to the estimated scheme $\hat{\mathcal{A}}$, relating its properties to those of $\mathcal{A}$. This involves quantifying the impact of using an estimated subspace and establishing relationships between the SequOOL parameters of $\hat{\mathcal{A}}$ and $\mathcal{A}$.

We will use the notation of lattices to relate the number of near-optimal cells in two different partitioning schemes.

**Definition 4.1.** (Vaikuntanathan, 2011) Given $m$ linearly independent vectors $\mathbf{b}_1, \ldots, \mathbf{b}_m \in \mathbb{R}^m$, the lattice generated by them is defined as $L(\mathbf{b}_1, \ldots, \mathbf{b}_m) = \{\sum_{i=1}^{m} a_i \mathbf{b}_i \mid a_i \in \mathbb{Z}\}$. We call $\mathbf{b}_1, \ldots, \mathbf{b}_m$ a basis of the lattice. We denote lattices formed by the standard basis vectors $\mathbf{e}_1, \mathbf{e}_2, \ldots, \mathbf{e}_m$ as $\Lambda$. Thus, $\Lambda = L(\mathbf{e}_1, \mathbf{e}_2, \ldots, \mathbf{e}_m) = \mathbb{Z}^m$. The lattice $\Lambda$ scaled by a scalar $\kappa$ is the same as $L(\kappa\mathbf{e}_1, \kappa\mathbf{e}_2, \ldots, \kappa\mathbf{e}_n)$.

For a $d$-dimensional vector $\mathbf{x} = [x_1, \ldots, x_d]^\top$, we use $\|\mathbf{x}\|_p$ and $\|\mathbf{x}\|_\infty$ to denote its $\ell_p$ and $\ell_\infty$ norm respectively. The following lemma bounds the number of cells from a finer partitioning scheme that cover a given region in the domain.

**Lemma 4.2.** *Let* $\kappa, \kappa' \in \mathbb{R}^+$ *with* $\kappa > \kappa'$, *and let* $B(\mathbf{x}, r) = \{\mathbf{y} \in \mathbb{R}^m : \|\mathbf{y} - \mathbf{x}\|_\infty \leq r\}$. *Consider a lattice* $\Lambda$ *as defined in Definition 4.1 , scaled by* $\kappa'$, *and translated by a vector* $\mathbf{t}$ *to form the lattice* $\Lambda + \mathbf{t}$. *Define the subset of all lattice points that cover* $B(\mathbf{0}, \kappa)$ *as* $\mathcal{C}(\kappa, \kappa', \mathbf{t}) \subseteq \Lambda + \mathbf{t}$, *i.e.,*

$$\mathcal{C}(\kappa, \kappa', \mathbf{t}) = \{\mathbf{c}_i \in \Lambda + \mathbf{t} : B(\mathbf{c}_i, \kappa') \cap B(\mathbf{0}, \kappa) \neq \emptyset$$
$$\text{and } B(\mathbf{0}, \kappa) \subseteq \bigcup_i B(\mathbf{c}_i, \kappa')\}.$$

*Then, the cardinality of* $\mathcal{C}$ *satisfies:*

$$\left(\frac{\kappa}{\kappa'}\right)^m \leq |\mathcal{C}(\kappa, \kappa', \mathbf{t})| \leq \left(\frac{\kappa}{\kappa'} + 2\right)^m$$

We switch from $\mathcal{A}$ to $\mathcal{T}$ to simplify analysis by working directly in the low-dimensional space. Detailed mathematical justification is in Appendix Section 11.13.

The following lemma provides an upper bound on the number of near-optimal cells in scheme $\mathcal{A}$ relative to scheme $\mathcal{P}$. This relationship is fundamental for comparing the near-optimality dimensions of the two schemes, which will be addressed in a subsequent lemma (Lemma 4.4).

**Lemma 4.3.** *For the partitioning schemes* $\mathcal{P}$ *and* $\mathcal{T}$ *from Definition 2.7, let* $\mathcal{N}_\mathcal{P}(\epsilon)$ *be the number of cells* $\mathcal{P}_{h,i}$ *at depth* $h$ *for which* $\sup_{\mathbf{x} \in \mathcal{P}_{h,i}} f(\mathbf{x}) \geq f(\mathbf{x}^*) - \epsilon$, *and similarly for*

$\mathcal{N}_\mathcal{T}(\epsilon)$. *Then*

$$\mathcal{N}_\mathcal{T}(\epsilon) \le C\mathcal{N}_\mathcal{P}(\epsilon) \quad \text{where} \quad C = 3^d d^{d-m} (12\sqrt{m})^m.$$

*Proof Sketch.* Consider the following $POpt$ relation:

$$POpt \triangleq \{(\mathcal{T}_{h,i}, \mathcal{P}_{h,j}) : \mathcal{T}_{h,i} \in \mathcal{N}_\mathcal{T}(\epsilon), \ \mathcal{P}_{h,j} \in \mathcal{N}_\mathcal{P}(\epsilon),$$
$$\mathcal{T}_{h,i} \cap \mathbf{A}\mathcal{P}_{h,j} \ne \emptyset\},$$
$$\mathbf{A}\mathcal{P}_{h,j} \triangleq \{\mathbf{Ax} \mid \mathbf{x} \in \mathcal{P}_{h,j}\}.$$

For any $h, i$, consider the cell $\mathcal{T}_{h,i} \in \mathcal{N}_\mathcal{T}(\epsilon)$. Let $l$ be the lower bound to the number of elements in $POpt$ that are of the form $(\mathcal{T}_{h,i}, \cdot)$, implying $|POpt| \ge |\mathcal{N}_\mathcal{T}(\epsilon)| \cdot l$. Similarly, for any $h, j$, consider the cell $\mathcal{P}_{h,j} \in \mathcal{N}_\mathcal{P}(\epsilon)$. Let $u$ be the upper bound to the number of elements in $POpt$ that are of the form $(\cdot, \mathcal{P}_{h,j})$. Then, we have: $|\mathcal{N}_\mathcal{P}(\epsilon)| \cdot u \ge |POpt|$. Combining these two inequalities, we get:

$$|\mathcal{N}_\mathcal{T}(\epsilon)| \le |\mathcal{N}_\mathcal{P}(\epsilon)| \cdot \frac{u}{l} \tag{2}$$

In our proof in Appendix 11.10, we further refine the upper bound on $u$ to $(12\sqrt{m})^m 3^d 3^{k(d-m)}$ and the lower bound on $l$ to $(1/d)^{d-m} 3^{k(d-m)}$, leading to the result. $\square$

The above lemma is a key technical result that enables us to relate the parameters of the partitioning schemes $\mathcal{A}$ and $\mathcal{P}$.

**Lemma 4.4.** *For a function in the multi-index class (1) with known $\mathbf{A} \in \mathbb{R}^{m \times d}$ and $m < d$, let $(\nu, \rho, \eta_\mathcal{P}, C)$, $(\nu_\mathcal{A}, \rho_\mathcal{A}, \eta_\mathcal{A}, C_\mathcal{A})$ be parameters of $\mathcal{P}, \mathcal{A}$. Let $l_f \triangleq f^* - \inf_{\mathbf{x} \in \kappa_1 \mathbb{H}_1^d} f(\mathbf{x})$. Then we have that*

$$\nu_\mathcal{A} = \max\{\nu, l_f\} \rho^{(1-\beta)(m-1)-\tilde{h}_1},$$
$$\rho_\mathcal{A} = \rho^\beta,$$
$$\eta_\mathcal{A}(\nu_\mathcal{A}, \rho_\mathcal{A}, C_\mathcal{A}) \le \eta_\mathcal{P}(\nu, \rho, C)/\beta,$$
$$\text{and } C_\mathcal{A} = 3^d d^{d-m} (12\sqrt{m})^m C \rho^{-\eta_\mathcal{P} \tilde{h}_3},$$

*where $\beta \triangleq 1 + \frac{d-m}{2m-1}, \tilde{h}_1 \triangleq d\lceil \log_3 \kappa_1 \rceil$ and*

$$\kappa_1 \triangleq 3^{\lceil \log_3 \sqrt{m}\alpha \rceil}, \quad \tilde{h}_3 \triangleq -\left\lfloor \log_\rho \frac{\nu_\mathcal{A}}{\nu} \right\rfloor. \tag{3}$$

*Proof Sketch.* Lemma 9.4 shows that $\mathcal{A}$ is a valid partitioning scheme and relates its parameters to those of the default partitioning scheme $\mathcal{P}$. We then use Lemma 4.3 to bound the number of near-optimal cells in $\mathcal{A}$, see details in Appendix 11.11. $\square$

Since $\beta \ge 1$, the previous lemma shows that $\eta_\mathcal{A} \le \eta_\mathcal{P}$. Furthermore, since $\rho \in (0, 1)$, $\rho_\mathcal{A} \le \rho$ and $\nu \le \nu_\mathcal{A}$. As an illustration, Example 2.9 satisfies the conditions of Lemma 4.4 with $d = 2$, $m = 1$, and hence $\beta = 1 + \frac{d-m}{2m-1} = 2$. This implies $\rho_\mathcal{A} = \rho^2$ and $\eta_\mathcal{A}(\nu_\mathcal{A}, \rho_\mathcal{A}, C_\mathcal{A}) \le \eta_\mathcal{P}(\nu, \rho, C)/2$.

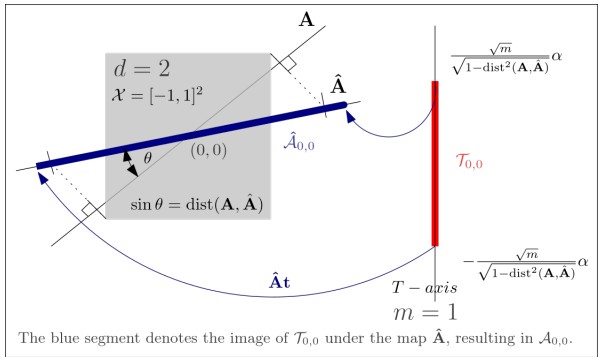

*Figure 2.* Illustration of Lemma 4.5 for $d = 2, m = 1$, showing the true subspace $\mathbf{A}$ and the estimated subspace $\hat{\mathbf{A}}$ used in defining the expansion factor.

We previously computed $\rho = 1/\sqrt{3}$ and $\rho_\mathcal{A} = 1/3$, which confirms the first conclusion. Additionally, since both $\eta_\mathcal{P}$ and $\eta_\mathcal{A}$ are 0, the second conclusion holds with equality.

When the low rank matrix $\mathbf{A}$ is unknown, we use the estimation guarantees for the learning algorithms of Fornasier et al. (2012) and Mousavi-Hosseini et al. (2023) that bound the subspace distance $\text{dist}(\mathbf{A}, \hat{\mathbf{A}})$. The subspace distance between the two row subspaces $\mathbf{A}$ and $\hat{\mathbf{A}}$ and is given by $\left\| \mathbf{A}^\top \mathbf{A} - \hat{\mathbf{A}}^\top \hat{\mathbf{A}} \right\|_2$, where $\|\cdot\|_2$ denotes the spectral norm.

The following lemma shows that we need an upper bound on the subspace distance $\text{dist}(\mathbf{A}, \hat{\mathbf{A}})$ to ensure that an optimizer exists within the low-dimensional search space.

**Lemma 4.5.** *Consider a multi-index function $f(\mathbf{x}) = g(\mathbf{Ax})$ over $\mathbb{R}^d$. Let $\mathbf{x}^*$ be an optimizer of $f$ within $[-1, 1]^d$, $\alpha$ be as in definition 2.4 and $\hat{\mathbf{A}}$ be an estimate of $\mathbf{A}$ satisfying $\text{dist}(\mathbf{A}, \hat{\mathbf{A}}) < 1$. Then there exists a $\mathbf{z}^* \in \mathbb{R}^m$ such that $f(\hat{\mathbf{A}}^\top \mathbf{z}^*) = f(\mathbf{x}^*)$ and $\|\mathbf{z}^*\|_\infty \le \frac{\sqrt{m}}{\sqrt{1-\text{dist}^2(\mathbf{A}, \hat{\mathbf{A}})}}\alpha$.*

*Proof Sketch.* Consider a $\mathbf{z}^*$ satisfying $\mathbf{A}\hat{\mathbf{A}}^\top \mathbf{z}^* = \mathbf{Ax}^*$. Since $\text{dist}(\mathbf{A}, \hat{\mathbf{A}}) < 1$, the matrix $\mathbf{A}\hat{\mathbf{A}}^\top$ is invertible, giving that $\mathbf{z}^* = (\mathbf{A}\hat{\mathbf{A}}^\top)^{-1} \mathbf{Ax}^*$ and $f(\hat{\mathbf{A}}^\top \mathbf{z}^*) = g(\mathbf{A}\hat{\mathbf{A}}^\top \mathbf{z}^*) = g(\mathbf{A}\hat{\mathbf{A}}^\top (\mathbf{A}\hat{\mathbf{A}}^\top)^{-1} \mathbf{Ax}^*) = f(x^*)$. Finally, we bound $\|\mathbf{z}^*\|_\infty$ using matrix norm inequalities and the subspace distance $\text{dist}(\mathbf{A}, \hat{\mathbf{A}})$ in Appendix 11.12. $\square$

Figure 2 shows an illustration of the lemma. Specifically, optimizing $f$ over the blue segment is enough to obtain its optimum value over $\mathcal{X} = [-1, 1]^2$. The value of $\hat{\alpha}$ used in Algorithm 1 is obtained from this lemma. If we have a $u \ge \text{dist}(\mathbf{A}, \hat{\mathbf{A}})$, then since $\alpha \le \sqrt{d}$, we set

$$\hat{\alpha} \triangleq \sqrt{dm/(1 - u^2)} \tag{4}$$

to ensure it is larger than $\frac{\sqrt{m}}{\sqrt{1-\text{dist}^2(\mathbf{A}, \hat{\mathbf{A}})}}\alpha$. The value of $u$

is obtained from Lemma 9.11 or Theorem 9.12, along with Lemma 9.8. In practice, $m$ is chosen to obtain a $\mathbf{A}$ that preserves 95% variance in the hidden layer weight matrix.

The next lemma bounds the number of near-optimal cells in $\hat{\mathcal{T}}$ in terms of those in $\mathcal{T}$. This is a key step toward comparing their near-optimality dimensions (see Lemma 4.7).

**Lemma 4.6.** *For the partitioning schemes $\mathcal{T}$ and $\hat{\mathcal{T}}$ in Definition 2.3, let $\mathcal{N}_{\mathcal{T}}(3\nu_{\mathcal{T}}\rho_{\mathcal{T}}^h)$ be the number of cells $\mathcal{T}_{h,i}$ at depth $h$ for which $\sup_{\mathbf{z}\in\mathcal{T}_{h,i}} g(\mathbf{z}) \geq f^* - 3\nu_{\mathcal{T}}\rho_{\mathcal{T}}^h$. Similarly, let $\mathcal{N}_{\hat{\mathcal{T}}}(3\nu_{\hat{\mathcal{T}}}\rho_{\hat{\mathcal{T}}}^h)$ be the number of cells $\hat{\mathcal{T}}_{h,i}$ at depth $h$ for which $\sup_{\mathbf{z}\in\hat{\mathcal{T}}_{h,i}} g(\mathbf{A}\hat{\mathbf{A}}^\top \mathbf{z}) \geq f^* - 3\nu_{\hat{\mathcal{T}}}\rho_{\hat{\mathcal{T}}}^h$. Then*

$$\forall h \geq 0, \quad \mathcal{N}_{\hat{\mathcal{T}}}(3\nu_{\hat{\mathcal{T}}}\rho_{\hat{\mathcal{T}}}^h) \leq 4^m \mathcal{N}_{\mathcal{T}}(3\nu_{\mathcal{T}}\rho_{\mathcal{T}}^h). \quad (5)$$

*Proof Sketch.* We aim to upper-bound the number of near-optimal cells in the partitioning scheme $\hat{\mathcal{T}}$ in terms of those in $\mathcal{T}$. The key idea is to relate the two schemes through the transformation $\mathbf{A}\hat{\mathbf{A}}^\top$, under which we optimize the function $g(\mathbf{A}\hat{\mathbf{A}}^\top \mathbf{z})$ over $\hat{\mathcal{T}}$. For a near-optimal cell $\mathcal{T}_{h,i} \in \mathcal{T}$, its preimage under this transformation defines a bounded set $B$ in $\mathbb{R}^m$, which we enclose within a hypercube of computable size. To bound the number of near-optimal cells in $\hat{\mathcal{T}}$, we count how many of its cells at height $h$ can intersect this hypercube. We invoke the Lemma 4.2 and show that at most $4^m$ such cells can intersect the enlarged region. The complete proof is provided in Appendix 11.17 □

**Lemma 4.7.** *Consider the partitioning scheme $\hat{\mathcal{A}}$ obtained using $\hat{\mathbf{A}}$. Let $l_g = g^* - \inf_{\mathbf{z}\in\kappa_2\alpha\mathbb{H}_1^m} g(\mathbf{A}\hat{\mathbf{A}}^\top \mathbf{z})$. Then,*

$$\nu_{\hat{\mathcal{A}}} = \max\{\nu_{\mathcal{A}}, l_g\}\rho_{\mathcal{A}}^{-\tilde{h}_2}, \quad \rho_{\hat{\mathcal{A}}} = \rho_{\mathcal{A}}, \quad \eta_{\hat{\mathcal{A}}} \leq \eta_{\mathcal{A}}$$

*with*

$$C_{\hat{\mathcal{A}}} = C_{\mathcal{A}}4^m \rho_{\mathcal{A}}^{\eta_{\mathcal{A}}\tilde{h}_4}, \tilde{h}_2 = m + m\left\lceil \log_3 \frac{2\sqrt{m}\kappa_2}{\kappa_2 - 1} \right\rceil$$

$$\kappa_2 = \frac{\sqrt{m}}{\sqrt{1 - \mathrm{dist}^2(\mathbf{A}, \hat{\mathbf{A}})}}, \quad \tilde{h}_4 = -\left\lfloor \log_{\rho_{\mathcal{A}}} \frac{\nu_{\hat{\mathcal{A}}}}{\nu_{\mathcal{A}}} \right\rfloor \quad (6)$$

*Proof Sketch.* We first relate the SequOOL parameters $(\nu, \rho)$ of the partitioning scheme $\hat{\mathcal{A}}$ to the scheme $\mathcal{A}$ using Lemma 9.6, yielding expressions for $(\nu_{\hat{\mathcal{A}}}, \rho_{\hat{\mathcal{A}}})$. Next, using the bound on the number of near-optimal cells in $\mathcal{A}$ (Definition 2.7), we upper bound the number of near-optimal cells in $\hat{\mathcal{A}}$ by composing with the Lemma 4.6. The complete proof is provided in Appendix 11.18. □

**Theorem 4.8.** *For a function in the multi-index class (1), the regret of SequOOL applied on the partitioning scheme using $\hat{\mathbf{A}}$ returned by Algorithm 1 and $\hat{\alpha} = \sqrt{dm}/\sqrt{1 - \mathrm{dist}^2(\mathbf{A}, \hat{\mathbf{A}})}$ satisfies*

$$r_n \leq \begin{cases} \gamma(\nu, \rho)\rho^{-\beta\tilde{h}_2}\rho^{\frac{\beta}{C_1}\lfloor \frac{n}{\log n} \rfloor} & \text{if } \eta_{\mathcal{P}} = 0, \\ \gamma(\nu, \rho)\rho^{-\beta\tilde{h}_2}\left(\frac{\tilde{n}}{\log \tilde{n}}\right)^{-\frac{\beta}{\eta_{\mathcal{P}}}} & \text{if } \eta_{\mathcal{P}} > 0, \end{cases}$$

*where $\gamma(\nu, \rho) = \max\{\max\{\nu, l_f\}\rho^{(1-\beta)(m-1)-\tilde{h}_1}, l_g\}$,*

$C_1 = 3^d d^{d-m}(12\sqrt{m})^m C\rho^{-\eta_{\mathcal{P}}\tilde{h}_3}4^m\rho^{\eta_{\mathcal{P}}\tilde{h}_4}$ *and* $\tilde{n} = \lfloor n/\overline{\log} n \rfloor \eta_{\mathcal{P}} \log(1/\rho)/C_1$,

*with $\tilde{h}_2 = m + m\left\lceil \log_3 \frac{2\sqrt{m}\kappa_2}{\kappa_2 - 1} \right\rceil$, $\tilde{h}_1 = d\lceil \log_3 \sqrt{m}\alpha \rceil$ and $\kappa_2 = \frac{\sqrt{m}}{\sqrt{1 - \mathrm{dist}^2(\mathbf{A}, \hat{\mathbf{A}})}}$, where $\tilde{h}_3, \tilde{h}_4$ are from equations (3) and (6) respectively.*

When $\eta_{\mathcal{P}} > 0$, Algorithm 1 has $r_n = \tilde{O}(n^{-\beta/\eta_{\mathcal{P}}}) = \tilde{O}(n^{-(1+\frac{d-m}{2m-1})/\eta_{\mathcal{P}}})$ while default SequOOL would give $\tilde{O}(n^{-1/\eta_{\mathcal{P}}})$, showing our approach reduces the regret at a faster rate. The proof of this theorem follows from applying the SequOOL parameters derived in Corollary 9.7 to Theorem 5 of Bartlett et al. (2019). Detailed proof is in Appendix 11.20.

## 5. EXPERIMENTS

All implementation details, benchmark functions, and experiment scripts can be found at our GitHub repository: https://github.com/raja-sunkara/Learned-Partitions-SequOOL

We evaluate our algorithms against baselines from various optimization categories. These include Bayesian Optimization (REMBO (Wang et al., 2016), HesBO (Nayebi et al., 2019)), Evolutionary Algorithms (CMA-ES (Hansen et al., 2003)), Dual Annealing (Pincus, 1970), Optimistic Optimization (SOO, SequOOL, RESOO (Qian & Yu, 2016), DiRect), and Random Search. The optimization functions used in our experiments include Sphere, Rastrigin, Different Powers, Rosenbrock, Styblinski-Tang, Hartmann-6, Branin, Ellipsoid, Sharp Ridge, and the CUSTOM function defined as $1 + (x_1 - 1)^2 + \sum_{i=d-m+2}^{d}(x_i - 1)^4$.

In our experimental setup, we construct the multi-index function as $f(\mathbf{x}) = g(\mathbf{A}\mathbf{x})$, where $\mathbf{x} \in \mathbb{R}^d$, $g : \mathbb{R}^m \to \mathbb{R}$ is a standard optimization test function, and $\mathbf{A} \in \mathbb{R}^{m\times d}$ is a randomly generated matrix satisfying $\mathbf{A}\mathbf{A}^\top = \mathbf{I}_m$. The Branin and Hartmann-6 functions are defined in 2 and 6 dimensions respectively; thus, we choose $m = 2$ for Branin and $m = 6$ for Hartmann-6 when constructing the multi-index function. To further evaluate the efficacy of our algorithm, we conducted benchmarks on low-dimensional multi-index functions with $d = 5$ and $m = 2$. Additional experimental results are in Appendix 12.2.

Our algorithm demonstrates superior performance on the Rastrigin, Sphere and Styblinski-Tang functions, achieving

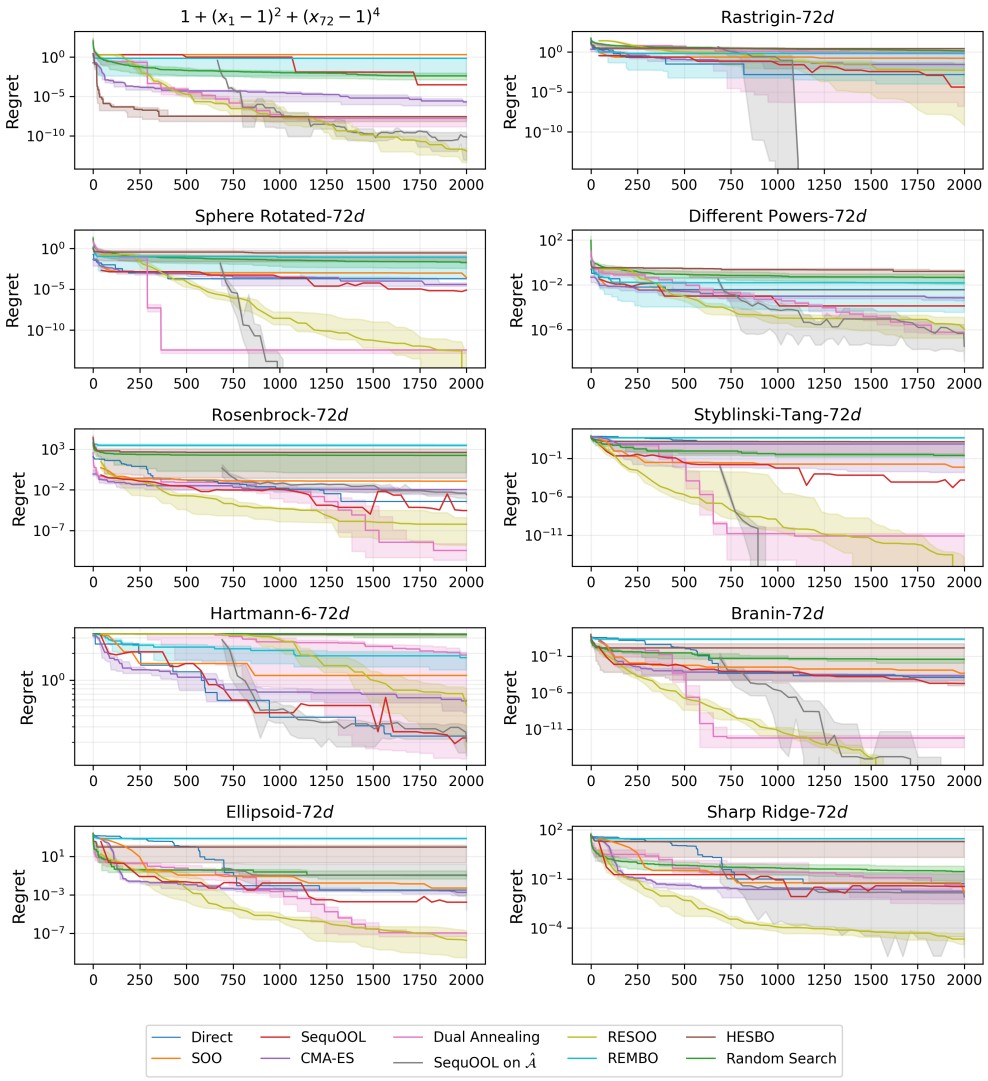

*Figure 3.* Regret Plots: Algorithm 1 (SequOOL on $\hat{\mathcal{A}}$) uses 650 additional samples to learn the subspace. Regret is plotted for 100 equally spaced budget values between 1 and 2000. For the randomized algorithms, we took 10 trials and plotted the median curve (thick line) and 0 and 95 percentile curves. Random Search is run on $\hat{\mathcal{A}}$.

zero regret with fewer samples compared to other methods. On the different powers function, we perform comparable to the other methods, however, on the $(x_1 - 1)^2 + (x_{72} - 1)^4$, Ellipsoid and Sharp Ridge function, our algorithm perform slightly worse than the RESOO. This may be attributed to the use of several (650) samples for the first stage in a 2000 budgeted experiment.

### 5.1. LLM Quantization

The AWQ (Lin et al., 2024) method for quantizing large language models formulates optimization problem as:

$$\alpha^* = \arg \min_{\alpha \in [0,1]} \mathcal{L}(\mathbf{s}_{\mathbf{X}}^{\alpha})$$

$$\mathcal{L}(\mathbf{s}) = \left\| Q(\mathbf{W} \cdot \mathbf{s})(\mathbf{s}^{-1} \cdot \mathbf{X}) - \mathbf{W}\mathbf{X} \right\|_2$$

Where $\mathbf{X}$ is the input features to the block which is cached from a calibration dataset. It uses the parameterization $\mathbf{s} = \mathbf{s}_{\mathbf{X}}^{\alpha}$, where $\mathbf{s}_{\mathbf{X}}$ is the activation scale computed from $\mathbf{X}$ and $\alpha \in [0, 1]$ and $Q$ as the quantization function and $\mathbf{W}$ as the original weights (full-precision). To determine the optimal $\alpha^*$, AWQ applies a 1D grid search over the interval

[0, 1]. This parameter controls the scale of activations and influences quantization error.

Each layer of the LLM contains three primary components: the attention matrices ($\mathbf{W}^Q$, $\mathbf{W}^K$, $\mathbf{W}^V$, and $\mathbf{W}^O$: all four matrices share a single $\alpha$ per layer), the first fully connected layer ($\mathbf{W}^{\texttt{fc1}}$: one $\alpha$ per layer), and the second fully connected layer ($\mathbf{W}^{\texttt{fc2}}$: one $\alpha$ per layer). Consequently, this leads to three optimization parameters per layer, resulting in a total of 3M parameters to optimize across M layers. Each of these parameters is derived from separate optimization problems, all of which are solved through the grid search method in the interval [0, 1] to find the optimal value of $\alpha^*$.

We propose a new approach which involves solving this LLM Quantization as high-dimensional black-box optimization problem. In our approach, we jointly optimize all layers to minimize perplexity: So, our approach has one optimization problem in $3M$ dimensional space, compared to AWQ which has $3M$ one-dimensional optimization problems. Let $\boldsymbol{\alpha} = [\alpha_1, ..., \alpha_{3M}]^\top$ represent the scales for all $M$ layers, with each layer having three parameters. We define our proposed optimization problem as:

$$\boldsymbol{\alpha}^* = \arg \min_{\boldsymbol{\alpha} \in [0,1]^{3M}} \mathrm{PPL}(\boldsymbol{\alpha})$$

Where, $\mathrm{PPL}(\boldsymbol{\alpha})$ is the perplexity on the calibration set after quantization using the scaling factors derived from $\boldsymbol{\alpha}$.

We evaluated our approach on the OPT-1.3B model (Zhang et al., 2022), with results presented in Table 2. Our proposed objective function using SequOOL over 72 dimensions outperformed AWQ, achieving lower perplexity on both WikiText-2 (Merity et al., 2016) and the calibration set (Pile dataset (Gao et al., 2020)). More details are in Appendix 12.3.

To quantify this improvement, we compare perplexity scores across methods. The perplexity of the unquantized model is 14.47, which we treat as a reference point for comparison. Quantization with the AWQ baseline increases perplexity to 16.92, resulting in a perplexity gap of 2.45. In contrast, our proposed method (Algorithm 2) achieves a perplexity of 16.68, reducing the gap to 2.21. This corresponds to a relative improvement of approximately 10% in perplexity degradation compared to AWQ, demonstrating the benefit of jointly optimizing the scaling factors across all layers.

## 6. DISCUSSION AND CONCLUSION

We have proposed an adaptive approach to learning a good partitioning scheme for black-box optimization. When the function belongs to a multi-index class, we prove that the adaptive partitioning scheme results in lower regret. To achieve this, we have developed a novel theoretical contribu-

*Table 2.* LLM Quantization Experiment Results

| Algorithm | Compute Time | WikiText-2 PPL | Calibration -Set PPL |
|---|---|---|---|
| Grid Search | ≈ 9 hours | 16.92 | 14.62 |
| SequOOL | ≈ 10 hours | 16.83 | **14.28** |
| Algorithm 1 | ≈ 10 hours | 16.96 | 14.42 |
| Algorithm 2 | ≈ 12 hours | **16.68** | 14.29 |

tion by relating the near-optimality dimensions of different partitioning schemes for the same function. Empirically, we observe that our proposed approach is better or comparable to existing methods over a wide range of benchmark functions and can be applied to high-dimensional functions. For example, on the Rastrigin function, SequOOL achieved a regret of $4.22 \times 10^{-5}$, while our method reached zero regret. Against RESOO, on both the Rastrigin and Styblinski-Tang functions, our method achieved zero regret compared to $5.5 \times 10^{-3}$ for RESOO on Rastrigin. For Styblinski-Tang, our method reached zero regret with approximately 900 evaluations, versus 2000 for RESOO.

Some limitations of our approach are estimating the low-dimensional subspace dimension and using an upper bound on the subspace distance in our algorithm. When the multi index assumption is not valid, we see only a mild benefit over SequOOL, e.g., in the AWQ experiments, our proposed objective function has function variation in all the directions and improvement over SequOOL is minimal. In the two-stage algorithm, the first stage chooses samples randomly for the subspace learning and these samples reduce the budget available for optimization.

There are several avenues for future work. We can extend our approach to the case when the function evaluations are noisy. We can consider other low-dimensional functions beyond the class of multi-index functions, e.g., functions with bounded second order variation in the Radon domain (Parhi & Nowak, 2022). Obtaining theoretical guarantees for direction selection strategy could be possible using active learning techniques, e.g., in (Mukherjee et al., 2022).

## Acknowledgements

This material is based upon work supported by the National Science Foundation under Grant No. 2246187. The authors thank the reviewers and meta-reviewers for their careful reading and insightful suggestions, which significantly improved the quality of the paper. Ardhendu Tripathy acknowledges the helpful discussions with Robert Nowak.

## Impact Statement

This paper presents work whose goal is to advance the field of Machine Learning. There are many potential societal consequences of our work, none which we feel must be specifically highlighted here.

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

## 7. Notation

Let $\mathbb{N}_0 = \mathbb{N} \cup \{0\}$ denote the set of non-negative integers and $[d]$ represents the set $\{1, 2, \ldots, d\}$. For vectors and matrices, we use $(\cdot)^\top$ to denote the transpose. For a $d$-dimensional vector $\mathbf{x} = [x_1, \ldots, x_d]^\top$, we use $\|\mathbf{x}\|_p$ and $\|\mathbf{x}\|_\infty$ to denote its $\ell_p$ and $\ell_\infty$ norm respectively. For any matrix $\mathbf{X} = [\mathbf{x}_1, \ldots, \mathbf{x}_n]$, $\|\mathbf{X}\|_2$ and $\|\mathbf{X}\|_F$ represent its spectral and Frobenius norms. We use $\sigma_i(\mathbf{X})$ for the $i^{th}$ largest singular value, with $\sigma_{\max}(\mathbf{X})$ and $\sigma_{\min}(\mathbf{X})$ denoting the largest and smallest singular values. Suppose $\mathbf{A} \in \mathbb{R}^{m \times d}$, with $\mathbf{A}\mathbf{A}^\top = \mathbf{I}_m$, then for any vector $\mathbf{v} \in \mathbb{R}^d$, we denote its orthogonal projection onto the span of the rows of $\mathbf{A}$ as $\mathbf{v}_\| = \mathbf{A}^\top \mathbf{A} \mathbf{v}$, with the orthogonal component given by $\mathbf{v}_\perp = \mathbf{v} - \mathbf{v}_\|$. For a matrix $\mathbf{W} \in \mathbb{R}^{p \times d}$, we denote $\mathbf{W}_\|$ and $\mathbf{W}_\perp$ as the projections applied to each row, i.e., $\mathbf{W}_\| = \mathbf{W}\mathbf{A}^\top \mathbf{A}$ and $\mathbf{W}_\perp = \mathbf{W} - \mathbf{W}_\|$. For a given scalar $\kappa > 0$, we denote $\kappa \mathcal{X}$ as the set $\{\kappa x : x \in \mathcal{X}\}$. By a partitioning scheme $\mathcal{P}$ with $\kappa$, we mean the domain for the partitioning scheme is the set $\{\kappa x : x \in [-1, 1]^d\}$. By the default partitioning scheme $\mathcal{P}$, we mean partitioning scheme $\mathcal{P}$ with $\kappa = 1$. To describe side-length of a hyper-rectangle, we use the notation $[3^{-\lfloor \frac{h+m-i}{m} \rfloor}]_{i=1}^m$. This expressions represents a vector of $m$ components, where each component is given by $3^{-\lfloor \frac{h+m-i}{m} \rfloor}$, with $i$ ranging from 1 to $m$.

## 8. Omitted details for Section 3

A single hidden layer neural network with $p$ hidden neurons maps input $\mathbf{x} \in \mathbb{R}^d$ to the scalar

$$\hat{y}(\mathbf{x}, \mathbf{W}, \mathbf{a}, \mathbf{b}) = \sum_{i=1}^p a_i \sigma(\mathbf{w}_i^\top \mathbf{x} + b_i), \tag{7}$$

where $\sigma$ is the non-linear activation function, $\mathbf{W}$ is the hidden layer weight matrix consisting of $p$ weight vectors denoted as $\mathbf{w}_i$, $b_i$ is the scalar bias for the $i$th hidden neuron, and $a_i$ are the components of the output layer weight vector. The following proposition shows that the class of single hidden layer neural networks can represent the important subspace of multi-index functions.

**Proposition 8.1.** *Consider a function $f(\mathbf{x}) = \sum_{i=1}^p v_i \sigma(\mathbf{w}_i^\top \mathbf{x} + b_i)$, where $\mathbf{x} \in \mathbb{R}^d$ and $\sigma$ is a non-linear function. Let $\mathbf{x} = \mathbf{P}\mathbf{x} + (\mathbf{I} - \mathbf{P})\mathbf{x}$, where $\mathbf{P}$ is the projection matrix that maps any $x \in \mathbb{R}^d$ to $\mathrm{Span}\{\mathbf{w}_1, \mathbf{w}_2, \ldots, \mathbf{w}_p\}$. Then $f(\mathbf{x}) = \sum_{i=1}^p v_i \sigma(\mathbf{w}_i^\top \mathbf{P}\mathbf{x} + b_i)$. And if $\mathbf{x}, \mathbf{x}' \in \mathbb{R}^d$ are such that $(\mathbf{x} - \mathbf{x}') \perp \mathrm{Span}\{\mathbf{w}_1, \mathbf{w}_2, \ldots, \mathbf{w}_p\}$, then $f(\mathbf{x}) = f(\mathbf{x}')$.*

We use SGD with weight decay to fit the neural network on the function evaluations obtained at uniform random locations in $\mathcal{X}$. The top $m$ right singular vectors of the learned weight matrix of the hidden layer is used as the estimated $\hat{\mathbf{A}}$ for obtaining the partitioning scheme $\hat{\mathcal{A}}$.

Algorithm 3 describes our lookahead direction selection strategy $\tau_h(\hat{f})$.

**Querying outside the domain and optimizing** $\hat{f}$   In practice, with our optimization domain set as $\mathcal{X} = [-1, 1]^d$, we may encounter situations during low-dimensional subspace optimization where $\mathbf{t} \in [-\alpha, \alpha]^m$ results in $\hat{\mathbf{A}}^\top \mathbf{t} \notin \mathcal{X}$. To ensure that $f$ can be evaluated in all cases, we employ a two-step approach. First, we attempt to solve the optimization problem: $\arg\min_{\mathbf{t}_c \in \mathbb{R}^{d-m}} \left\|\hat{\mathbf{A}}_c^\top \mathbf{t}_c\right\|_2$ subject to $\hat{\mathbf{A}}^\top \mathbf{t} + \hat{\mathbf{A}}_c^\top \mathbf{t}_c \in \mathcal{X}$, where $\hat{\mathbf{A}}_c$ consists of the remaining $d - m$ columns of $\hat{\mathbf{A}}$ that are not in $\hat{\mathbf{A}}$. If this optimization problem has no feasible solution, we then employ Euclidean projection onto $\mathcal{X}$: $\arg\min_{x \in \mathcal{X}} \left\|\mathbf{x} - \hat{\mathbf{A}}^\top \mathbf{t}\right\|_2$. This projection method is applied whenever $\hat{\mathbf{A}}^\top \mathbf{t} \notin \mathcal{X}$, ensuring that we always have a valid point within our optimization domain. In practice, we estimate the minimum of $\hat{f}$ on the domain $\mathcal{T}_{h+1}$ using random sampling or any of the other black-box optimization algorithms, since $\hat{f}$ is cheap to evaluate and gradients are also available.

## 9. Omitted details for Section 4

Proofs are in Section 11.

First, we start with relating the default partitioning scheme $\mathcal{P}$ with $\mathcal{A}$ partitioning scheme. To establish the relationship between the near-optimality dimensions of the two schemes, we first need to compare the parameters $(\nu, \rho)$ of SequOOL across both partitioning schemes. This requires a characterization of the cells $\mathcal{A}_h^*$ and $\mathcal{P}_h^*$. The following proposition provides this characterization.

**Proposition 9.1.** *Let $\kappa > 0$ and $\boldsymbol{\alpha}_h^* \in \mathbb{R}^m$ be the representative of the $\mathcal{A}_h^*$ cell containing a global maxima of the function. Using fraction of two vectors to denote component-wise division,*

---

**Algorithm 3** Implementing lookahead direction selection strategy $\tau_h(\hat{f})$

---

**Require:** Current partition tree $\mathcal{T}$, height $h$, estimated function $\hat{f}$

$\hat{\mathbf{x}}h^* \leftarrow \arg\max_i f(\mathbf{x}h, i)$, representative of cell $\mathcal{T}h$ with the largest function value at height $h$

$\mathcal{T}h \leftarrow$ cell at height $h$ in $\mathcal{T}$ whose representative is $\hat{\mathbf{x}}h^*$, $axis\_to\_split \leftarrow 0$, $minimum \leftarrow \infty$

**for** $i \leftarrow 1$ $m$ **do**

    $\mathcal{T}h + 1 \leftarrow$ child cell of $\mathcal{T}h$ after trisecting axis $i$ and having the same representative $\hat{\mathbf{x}}^*h$

    $temp \leftarrow$ Compute minimum of $\hat{f}$ on the domain $\mathcal{T}_{h+1}$

    **if** $temp$ ¡ $minimum$ **then**

        $axis\_to\_split \leftarrow i$

        $minimum \leftarrow temp$

    **end if**

**end for**

**return** $axis\_to\_split$

---

$$\mathcal{A}_h^* = \{\mathbf{A}^\top\boldsymbol{\alpha} : \boldsymbol{\alpha} \in \mathbb{R}^m, \left\|\frac{\boldsymbol{\alpha} - \boldsymbol{\alpha}_h^*}{\mathbf{s}}\right\|_\infty \leq \alpha \quad with \quad \mathbf{s} = [3^{-\left\lfloor\frac{h+m-i}{m}\right\rfloor}]_{i=1}^m\}. \tag{8}$$

*Similarly, if $\mathbf{x}_h^* \in \mathbb{R}^d$ is the representative of the $\mathcal{P}_h^*$ cell containing the same global maxima,*

$$\mathcal{P}_h^* = \{\mathbf{x} : \mathbf{x} \in \mathbb{R}^d, \left\|\frac{\mathbf{x} - \mathbf{x}_h^*}{\mathbf{c}}\right\|_\infty \leq \kappa \quad with \quad \mathbf{c} = [3^{-\left\lfloor\frac{h+d-i}{d}\right\rfloor}]_{i=1}^d\}. \tag{9}$$

Proposition 2.5 shows that we can use $\mathcal{A}$ partitioning scheme to perform optimization. We now relate the $\nu$ and $\rho$ parameters (see Assumption 2.6) of the partitioning schemes $\mathcal{P}$ and $\mathcal{A}$.

To establish relationships between the parameters of the partitioning schemes $\mathcal{A}$ and the default scheme $\mathcal{P}$, we need to connect the sets $\mathcal{P}_h^*$ and $\mathcal{A}_h^*$. The following lemma provides this connection:

**Lemma 9.2.** *Let the domain for $\mathcal{P}$ be $\kappa\mathbb{H}_1^d = \{\kappa\mathbf{x} \mid \mathbf{x} \in \mathbb{H}_1^d\}$ with $\kappa \geq 3^{\lceil\log_3\sqrt{m}\alpha\rceil}$. Suppose $\mathbf{x}^* \in \mathcal{P}_h^*$ is such that $\mathbf{x}^* = \mathbf{A}^\top\mathbf{A}\mathbf{x}^*$. Then, $\forall i \in [1 : m-1], \forall k \in \mathbb{N}_0$ we have that $\mathcal{A}_{km+i}^* \subseteq \mathcal{P}_{kd+i}^*$.*

Having established the relationship between the star cells of the partitioning schemes $\mathcal{A}$ and $\mathcal{P}$, we can now proceed to relate their respective parameters. The following two lemmas establish these relationships.

This lemma connects the parameters $(\nu, \rho)$ of the partitioning scheme $\mathcal{P}$ with $\kappa \geq 3^{\lceil\log_3\sqrt{m}\alpha\rceil}$ to the parameters $(\nu_\mathcal{A}, \rho_\mathcal{A})$ of the scheme $\mathcal{A}$.

**Lemma 9.3.** *Let the parameters for the partitioning schemes $\mathcal{P}, \mathcal{A}$ be $(\nu, \rho), (\nu_\mathcal{A}, \rho_\mathcal{A})$ respectively. If Lemma 9.2 is applicable and $\mathcal{P}$ satisfies Assumption 2.6. Then we have that $\nu_\mathcal{A} = \nu\rho^{(1-\beta)(m-1)}, \rho_\mathcal{A} = \rho^\beta$ where $\beta = 1 + \frac{d-m}{2m-1}$.*

This lemma establishes the relationship between the parameters $(\nu_\mathcal{A}, \rho_\mathcal{A})$ of scheme $\mathcal{A}$ and $(\nu, \rho)$ of the default partitioning scheme $\mathcal{P}$.

**Lemma 9.4.** *The parameters $(\nu_\mathcal{A}, \rho_\mathcal{A}), (\nu, \rho)$ associated with partitioning schemes $\mathcal{A}$ and $\mathcal{P}$ with $\kappa = 1$. Let $l_f = f^* - \inf_{\mathbf{x} \in \kappa_1\mathbb{H}_1^d} f(\mathbf{x})$. Then*

$$\rho_\mathcal{A} = \rho^\beta, \nu_\mathcal{A} = \max\{\nu, l_f\}\rho^{(1-\beta)(m-1)-\tilde{h}_1}$$

*where $\tilde{h}_1 = d\lceil\log_3\kappa_1\rceil, \beta = 1 + \frac{d-m}{2m-1}$ and $\kappa_1 = 3^{\lceil\log_3\sqrt{m}\alpha\rceil}$.*

The previous lemmas established relationships between the parameters of the partitioning schemes $\mathcal{A}$ and $\mathcal{P}$. They demonstrate that $\mathcal{A}$ is a valid partitioning scheme with a reduced sequOOL parameter $\rho_\mathcal{A}$ compared to the default partitioning scheme $\rho$.

## 9.1. Using $\hat{\mathcal{A}}$ defined over the estimated $\hat{A}$

We obtain $\hat{\mathbf{A}}$ using a subspace learning algorithm with evaluations of $f$ at selected points in $\mathcal{X}$. We then use $\hat{\mathbf{A}}$ to define a partitioning scheme $\hat{\mathcal{A}}$ (as per Definition 2.3) and apply SequOOL to it. The impact of using an estimated matrix $\hat{\mathbf{A}}$ instead

of the true matrix $\mathbf{A}$ in our optimization problem can be quantified using subspace distance. The following is the definition of the subspace distance.

**Definition 9.5.** ((Chen et al., 2021, Lemma 2.5)) Let $\mathbf{A}, \hat{\mathbf{A}} \in \mathbb{R}^{m \times d}$ consists of orthonormal rows such that $\mathbf{A}\mathbf{A}^\top = \mathbf{I}_m$ and $\hat{\mathbf{A}}\hat{\mathbf{A}}^\top = \mathbf{I}_m$. Define two more matrices $\mathbf{A}_\perp, \hat{\mathbf{A}}_\perp \in \mathbb{R}^{(d-m) \times d}$ such that $\begin{bmatrix} \mathbf{A}^\top & \mathbf{A}_\perp^\top \end{bmatrix}$ and $\begin{bmatrix} \hat{\mathbf{A}}^\top & \hat{\mathbf{A}}_\perp^\top \end{bmatrix}$ form two orthonormal bases for $\mathbb{R}^d$. Then the subspace distance between the two row subspaces $(\mathbf{A}, \hat{\mathbf{A}})$ is given by

$$\text{dist}\,(\mathbf{A}, \hat{\mathbf{A}}) = \left\| \mathbf{A}^\top \mathbf{A} - \hat{\mathbf{A}}^\top \hat{\mathbf{A}} \right\|_2 = \left\| \hat{\mathbf{A}}_\perp \mathbf{A}^\top \right\|_2 = \left\| \mathbf{A}_\perp \hat{\mathbf{A}}^\top \right\|_2 = \left\| \sin \Theta(\mathbf{A}, \hat{\mathbf{A}}) \right\|_2, \tag{10}$$

where $\|\cdot\|_2$ denotes the spectral norm, and $\sin \Theta$ is a diagonal matrix of $\{\sin(\arccos(\sigma_i)) : i = 1, 2, \ldots, m\}$ where $\sigma_i$ are the singular values of $\hat{\mathbf{A}}\mathbf{A}^\top$ in decreasing order.

Moreover, we have the following equality:

$$\sigma_{min}(\hat{\mathbf{A}}\mathbf{A}^\top) = \cos \theta_m = \sqrt{1 - \sin^2 \theta_m} = \sqrt{1 - \left\| \sin \Theta(\mathbf{A}, \hat{\mathbf{A}}) \right\|_2^2} = \sqrt{1 - \text{dist}^2(\mathbf{A}, \hat{\mathbf{A}})} \tag{11}$$

We observe that assuming $\text{dist}(\mathbf{A}, \hat{\mathbf{A}}) < 1$ implies that $\sigma_{min}(\hat{\mathbf{A}}\mathbf{A}^\top) \neq 0$ and $\text{rank}(\hat{\mathbf{A}}\mathbf{A}^\top) = m$.

Given that we only have an estimate $\hat{\mathbf{A}}$ of the true matrix $\mathbf{A}$, and we perform optimization on the subspace spanned by $\hat{\mathbf{A}}$, Lemma 4.5 guarantees that we can use $\hat{\mathbf{A}}$ and recover $f^*$. Now, we relate the SequOOL parameters between the partitioning schemes $\mathcal{A}$ and $\hat{\mathcal{A}}$ partitioning schemes.

**Lemma 9.6.** Let $\mathcal{A}$ and $\hat{\mathcal{A}}$ be the partitioning schemes defined in Definition 2.3. Suppose $\mathcal{A}$ satisfies Assumption 2.6 with parameters $\nu_\mathcal{A}, \rho_\mathcal{A}$. Let $l_g = g^* - \inf_{\mathbf{z} \in \kappa_2 \alpha \mathbb{H}_1^m} g(\mathbf{A}\hat{\mathbf{A}}^\top \mathbf{z})$. Then $\hat{\mathcal{A}}$ satisfies Assumption 2.6 with parameters

$$\nu_{\hat{\mathcal{A}}} = \max\{\nu_\mathcal{A}, l_g\} \rho_\mathcal{A}^{-\tilde{h}_2}, \rho_{\hat{\mathcal{A}}} = \rho_\mathcal{A}$$

where $\tilde{h}_2 = m + m \left\lceil \log_3 \frac{2\sqrt{m}\kappa_2}{\kappa_2 - 1} \right\rceil$ and $\kappa_2 = \frac{\sqrt{m}}{\sqrt{1 - \text{dist}^2(\mathbf{A}, \hat{\mathbf{A}})}}$.

The above lemma demonstrates that the partitioning scheme $\hat{\mathcal{A}}$ is valid and satisfies Assumption 2.6 and it establishes the relationship between the SequOOL parameters of the $\hat{\mathcal{A}}$ and $\mathcal{A}$ partitioning schemes.

This corollary synthesizes the relationships established in the preceding lemmas, providing a direct comparison between the estimated scheme $\hat{\mathcal{A}}$ and the default partitioning scheme $\mathcal{P}$. It combines the two-step process of relating $\mathcal{A}$ to $\mathcal{P}$ and then $\hat{\mathcal{A}}$ to $\mathcal{A}$, yielding a comprehensive set of relationships for the SequOOL parameters, near-optimality dimensions, and associated constants.

**Corollary 9.7.** Referring to Lemma 4.4, for the partitioning scheme $\mathcal{P}$ with $\kappa = 1$, we have $\rho_\mathcal{A} = \rho^\beta, \nu_\mathcal{A} = \max\{\nu, l_f\} \rho^{(1-\beta)(m-1) - \tilde{h}_1}, \eta_\mathcal{A}(\nu_\mathcal{A}, \rho_\mathcal{A}, C_\mathcal{A}) \leq \frac{\eta_\mathcal{P}(\nu, \rho, C)}{\beta}$ and $C_\mathcal{A} = 3^d d^{d-m} (12\sqrt{m})^m C \rho^{-\eta_\mathcal{P} \tilde{h}_3}$. By utilizing Lemma 9.6 and Lemma 4.7, we establish the following relationships among the parameters associated with $\hat{\mathcal{A}}$ and $\mathcal{P}$.

$$\rho_{\hat{\mathcal{A}}} = \rho^\beta, \nu_{\hat{\mathcal{A}}} = \max\{\max\{\nu, l_f\} \rho^{(1-\beta)(m-1) - \tilde{h}_1}, l_g\} \rho_\mathcal{A}^{-\tilde{h}_2},$$

$$\eta_{\hat{\mathcal{A}}}(\nu_{\hat{\mathcal{A}}}, \rho_{\hat{\mathcal{A}}}, C_{\hat{\mathcal{A}}}) \leq \frac{\eta_\mathcal{P}(\nu, \rho, C)}{\beta}, C_{\hat{\mathcal{A}}} = 3^d d^{d-m} (12\sqrt{m})^m C \rho^{-\eta_\mathcal{P} \tilde{h}_3} 4^m \rho^{\eta_\mathcal{P} \tilde{h}_4}.$$

with $\tilde{h}_2 = m + m \left\lceil \log_3 \frac{2\sqrt{m}\kappa_2}{\kappa_2 - 1} \right\rceil$ and $\tilde{h}_1 = d \left\lceil \log_3 3^{\lceil \log_3 \sqrt{m}\alpha \rceil} \right\rceil$ with $\kappa_2 = \frac{\sqrt{m}}{\sqrt{1 - \text{dist}^2(\mathbf{A}, \hat{\mathbf{A}})}}$

$$\tilde{h}_3 = -\left\lfloor \log_\rho(\max\{\nu, l_f\} \rho^{(1-\beta)(m-1) - \tilde{h}_1}) - \log_\rho(\nu) \right\rfloor, \tilde{h}_4 = -\left\lfloor \log_{\rho_\mathcal{A}}(\max\{\nu_\mathcal{A}, l_g\} \rho_\mathcal{A}^{-\tilde{h}_2}) - \log_{\rho_\mathcal{A}}(\nu_\mathcal{A}) \right\rfloor$$

(Mousavi-Hosseini et al., 2023), controls the closeness between true and estimated subspace expressed in terms of $\|\mathbf{W}_\perp\|_F$. However, for out Algorithm 2, we require an upper bound on the distance between the true subspace $\mathbf{A}$ and its estimate $\hat{\mathbf{A}}$. This lemma bridges this gap by providing an upper bound on $\text{dist}\,(\mathbf{A}, \hat{\mathbf{A}})$ in terms of $\|\mathbf{W}_\perp\|_F$ and the singular values of $\mathbf{W}$.

**Lemma 9.8.** *Given $\mathbf{A} \in \mathbb{R}^{m \times d}$ satisfying $\mathbf{A}\mathbf{A}^\top = \mathbf{I}_m$, let $\mathbf{W} \in \mathbb{R}^{p \times d}$ be any matrix such that $\mathrm{rank}(\mathbf{W}) \geq m$ and $d \geq m$. Consider the singular value decomposition of $\mathbf{W} = \mathbf{U}\mathbf{S}\mathbf{V}^\top$ and collect the top $m$ right singular vectors in the matrix $\hat{\mathbf{A}} = \begin{bmatrix} \mathbf{v}_1 & \mathbf{v}_2 & \cdots & \mathbf{v}_m \end{bmatrix}^\top$, where $\mathbf{v}_i$ is the ith column of $\mathbf{V}$. Recollect the definition of $\sin\Theta(\mathbf{A}, \hat{\mathbf{A}})$ and $\mathrm{dist}(\mathbf{A}, \hat{\mathbf{A}})$ from Definition 9.5 and the notation of $\mathbf{W}_\perp$ from section 7. Using $\|\cdot\|_F$ to denote Frobenius norm and $\sigma_m$ to denote the $m^{th}$ singular value of $\mathbf{W}$, we have that*

$$\mathrm{dist}(\mathbf{A}, \hat{\mathbf{A}}) = \left\| \sin\Theta(\mathbf{A}, \hat{\mathbf{A}}) \right\|_2 \leq \frac{\|\mathbf{W}_\perp\|_F}{\sigma_m}. \tag{12}$$

### 9.2. Supporting lemmas

The following lemmas and assumptions are needed to obtain guarantees on learning a good estimate $\hat{\mathbf{A}}$.

**Assumption 9.9.** (Mousavi-Hosseini et al., 2023) The student model is a two-layer neural network Equation (7) trained over the data set $\{(\mathbf{x}^{(i)}, y^{(i)})\}_{i \geq 1}$, where the target values $y^{(i)}$ are generated according to the teacher model Equation (1) and the inputs satisfy $x^{(i)} \overset{iid}{\sim} \mathcal{N}(\mathbf{0}, \mathbf{I}_d)$. The link function $g(\cdot)$ is weakly differentiable.

**Assumption 9.10.** (Mousavi-Hosseini et al., 2023) For all $1 \leq i \leq m, 1 \leq j \leq d$, we initialize the NN weights and biases with $\sqrt{d}\mathbf{W}_{ij}^0 \overset{iid}{\sim} \mathcal{N}(0,1), ma_i^0 \overset{iid}{\sim} \mathrm{Unif}([-1,1])$, and $b_i^0 \overset{iid}{\sim} \mathrm{Unif}(\{-1,1\})$

**Lemma 9.11** ((Mousavi-Hosseini et al., 2023), Theorem 3). *Consider running $T$ SGD iterations over samples satisfying Assumption 9.9, with an initialization satisfying Assumption 9.10, and using the following decaying step size schedule. Assuming ReLU non-linearity, let $\zeta := 2\sqrt{2/e\pi}$. Choose the decreasing step size $\eta_t = m\frac{2(t+t^*)+1}{\gamma(t+t^*+1)^2}, \tilde{\lambda} \geq \gamma + \zeta$ and $t^* \asymp \frac{\tilde{\lambda}}{\gamma}$ for any $\gamma > 0$. Then, for $\lambda = \frac{\tilde{\lambda}}{m}$, with probability at least $1 - \delta$,*

$$\frac{\|\mathbf{W}_\perp^\top\|_F}{\sqrt{m}} \lesssim \sqrt{\frac{(d + \log(1/\delta)}{\gamma^2 T}}$$

*whenever $m \gtrsim \log(1/\delta)$ and $T \gtrsim \frac{\tilde{\lambda}^2}{d+\log(1/\delta)}$.*

The following Lemma is to control the subspace distance using compressed sensing algorithm.

**Theorem 9.12** ((Fornasier et al., 2012), Theorem 4.1). *Let $f(\mathbf{x}) = g(\mathbf{A}\mathbf{x})$ be a function where $\mathbf{A}$ is a $k \times d$ matrix with orthonormal rows, and $g$ is a twice continuously differentiable function. Assume that $\mathbf{H}^f = \mathbf{A}^\top \mathbf{H}_g \mathbf{A}$ is well-conditioned with $\sigma_k(\mathbf{H}^f) \geq \alpha > 0$. Let $\hat{\mathbf{A}}$ be the matrix obtained from the Dantzig Selector approximation $\hat{\mathbf{X}}$ of the matrix $\mathbf{X}$ of gradients of $f$ at $m_X$ random points. Then, with high probability, the distance between the subspaces spanned by the rows of $\mathbf{A}$ and $\hat{\mathbf{A}}$ is bounded by:*

$$\left\| \mathbf{A}^\top \mathbf{A} - \hat{\mathbf{A}}^\top \hat{\mathbf{A}} \right\|_F \leq \frac{2\nu_2}{\sqrt{\alpha(1-s)} - \nu_2}$$

*where*

$$\nu_2 = Ck^{1/q}\left(\frac{m_\Phi}{\log(d/m_\Phi)}\right)^{1/2-1/q} + \frac{\epsilon k^2}{\sqrt{m_\Phi}}$$

*and $C$ is a constant depending on the parameters $C_1$ and $C_2$ from the conditions on $\mathbf{A}$ and $g$, $m_\Phi$ is the number of derivative directions, $\epsilon$ is the step size used in the finite difference approximation, $d$ is the ambient dimension, and $s \in (0,1)$ is a parameter.*

## 10. Illustrative experiments to motivate lookahead direction selection

Using a partitioning scheme with a lower near-optimality dimension can lead to a faster decrease in regret. Empirically, we observe that the regret for SequOOL applied for a budget of 200 evaluations of the function in the Example 2.11 was $5 \times 10^{-10}$ for the default partitioning scheme and $5.8 \times 10^{-12}$ for the direction selection strategy in Example (2.11). This example indicates that it can be beneficial to use a direction splitting strategy that adapts to the function being optimized.

Additionally, we evaluate the regret for different choices of $\mathbf{A}$, by parameterizing $\mathbf{A} = \begin{bmatrix} \cos\theta & -\sin\theta \\ \sin\theta & \cos\theta \end{bmatrix}$ and changing the rotation angle from $0$ to $\pi/8$ while keeping the direction selection strategy the same as in Example (2.11). Figure 5 shows that the regret varies significantly over the range of angles. This shows that minimizing the angle of discrepancy between $\mathbf{A}$ and the true directions of variation (which are the standard $x_1$ and $x_2$ axes in this example) is beneficial in reducing the regret.

Given a particular $\mathbf{A}$ and function $f$, we consider the question of identifying an appropriate direction selection strategy that minimizes $\mathcal{N}_{\mathcal{A}}(3\nu_{\mathcal{A}}\rho_{\mathcal{A}}^{h})$ at all heights. The two example strategies for $f(x_1, x_2) = 1 - |x_1| - x_2^2$ we have seen so far are the default round-robin (equivalently 1:1) splitting and the 2:1 splitting in Example(2.11). For an $\mathbf{A}$ with $\theta = \pi/48$, we minimize the number of near-optimal cells $\mathcal{N}_{\mathcal{A}}(3\nu_{\mathcal{A}}\rho_{\mathcal{A}}^{h})$ at different heights by choosing the best splitting ratio at each height and plot the ratios as the blue line in Figure 4. We see that as the height increases to infinity the optimal split ratio converges to 1. However, at lower heights, the optimal split ratio is greater than 1 and takes its maximum value 1.83 at $h = 3$.

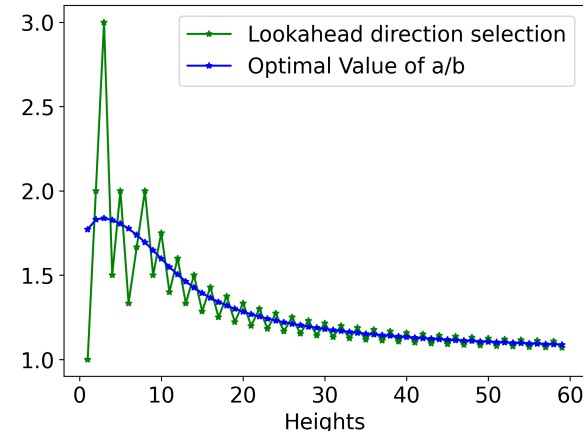

Figure 4. Optimal splitting ratio between the first and second directions. Green curve is obtained using the strategy in Definition 10.1

**Definition 10.1.** Lookahead strategy for direction selection. Given an estimated $\hat{f}$ and the current tree of partitions till depth $h$, the lookahead strategy first evaluates the different number of near-optimal cells for $\hat{f}$ at depth $h + 1$ by splitting along each of the $m$ directions. It then greedily selects the direction to be split at $h + 1$ as the direction that results in the lowest number of near-optimal cells.

In a numerical experiment, we see that the lookahead strategy closely matches the optimal split ratio (shown as green points in Fig 4) over all heights. This also results in it having a low regret than the default partitioning scheme. Empirically, we observe that the regret for SequOOL applied for a budget of $500$ evaluations of the function in the previous lemma was $5.68 \times 10^{-10}$ for the partitioning scheme $\mathcal{A}$ using the lookahead strategy for direction selection, $1.98 \times 10^{-5}$ for $\mathcal{A}$ with 1:1 splitting, and $1.8 \times 10^{-4}$ for $\mathcal{A}$ with the 2:1 splitting strategy from Example (2.11).

# 11. Proofs

Some of the facts which we use in our proofs.

Key inequalities for the matrix norms include: $\|\mathbf{x}\|_{\infty} \leq \|\mathbf{x}\|_{2} \leq \|\mathbf{x}\|_{1}$. Holder's inequality, applicable for $p, q \geq 1$ where $\frac{1}{p} + \frac{1}{q} = 1$, states that $|\mathbf{x}^{\top}\mathbf{y}| \leq \|\mathbf{x}\|_{p} \|\mathbf{y}\|_{q}$. The triangle inequality, valid for any $p \geq 1$, asserts that $\|\mathbf{x} + \mathbf{y}\|_{p} \leq \|\mathbf{x}\|_{p} + \|\mathbf{y}\|_{p}$.

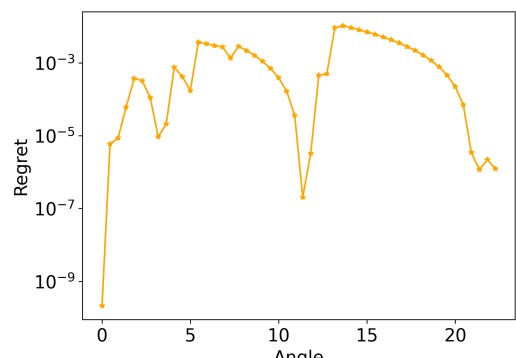

Figure 5. Regret at $n = 300$ for SequOOL on $\mathbf{A}$ with varying $\theta$.

Matrix Norms: For a matrix $\mathbf{A} \in \mathbb{R}^{m \times n}$, the operator norm is:

$$\|\mathbf{A}\|_{p} = \sup_{\mathbf{x} \neq 0} \frac{\|\mathbf{A}\mathbf{x}\|_{p}}{\|\mathbf{x}\|_{p}}$$

When $p = \infty$, $\|\mathbf{A}\|_{\infty} = \max_{1 \leq i \leq m} \sum_{j=1}^{n} |a_{ij}|$ and when $p = 2$, $\|\mathbf{A}\|_{2} = \sigma_{\max}(\mathbf{A})$, Where $\sigma_{\max}(\mathbf{A})$ represents the largest singular value of matrix $\mathbf{A}$. Additionally, the Frobenius norm is given by $\|\mathbf{A}\|_{F} = \sqrt{\sum_{i=1}^{m} \sum_{j=1}^{n} |a_{ij}|^{2}}$.

For a matrix $\mathbf{A} \in \mathbb{R}^{m \times n}$ of $\operatorname{rank} r$, the following inequalities hold:

$$\|\mathbf{A}\|_{2} \leq \|\mathbf{A}\|_{F} \leq \sqrt{r} \|\mathbf{A}\|_{2}, \frac{1}{\sqrt{n}} \|\mathbf{A}\|_{\infty} \leq \|\mathbf{A}\|_{2} \leq \sqrt{m} \|\mathbf{A}\|_{\infty} \tag{13}$$

For the operator norm $\|\cdot\|_2$, one has

$$\|\mathbf{AB}\|_2 \leq \|\mathbf{A}\|_2 \|\mathbf{B}\|_2, \|\mathbf{AB}\|_2 \geq \|\mathbf{A}\|_2 \sigma_{\min}(\mathbf{B}), \|\mathbf{AB}\|_2 \geq \|\mathbf{B}\|_2 \sigma_{\min}(\mathbf{A}) \tag{14}$$

Suppose a matrix $\mathbf{A}$ consists of orthonormal rows or columns, then $\|\mathbf{A}\|_2 = \left\|\mathbf{A}^\top\right\|_2 = 1$

### 11.1. Proof of Example 2.8

*Proof.* Consider the function $f(x_1, x_2) = g(\mathbf{Ax}) = 1-|x_1|$ with $\mathbf{A} = [1, 0]$ and $g(z) = 1-|z|$. Let $\mathcal{P}, \mathcal{A}$ be the partitioning schemes defined in Definition 2.2 with $\kappa = 1$ and parameters $(\nu, \rho), (\nu_\mathcal{A}, \rho_\mathcal{A})$ respectively. For the $\mathcal{P}$ partitioning scheme, along the $X$ axis, the side lengths of the children at depth $h$ are given by $3^{-\lceil h/2 \rceil}$ and along the $Y$ axis, it is $3^{-\lfloor h/2 \rfloor}$. Consider $f(x_1^*, x_2^*) - f(x_1, x_2) = |x_1|$, and the cell with the representative origin is the $\mathcal{P}_h^*$ cell, therefore

$$|x_1| = 3^{-\lceil h/2 \rceil} \leq 3^{-h/2} \leq (1/\sqrt{3})^h.$$

According to Assumption 2.6, the appropriate values are $\nu = 1, \rho = 1/\sqrt{3}$. Now consider a rectangle region $R$ with corners $\{(-3\nu\rho^h, -1), (3\nu\rho^h, -1), (3\nu\rho^h, 1), (-3\nu\rho^h, 1)\}$. Since $f^* - f(x_1, x_2) = |x_1|, \forall(x_1, x_2) \in R, f^* - f(x_1, x_2) \leq 3\nu\rho^h$. Thus any cell in $\mathcal{P}$ that has a non-empty intersection with $R$ is a near-optimal cell. Each cell at depth $h$ has an area of $3^{-\lceil h/2 \rceil}3^{-\lfloor h/2 \rfloor}$, therefore

$$\mathcal{N}_\mathcal{P}(3\nu\rho^h) \geq \frac{\text{Area}(R)}{\text{Area}(\mathcal{P}_{h,i})} = \frac{4(3(1/\sqrt{3})^h)}{3^{-\lceil h/2 \rceil}3^{-\lfloor h/2 \rfloor}},$$

yielding that since $\mathcal{N}_\mathcal{P}(3\nu\rho^h) = \Omega(\rho^{-h})$, from the Definition 2.7 we get $\eta_\mathcal{P} = 1$.

For the $\mathcal{A}$ partitioning scheme, we first note that $\alpha = 1$. Along $X$ axis, the side length of the children at depth $h$ is $3^{-h}$, therefore

$$f(x_1^*, x_2^*) - f(x_1, x_2) = |x_1| = 3^{-h} \leq (1/3)^h,$$

yielding the values are $\nu_\mathcal{A} = 1, \rho_\mathcal{A} = 1/3$. Now consider a line segment $L$ with endpoints $\{(-3\nu_\mathcal{A}\rho_\mathcal{A}^h, 0), (3\nu_\mathcal{A}\rho_\mathcal{A}^h, 0)\}$. Since $f^* - f(x_1, x_2) = |x_1|, \forall(x_1, 0) \in L, f^* - f(x_1, x_2) \leq 3\nu_\mathcal{A}\rho_\mathcal{A}^h$. Thus any cell in $\mathcal{A}$ that has an intersection with the $L$ is a near-optimal cell. Every $\mathcal{A}_{h,i}$ cell at depth $h$ has a length of $3^{-h}$, therefore,

$$\mathcal{N}_\mathcal{A}(3\nu_\mathcal{A}\rho_\mathcal{A}^h) \leq 2 + \frac{\text{len}(L)}{\text{len}(\mathcal{A}(h,i))} = 2 + \frac{2(3(1/3)^h)}{3^{-h}} = 8,$$

where the additional term 2 accounts for cells in $\mathcal{A}$ that partially intersect $L$ at its endpoints. Hence $\mathcal{N}_\mathcal{A}(3\nu_\mathcal{A}\rho_\mathcal{A}^h)$ is a constant and $\eta_\mathcal{A} = 0$. $\square$

### 11.2. Proof of Example 2.9

*Proof.* Consider the function $g(k) = 1 - |k|$ with $\mathbf{A} = [1, 1]$. Then, $f(x_1, x_2) = g(\mathbf{Ax}) = 1 - |x_1 + x_2|$. For the $\mathcal{P}$ partitioning scheme with $\kappa = 1$, along the $X$ axis, the side lengths of the children at depth $h$ along the $X, Y$ axes are $3^{-\lceil h/2 \rceil}, 3^{-\lfloor h/2 \rfloor}$ respectively. Consider $f(x_1^*, x_2^*) - f(x_1, x_2) = |x_1 + x_2|$, and the cell with the representative origin is the $\mathcal{P}_h^*$ cell, therefore

$$|x_1 + x_2| = 3^{-\lceil h/2 \rceil} + 3^{-\lfloor h/2 \rfloor} \leq 2 \cdot 3^{-(h-1)/2} \leq 2\sqrt{3}(1/\sqrt{3})^h.$$

According to Assumption 2.6, $\nu = 2\sqrt{3}, \rho = 1/\sqrt{3}$. Consider a rhombus region $T$ formed by coordinates $\{(-3\nu\rho^h, 0), (0, -3\nu\rho^h), (3\nu\rho^h, 0), (0, 3\nu\rho^h)\}$. Since $f^* - f(x_1, x_2) = |x_1 + x_2|, \forall(x_1, x_2) \in T, f^* - f(x_1, x_2) \leq 3\nu\rho^h$. This says, any cell in $\mathcal{P}$ that has an intersection with the $T$ is near-optimal cell. Each cell at depth $h$ has an area of $3^{-\lceil h/2 \rceil}3^{-\lfloor h/2 \rfloor}$, therefore,

$$\mathcal{N}_\mathcal{P}(3\nu\rho^h) \leq \left(1 + \frac{3\nu\rho^h}{3^{-\lceil h/2 \rceil}}\right)\left(1 + \frac{3\nu\rho^h}{3^{-\lfloor h/2 \rfloor}}\right) \leq O(1)$$

Hence $\mathcal{N}_\mathcal{P}(3\nu\rho^h)$ is independent of $h$ and $\eta_\mathcal{P} = 0$.

For the $\mathcal{A}$ partitioning scheme, along the X axis and Y axis, the side lengths of the children at depth $h$ is given by $3^{-h}$ and $3^{-h}$, giving that

$$f(x_1^*, x_2^*) - f(x_1, x_2) = |x_1 + x_2| = 2 \cdot 3^{-h} \leq 2(1/3)^h.$$

According to Assumption 2.6, the values are $\nu_\mathcal{A} = 2, \rho_\mathcal{A} = 1/3$.

Further, we see from Definition 2.4 that $\alpha = \sqrt{2}$ in this example. $\square$

### 11.3. Proof of Example 2.11

*Proof.* For the $\mathcal{P}$ partitioning scheme, the side lengths of the children at depth $h$ are as follows: along the $X$ axis, $3^{-\lceil h/2 \rceil}$. Along the $Y$ axis, it is $3^{-\lfloor h/2 \rfloor}$. Consider $f(x_1^*, x_2^*) - f(x_1, x_2) = |x_1| + x_2^2$, and the cell with the representative origin is the $\mathcal{P}_h^*$ cell, therefore

$$|x_1| + x_2^2 = 3^{-\lceil h/2 \rceil} + 3^{-2\lfloor h/2 \rfloor} \leq 3^{-\lfloor h/2 \rfloor} + 3^{-2\lfloor h/2 \rfloor}$$
$$\leq 2 \cdot 3^{-\lfloor h/2 \rfloor} \leq 2 \cdot 3^{-(h-1)/2} \leq 2\sqrt{3}(1/\sqrt{3})^h$$

According to Assumption 2.6, the appropriate values are $\nu = 2\sqrt{3}$, $\rho = 1/\sqrt{3}$ and $\nu\rho^h = 2\sqrt{3}(1/\sqrt{3})^h$.

Consider a region $R$ which is given by $|x_1| + x_2^2 \leq 3\nu\rho^h$. Since $f^* - f(x_1, x_2) = |x_1| + x_2^2, \forall (x_1, x_2) \in R, f^* - f(x_1, x_2) \leq 3\nu\rho^h$. Thus any cell in $\mathcal{P}$ that has a non-empty intersection with $R$ is a near-optimal cell. Each cell at depth $h$ has an area of $3^{-\lceil h/2 \rceil}3^{-\lfloor h/2 \rfloor}$, therefore

$$\mathcal{N}_{\mathcal{P}}(3\nu\rho^h) \geq \frac{\text{Area}(R)}{\text{Area}(\mathcal{P}_{h,i})}$$

Now, we compute the area of the region formed by the curve $|x_1| + x_2^2 = 3\nu\rho^h$ which is given by

$$4 \int_0^{\sqrt{3\nu\rho^h}} \int_0^{3\nu\rho^h - x_2^2} dx_1 dx_2 = 8/3(3\nu\rho^h)^{3/2}$$

Thus,

$$\mathcal{N}_{\mathcal{P}}(3\nu\rho^h) \geq \frac{\text{Area}(R)}{\text{Area}(\mathcal{P}_{h,i})} = \frac{8/3(6\sqrt{3}(\frac{1}{\sqrt{3}})^h)^{3/2}}{3^{-\lceil h/2 \rceil}3^{-\lfloor h/2 \rfloor}} = \Omega\left((\frac{1}{\sqrt{3}})^{-h/2}\right)$$

Hence, $\eta_{\mathcal{P}} \geq 0.5$

The $\mathcal{P}_h^*$ cell contains the point $(0, 0)$. Since $\mathcal{P}$ is an axis-aligned partitioning scheme, we can bound the number of cells directly above, i.e., having the same x coordinate of their representative as that of, $\mathcal{P}_h^*$ by the value $\sqrt{3\nu\rho^h}/3^{-\lfloor h/2 \rfloor}$. In a similar manner, we can bound the maximum number of cells having their representative's y coordinate to be the same as that of $\mathcal{P}_h^*$ by $3\nu\rho^h/3^{-\lceil h/2 \rceil}$. Since $\mathcal{P}$ is an axis aligned partitioning scheme, the previous two bounds imply that number of near optimal cells are upper bounded by the product of number of cells along $x$ axis times number of cells along $y$ axis. Thus,

$$\mathcal{N}_{\mathcal{P}}(3\nu\rho^h) \leq \left(1 + \left\lceil \frac{\sqrt{3 \cdot 2\sqrt{3}(1/\sqrt{3})^h}}{3^{-\lceil h/2 \rceil}} \right\rceil\right)\left(1 + \left\lceil \frac{3 \cdot 2\sqrt{3}(1/\sqrt{3})^h}{3^{-\lfloor h/2 \rfloor}} \right\rceil\right) = O\left((\frac{1}{\sqrt{3}})^{-h/2}\right)$$

Thus, $\eta_{\mathcal{P}} \leq 0.5$, hence $\eta_{\mathcal{P}} = 0.5$

For the $\mathcal{A}$ partitioning scheme, the side lengths of the children at depth $h$ are as follows: along the $X$ axis, $3^{-(h+1)}$ if $h$ is odd; otherwise, $3^{-h}$. Along the $Y$ axis, it is $3^{-\lfloor h/2 \rfloor}$. Let $h$ is even. consider $f(x_1^*, x_2^*) - f(x_1, x_2) = |x_1| + x_2^2$ and the cell with the representative origin is the $\mathcal{A}_h^*$ cell, therefore

$$|x_1| + x_2^2 = 3^{-h} + 3^{-2\lfloor h/2 \rfloor} \leq 3^{-h} + 3^{-2(h-1)/2} \leq 4 \cdot 3^{-h}$$

According to Assumption 2.6, we have $\nu_{\mathcal{A}} = 4$, $\rho_{\mathcal{A}} = 1/3$, and $3\nu_{\mathcal{A}}\rho_{\mathcal{A}}^h = 12 \cdot 3^{-h}$.

The $\mathcal{A}_h^*$ cell contains the point $(0, 0)$. Since $\mathcal{A}$ is an axis-aligned partitioning scheme, we can bound the number of cells directly above, i.e., having the same x coordinate of their representative as that of, $\mathcal{A}_h^*$ by the value $\sqrt{3\nu\rho^h}/3^{-h/2}$. In a similar manner, we can bound the maximum number of cells having their representative's y coordinate to be the same as that

of $\mathcal{P}_h^*$ by $3\nu\rho^h/3^{-h}$. Since $\mathcal{P}$ is an axis aligned partitioning scheme, the previous two bounds imply that number of near optimal cells are upper bounded by the product of number of cells along $x$ axis times number of cells along $y$ axis. Thus,

$$\mathcal{N}_\mathcal{A}(3\nu_\mathcal{A}\rho_\mathcal{A}^h) \leq \left(1 + \left\lceil \frac{\sqrt{3 \cdot 4 \cdot 3^{-h}}}{3^{-h/2}} \right\rceil\right)\left(1 + \left\lceil \frac{3 \cdot 4 \cdot 3^{-h}}{3^{-h}} \right\rceil\right) = 65$$

When $h$ is odd, following the same steps will give $\mathcal{N}_\mathcal{A}(\nu_\mathcal{A}\rho_\mathcal{A}^h) \leq 148$.

Hence, $\mathcal{N}_\mathcal{A}(\nu_\mathcal{A}\rho_\mathcal{A}^h)$ is a constant and $\eta_\mathcal{A} = 0$. $\qquad\square$

### 11.4. Proof of Proposition 2.5

*Proof.* We claim that $\mathbf{z}^* = \mathbf{A}\mathbf{x}^*$, where $\mathbf{x}^* \in \mathcal{X}$ is the optimizer of $f$. This choice of $\mathbf{z}^*$ satisfies $f(\mathbf{A}^\top \mathbf{z}^*) = f(\mathbf{x}^*)$ as required in the Lemma. we will now show that the infinity norm of this $\mathbf{z}^*$ is less than or equal to $\alpha$.

First, observe that for any $\mathbf{x} \in [-1, 1]^d$, we can express $\mathbf{x}$ as a convex combination of corner points:

$$\mathbf{x} = \sum_{j=1}^{2^d} \mathbf{c}_j \alpha_j, \quad \alpha_j \geq 0, \sum_{j=1}^{2^d} \alpha_j = 1 \tag{15}$$

Now, let's consider the infinity norm of $\mathbf{A}\mathbf{x}^*$:

$$\|\mathbf{A}\mathbf{x}^*\|_\infty = \left\|\sum_{j=1}^{2^d} \mathbf{A}\mathbf{c}_j \alpha_j\right\|_\infty \qquad \text{(Using Equation 15)}$$

$$\leq \sum_{j=1}^{2^d} \|\mathbf{A}\mathbf{c}_j \alpha_j\|_\infty \qquad \text{(Triangle Inequality of Norms)}$$

$$= \sum_{j=1}^{2^d} |\alpha_j| \|\mathbf{A}\mathbf{c}_j\|_\infty \qquad \text{(Absolute Homogeneity property of Norms)}$$

$$\leq \alpha \sum_{j=1}^{2^d} |\alpha_j| \leq \alpha \qquad \text{(From the definition of } \alpha \text{ in the Proposition statement)}$$

Therefore, we have shown that $\|\mathbf{z}^*\|_\infty = \|\mathbf{A}\mathbf{x}^*\|_\infty \leq \alpha$, which completes the proof. $\qquad\square$

### 11.5. Proof of Proposition 9.1

*Proof.* Let $\mathbf{x}_h^*$ be the representative point (midpoint) of the $\mathcal{P}_h^*$ cell and $\mathbf{x}$ is within the $\mathcal{P}_h^*$ cell.

Let $\mathbf{x}$ be a point in a hyperrectangle centered at $\mathbf{x}_h^*$ with side lengths $2\kappa\mathbf{c}$, where $\mathbf{c}$ is a vector of side lengths. Then, this point $\mathbf{x}$ satisfies $|x_i - x_{h,i}^*| \leq \kappa c_i$ for all $i \in \{1, \ldots, d\}$. Equivalently, this set of inequalities can be written as $\max_{i\in\{1,\ldots,d\}} \left|\frac{x_i - x_{h,i}^*}{c_i}\right| \leq \kappa$. Using the infinity norm, we can concisely express this conditions as $\left\|\frac{\mathbf{x} - \mathbf{x}_h^*}{\mathbf{c}}\right\|_\infty \leq \kappa$, where the division is performed element-wise.

For the partition scheme, we perform trisection along each axis in a round-robin manner. After $h$ iterations, the side lengths are given by:

$$c_i = \kappa 3^{-\lfloor \frac{h+d-i}{d} \rfloor}, \quad i \in \{1, \ldots, d\} \tag{16}$$

where $\lfloor \frac{h+d-i}{d} \rfloor$ represents the number of trisections applied to dimension $i$. Therefore, the cell $\mathcal{P}_h^*$ can be described as:

$$\mathcal{P}_h^* = \left\{\mathbf{x} \in \mathbb{R}^d : \left\|\frac{\mathbf{x} - \mathbf{x}_h^*}{\mathbf{c}}\right\|_\infty \leq \kappa \quad \text{with} \quad \mathbf{c} = [3^{-\lfloor \frac{h+d-i}{d} \rfloor}]_{i=1}^d.\right\} \tag{17}$$

From the Definition 2.3, we have $\mathcal{A}_h^* \triangleq \{\mathbf{A}^\top \boldsymbol{\alpha} : \boldsymbol{\alpha} \in \mathcal{T}_h^*\}$ and

$$\mathcal{T}_h^* = \left\{ \boldsymbol{\alpha} \in \mathbb{R}^m : \left\| \frac{\boldsymbol{\alpha} - \boldsymbol{\alpha}_h^*}{\mathbf{s}} \right\|_\infty \leq \alpha \quad \text{with} \quad \mathbf{s} = [3^{-\lfloor \frac{h+m-i}{m} \rfloor}]_{i=1}^d. \right\} \tag{18}$$

Hence,

$$\mathcal{A}_h^* = \left\{ \mathbf{A}^\top \boldsymbol{\alpha} : \boldsymbol{\alpha} \in \mathbb{R}^m, \left\| \frac{\boldsymbol{\alpha} - \boldsymbol{\alpha}_h^*}{\mathbf{s}} \right\|_\infty \leq \alpha \quad \text{with} \quad \mathbf{s} = [3^{-\lfloor \frac{h+m-i}{m} \rfloor}]_{i=1}^d. \right\} \tag{19}$$

$\square$

### 11.6. Proof of Lemma 9.2

*Proof.* Following the Definition 2.2, consider the partitioning schemes $\mathcal{P}$ and $\mathcal{A}$.

According to Proposition 9.1, we can express the cell $\mathcal{A}_h^*$ as:

$$\mathcal{A}_h^* = \left\{ \mathbf{A}^\top \boldsymbol{\alpha} : \boldsymbol{\alpha} \in \mathbb{R}^m, \left\| \frac{\boldsymbol{\alpha} - \boldsymbol{\alpha}_h^*}{\mathbf{s}} \right\|_\infty \leq \alpha \quad \text{with} \quad \mathbf{s} = [3^{-\lfloor \frac{h+m-i}{m} \rfloor}]_{i=1}^m \right\}$$

At depth $h = km + i$, for $i \in [1 : m-1]$ and $k \in \mathbb{N}_0$, the side lengths simplifies to: $\mathbf{s} = [3^{-\mathbb{1}(j \leq i) - k}]_{j=1}^m$ Similarly,

$$\mathcal{P}_h^* = \left\{ \mathbf{x} : \mathbf{x} \in \mathbb{R}^d, \left\| \frac{\mathbf{x} - \mathbf{x}_h^*}{\mathbf{c}} \right\|_\infty \leq 1 \right\} \quad \text{with} \quad \mathbf{c} = [3^{-\lfloor \frac{h+d-i}{d} \rfloor}]_{i=1}^d.$$

At depth $h = kd + i$, for $i \in [1 : d-1]$ and $k \in \mathbb{N}_0$, the side lengths for $\mathcal{P}_h^*$ become: $\mathbf{c} = [3^{-\mathbb{1}(j \leq i) - k}]_{j=1}^d$.

And let us consider another partitioning scheme $\mathcal{G} = \kappa \mathcal{P}$. Let us denote $\tilde{\mathbf{x}}_h^*$ to be the representative of the $\mathcal{G}_h^*$ cell. Using Proposition 9.1, the cell $\mathcal{G}_h^*$ can be written as

$$\mathcal{G}_{kd+i}^* = \left\{ \mathbf{x} : \mathbf{x} \in \mathbb{R}^d, \left\| \frac{\mathbf{x} - \tilde{\mathbf{x}}_{kd+i}^*}{\mathbf{c}} \right\|_\infty \leq \kappa \right\} \quad \text{with} \quad \mathbf{c} = [3^{-\mathbb{1}(j \leq i) - k}]_{j=1}^d$$

Consider an element $\mathbf{x} = \mathbf{A}^\top \boldsymbol{\alpha} \in \mathcal{A}_{km+i}^*$, we now proceed with the following chain of inequalities:

$$
\begin{aligned}
\left\| \mathbf{x} - \tilde{\mathbf{x}}_{kd+i}^* \right\|_\infty &\leq \left\| \mathbf{x} - \mathbf{x}^* \right\|_\infty + \left\| \mathbf{x}^* - \tilde{\mathbf{x}}_{kd+i}^* \right\|_\infty && \text{(Vector Norm property)} \\
&= \left\| \mathbf{A}^\top \boldsymbol{\alpha} - \mathbf{A}^\top \boldsymbol{\alpha}^* \right\|_\infty + \left\| \mathbf{x}^* - \tilde{\mathbf{x}}_{kd+i}^* \right\|_\infty && (\mathbf{x}^* = \mathbf{A}^\top \boldsymbol{\alpha}^*) \\
&= \left\| \mathbf{A}^\top (\boldsymbol{\alpha} - \boldsymbol{\alpha}^*) \right\|_\infty + \left\| \mathbf{x}^* - \tilde{\mathbf{x}}_{kd+i}^* \right\|_\infty && \\
&\leq \left\| \mathbf{A}^\top \right\|_\infty \left\| \boldsymbol{\alpha} - \boldsymbol{\alpha}^* \right\|_\infty + \left\| \mathbf{x}^* - \tilde{\mathbf{x}}_{kd+i}^* \right\|_\infty && \text{(Matrix Norm definition)} \\
&\leq \sqrt{m} 3^{-k} \alpha + 3^{-k} \kappa && \text{(From the Matrix Inequality 13)} \\
&\leq 2 \cdot 3^{-k} \kappa && (\text{ From the Lemma statement, } \kappa \geq 3^{\lceil \log_3 \sqrt{m} \alpha \rceil}) \\
&\leq 3^{-(k-1)} \kappa &&
\end{aligned}
$$

This sequence of inequalities demonstrates that $\mathbf{x} \in \mathcal{G}_{(k-1)d}^*$. Moreover, we know that $\mathcal{G}_{(k-1)d}^* \subseteq \mathcal{G}_{kd+i}^*$. Therefore, partitioning scheme $\mathcal{P}$ with $\kappa \geq 3^{\lceil \log_3 \sqrt{m} \alpha \rceil}$ will satisfy:

$$\mathcal{A}_{km+i}^* \subseteq \mathcal{P}_{kd+i}^* \quad \forall i \in [1 : m-1], \forall k \in \mathbb{N}_0$$

$\square$

### 11.7. Proof of Lemma 9.3

*Proof.* Since Lemma 9.2 is assumed to be applicable, consider a partitioning scheme $\mathcal{P}$ with $\kappa = 3^{\lceil \log_3 \sqrt{m}\alpha \rceil}$ and suppose $f_1^* = \sup_{\mathbf{x} \in \kappa \mathbb{H}_1^d} f(\mathbf{x})$. Let $\mathcal{P}$ satisfies Assumption 2.6, then, there exist constants $\nu$ and $0 < \rho < 1$ such that

$$\sup_{\mathbf{x} \in \mathcal{P}_h^*} (f_1^* - f(\mathbf{x})) \leq \nu \rho^h. \quad \forall h \in \mathbb{N}_0 \tag{20}$$

Since $\kappa \geq 3^{\lceil \log_3 \sqrt{m}\alpha \rceil}$, we can incorporate Lemma 9.2, which gives,

$$\mathcal{A}_{km+i}^* \subseteq \mathcal{P}_{kd+i}^* \quad \forall i \in [1:m-1], \forall k \in \mathbb{N}_0$$

Therefore, we get

$$\sup_{\mathbf{x} \in \mathcal{A}_{km+i}^*} (f^* - f(\mathbf{x})) \leq \sup_{\mathbf{x} \in \mathcal{P}_{kd+i}^*} (f^* - f(\mathbf{x})) \quad \forall i \in [1:m-1], \forall k \in \mathbb{N}_0 \tag{21}$$

Combining Equations (20) and (21), and using the fact $f^* \leq f_1^*$, we have

$$\sup_{\mathbf{x} \in \mathcal{A}_{km+i}^*} (f^* - f(\mathbf{x})) \leq \sup_{\mathbf{x} \in \mathcal{P}_{kd+i}^*} (f^* - f(\mathbf{x})) \leq \nu \rho^{kd+i} \leq \nu (\rho^{\frac{kd+i}{km+i}})^{km+i} \tag{22}$$

Therefore, we have

$$\sup_{\mathbf{x} \in \mathcal{A}_{km+i}^*} (f^* - f(\mathbf{x})) \leq \nu (\rho^{\frac{kd+i}{km+i}})^{km+i} \quad \forall i \in [1:m-1], \forall k \in \mathbb{N}_0 \tag{23}$$

Consider $\rho_\mathcal{A} = \rho^\beta$ where

$$\beta = \min \left\{ \frac{kd+i}{km+i} \;\middle|\; i \in [1:m-1], \; k \in \mathbb{N} \right\} \tag{24}$$

Then the sequence of inequalities $\sup_{\mathbf{x} \in \mathcal{A}_{km+i}^*} (f^* - f(\mathbf{x})) \leq \nu (\rho^\beta)^{km+i} \quad \forall i \in [1:m-1], \forall k \in \mathbb{N}$ ensures that (23) is satisfied for all heights $h \geq m$. We show that $\beta = \frac{d+m-1}{2m-1}$ is the solution to (24). We consider the sequence for $k = 1$:

$$t_i = \frac{d+i}{m+i}, \quad \forall i \in [1, m-1]$$

First, we show that this sequence is decreasing. For any $i$ in the range $[2, m-2]$, consider the difference between consecutive terms:

$$t_{i+1} - t_i = \frac{d+(i+1)}{m+(i+1)} - \frac{d+i}{m+i} = \frac{m-d}{(m+i)(m+i+1)}$$

Since $m > d$, we have $m - d > 0$, but the denominator $(m+i)(m+i+1)$ is positive. Thus, $t_{i+1} < t_i$, which says that sequence $t_i$ is decreasing and the minimum value of $t_i$ occurs at $i = m - 1$:

$$t_{m-1} = \frac{d+(m-1)}{m+(m-1)} = \frac{d+m-1}{2m-1}$$

We now consider the general form for any $k \in \mathbb{N}$:

$$\beta = \min \left\{ \frac{kd+i}{km+i} \;\middle|\; i \in [1:m-1] \right\}$$

With the observation that

$$\frac{(k+1)d+m-1}{(k+1)m+m-1} \geq \frac{kd+m-1}{km+m-1}$$

we get $\beta = (d+m-1)/(2m-1)$. With this choice of $\beta$ we have

$$\sup_{\mathbf{x} \in \mathcal{A}_{km+i}^*} (f^* - f(\mathbf{x})) \leq \nu (\rho^{\frac{kd+i}{km+i}})^{km+i} \leq \nu (\rho_\mathcal{A}^\beta)^{km+i} \quad \forall i \in [1:m-1], \forall k \in \mathbb{N} \tag{25}$$

Now, we choose $\nu_{\mathcal{A}}$ to satisfy (23) for the first $m-1$ heights. From Equation 22,

$$\sup_{\mathbf{x} \in \mathcal{A}_i^*} (f^* - f(\mathbf{x})) \le \nu \rho^i = \nu \rho^{i - \beta i} \rho^{\beta i} \quad \forall i \in [1:m-1] \tag{26}$$

$$\le \nu \rho^{(1-\beta)(m-1)} \rho^{\beta i} \quad \forall i \in [1:m-1] \tag{27}$$

Therefore, $\nu_{\mathcal{A}} = \nu \rho^{(1-\beta)(m-1)}$ and $\rho_{\mathcal{A}} = \rho^\beta$ will satisfy the Assumption 2.6 for the $\mathcal{A}$ partitioning scheme.

$\square$

### 11.8. Proof of Lemma 9.4

*Proof.* Consider partitioning schemes $\mathcal{P}$ ($\kappa = 1$) and $\mathcal{G}$ ($\kappa = \kappa_1 = 3^{\lceil \log_3 \sqrt{m}\alpha \rceil}$) with star cells:

$$\mathcal{P}_h^* = \{\mathbf{x} : \mathbf{x} \in \mathbb{R}^d, \left\| \frac{\mathbf{x} - \mathbf{x}_h^*}{\mathbf{c}} \right\|_\infty \le 1\}, \mathcal{G}_h^* = \{\mathbf{x} : \mathbf{x} \in \mathbb{R}^d, \left\| \frac{\mathbf{x} - \tilde{\mathbf{x}}_h^*}{\mathbf{c}} \right\|_\infty \le \kappa\}$$

. where $\mathbf{c} = [3^{-\lfloor \frac{h+d-i}{d} \rfloor}]_{i=1}^d$

At height $\tilde{h}_1 = d \lceil \log_3 \sqrt{m}\alpha \rceil$, there are $\lceil \log_3 \sqrt{m}\alpha \rceil$ divisions along each of the $d$-axis. Since at each division the cell is divided into three parts, after $\lceil \log_3 \sqrt{m}\alpha \rceil$ divisions, side length along any axis becomes $\kappa_1 / 3^{\lceil \log_3 \sqrt{m}\alpha \rceil} = 1$. Since we are focusing on the $\mathbf{x}^*$ which lies in the domain $\mathcal{P}_0^*$, we have $\mathcal{G}_{\tilde{h}_1}^* = \mathcal{P}_0^*$. Subsequently, we deduce that $\mathcal{G}_{h+\tilde{h}_1}^* \subseteq \mathcal{P}_h^*$.

implying:

$$\sup_{\mathbf{x} \in \mathcal{G}_{\tilde{h}_1 + h}^*} (f^* - f(\mathbf{x})) = \sup_{\mathbf{x} \in \mathcal{P}_h^*} (f^* - f(\mathbf{x})) \quad \forall h \in \mathbb{N}_0 \tag{28}$$

By lemma assumption $\mathcal{P}$ satisfies Assumption 2.6, with parameters $(\nu, \rho)$. Therefore we have

$$\sup_{\mathbf{x} \in \mathcal{P}_h^*} (f^* - f(\mathbf{x})) \le \nu \rho^h \quad \forall h \in \mathbb{N}_0$$

Therefore,

$$\sup_{\mathbf{x} \in \mathcal{G}_{\tilde{h}_1 + h}^*} (f^* - f(\mathbf{x})) \le \nu \rho^h = \nu \rho^{-\tilde{h}_1} \rho^{\tilde{h}_1 + h} \quad \forall h \in \mathbb{N}_0 \tag{29}$$

For depths $h \in [1 : \tilde{h}_1 - 1]$:

$$\sup_{\mathbf{x} \in \mathcal{G}_h^*} (f^* - f(\mathbf{x})) \le f^* - \inf_{\mathbf{x} \in \kappa_1 \mathbb{H}_1^d} f(\mathbf{x})$$

$$\le (f^* - \inf_{\mathbf{x} \in \kappa_1 \mathbb{H}_1^d} f(\mathbf{x})) \rho^{-\tilde{h}_1} \rho^h$$

Using inequality 29 and the above inequality, we conclude that $\mathcal{G}$ satisfies Assumption 2.6 with parameters $(\rho, \max\{\nu, f^* - \inf_{\mathbf{x} \in \kappa_1 \mathbb{H}_1^d} f(\mathbf{x})\} \rho^{-\tilde{h}_1})$.

By Lemma 9.3 and $\kappa \ge 3^{\lceil \log_3 \sqrt{m}\alpha \rceil}$, we conclude:

$$\rho_{\mathcal{A}} = \rho^\beta, \nu_{\mathcal{A}} = \max\{\nu, f^* - \inf_{\mathbf{x} \in \kappa_1 \mathbb{H}_1^d} f(\mathbf{x})\} \rho^{(1-\beta)(m-1) - \tilde{h}_1}$$

$\square$

### 11.9. Proof of Lemma 4.2

*Proof.* Let $B^\circ(\mathbf{x}, r)$ be the open ball corresponding to the closed ball $B(\mathbf{x}, r)$. We will first demonstrate that $B^\circ(\mathbf{c}_i, \kappa') \cap B^\circ(\mathbf{c}_j, \kappa') = \emptyset \quad \forall i \ne j$. Suppose there exists a point $\mathbf{y} \in B^\circ(\mathbf{c}_i, \kappa') \cap B^\circ(\mathbf{c}_j, \kappa')$, then:

$$\|\mathbf{c}_i - \mathbf{c}_j\|_\infty \le \|\mathbf{c}_i - \mathbf{y}\|_\infty + \|\mathbf{c}_j - \mathbf{y}\|_\infty < 2\kappa'.$$

However, for lattice points, we have $\|\mathbf{c}_i - \mathbf{c}_j\|_\infty \geq 2\kappa'$, contradicting the above inequality.

Next, we show that

$$B(\mathbf{c}_i, \kappa') \subseteq (\kappa + 2\kappa')\mathbb{H}_1^m \quad \forall \mathbf{c}_i \in \mathcal{C}. \tag{30}$$

For any $\mathbf{y} \in B(\mathbf{c}_i, \kappa')$ we have that:

$$
\begin{aligned}
\|\mathbf{y}\|_\infty &\leq \|\mathbf{y} - \mathbf{c}_i\|_\infty + \|\mathbf{c}_i\|_\infty \\
&\leq \kappa' + \|\mathbf{c}_i\|_\infty \quad \text{(since } \mathbf{y} \in B(\mathbf{x}_i, \kappa') \\
&= \kappa' + \|\mathbf{c}_i - \mathbf{z} + \mathbf{z}\|_\infty \qquad\qquad \text{(for some } \mathbf{z} \in B(\mathbf{0}, \kappa) \cap B(\mathbf{c}_i, \kappa') \neq \emptyset) \\
&\leq \kappa' + \|\mathbf{c}_i - \mathbf{z}\|_\infty + \|\mathbf{z}\|_\infty \\
&\leq \kappa' + \kappa' + \kappa \\
&= \kappa + 2\kappa'.
\end{aligned}
$$

Therefore, $\mathbf{y} \in (\kappa + 2\kappa')\mathbb{H}_1^m$, which proves (30). Let $N$ be the number of balls $B(\mathbf{c}_i, \kappa')$. Since $B^\circ(\mathbf{x}_i, \kappa')$ are disjoint and all these balls are contained in $(\kappa + 2\kappa')\mathbb{H}_1^m$, we have:

$$
\begin{aligned}
N \cdot \text{Vol}(B^\circ(\mathbf{x}_i, \kappa')) &\leq \text{Vol}((\kappa + 2\kappa')\mathbb{H}_1^m) \\
N \cdot (2\kappa')^m &\leq (\kappa + 2\kappa')^m,
\end{aligned}
$$

giving that $N \leq \left(2 + \frac{\kappa}{\kappa'}\right)^m$. For the lower bound to $N$, since, $B(\mathbf{0}, \kappa) \subseteq \bigcup_i B(\mathbf{c}_i, \kappa')$, we have that

$$\text{Vol}(B(\mathbf{0}, \kappa)) = (2\kappa)^m \leq \text{Vol}\left(\bigcup_i B(\mathbf{c}_i, \kappa')\right)$$

$$\leq \sum_{i=1}^N \text{Vol}(B(\mathbf{c}_i, \kappa')) = N\text{Vol}(B(\mathbf{c}_0, \kappa')) = N(2\kappa')^m.$$

Hence $N \geq \left(\frac{\kappa}{\kappa'}\right)^m$. $\qquad\square$

**Lemma 11.1.** *Let* $P = [-1/\sqrt{d}, 1/\sqrt{d}]^d$, $P_{rot} = \{\mathbf{Q}^\top \mathbf{x} : \mathbf{x} \in P\}$*, where* $\mathbf{Q} = \begin{bmatrix} \mathbf{A}^\top & \mathbf{A}_\perp^\top \end{bmatrix}$ *is an orthonormal matrix, with* $\mathbf{A} \in \mathbb{R}^{m \times d}$ *and* $\mathbf{A}_\perp \in \mathbb{R}^{(d-m) \times d}$*. Denote,* $\text{Proj}_{\mathbf{A}_\perp}(S) = \{\boldsymbol{\beta} \in \mathbb{R}^{d-m} : \boldsymbol{\beta} = \mathbf{A}_\perp \mathbf{x}, \mathbf{x} \in S \subseteq \mathbb{R}^d\}$ *and*

$$\boldsymbol{\alpha}_{\max}' \triangleq \begin{bmatrix} \max_{1 \leq j \leq 2^d} \mathbf{q}_1^\top \mathbf{c}_j, & \cdots, & \max_{1 \leq j \leq 2^d} \mathbf{q}_m^\top \mathbf{c}_j, & \max_{1 \leq j \leq 2^d} \mathbf{q}_{m+1}^\top \mathbf{c}_j, & \cdots, & \max_{1 \leq j \leq 2^d} \mathbf{q}_d^\top \mathbf{c}_j \end{bmatrix}^\top,$$

*where* $\mathbf{c}_j$ *are corners of hypercube* $P$*. Then:*

$$\text{Vol}(\text{Proj}_{\mathbf{A}_\perp}(P_{rot})) = \prod_{i=m+1}^d 2\boldsymbol{\alpha}_{\max,i}' \quad \text{and} \quad \text{Vol}(\text{Proj}_{\mathbf{A}_\perp}([-1, 1]^d)) \geq \prod_{i=m+1}^d 2\boldsymbol{\alpha}_{\max,i}'.$$

*Proof.* The projection of $P_{\text{rot}}$ onto the subspace orthogonal to $\mathbf{A}$ is determined by:

$$\text{Proj}_{\mathbf{A}_\perp}(P_{\text{rot}}) = \left\{ \mathbf{A}_\perp^\top \mathbf{y} : \mathbf{y} \in P_{\text{rot}} \right\}.$$

Let us denote $\boldsymbol{\alpha}_{\max}'[1 : m]$ and $\boldsymbol{\alpha}_{\max}'[m + 1 : d]$ as first $m$ components and last $d - m$ components of the vector $\boldsymbol{\alpha}_{\max}'$. Since $P_{\text{rot}} = \{\mathbf{Q}^\top \mathbf{x} : \mathbf{x} \in P\}$, the corners of $P_{\text{rot}}$ are given by:

$$\mathcal{C}_{\text{rot}} = \left\{ \mathbf{A}^\top(\boldsymbol{\alpha}_{\max}'[1 : m] \odot \mathbf{s}_1) + \mathbf{A}_\perp^\top(\boldsymbol{\alpha}_{\max}'[m + 1 : d] \odot \mathbf{s}_2) : \mathbf{s}_1 \in \{-1/\sqrt{d}, 1/\sqrt{d}\}^m, \mathbf{s}_2 \in \{-1/\sqrt{d}, 1/\sqrt{d}\}^{d-m} \right\}.$$

The volume of the projection onto the subspace orthogonal to $\mathbf{A}$ is:

$$\text{Vol}(\text{Proj}_{\mathbf{A}_\perp}(P_{\text{rot}})) = \prod_{i=m+1}^d 2\boldsymbol{\alpha}_{\max,i}'.$$

Let $\mathbf{y} \in P_{\text{rot}}$. By definition, $\mathbf{y} = \mathbf{Q}^\top \mathbf{x}$ for some $\mathbf{x} \in P$. Then:

$$\|\mathbf{y}\|_\infty = \|\mathbf{Q}^\top \mathbf{x}\|_\infty.$$

Using the properties of the norm and orthonormality of $\mathbf{Q}$:

$$\|\mathbf{y}\|_\infty \leq \left\|\mathbf{Q}^\top\right\|_\infty \|\mathbf{x}\|_\infty \leq \sqrt{d} \left\|\mathbf{Q}^\top\right\|_2 \|\mathbf{x}\|_\infty \leq 1$$

Since $\mathbf{Q}$ is orthonormal, $\|\mathbf{Q}^\top\|_\infty = 1$. Additionally, $\|\mathbf{x}\|_\infty \leq 1/\sqrt{d}$ because $\mathbf{x} \in [-1/\sqrt{d}, 1/\sqrt{d}]^d$. Thus:

$$\|\mathbf{y}\|_\infty \leq 1,$$

showing that $\mathbf{y} \in [-1,1]^d$. Hence, $P_{\text{rot}} \subseteq [-1,1]^d$.

Since $P_{\text{rot}} \subseteq [-1,1]^d$, the volume of the projection satisfies:

$$\text{Vol}(\text{Proj}_{\mathbf{A}_\perp}(P_{\text{rot}})) \leq \text{Vol}(\text{Proj}_{\mathbf{A}_\perp}([-1,1]^d)).$$

Thus:

$$\text{Vol}(\text{Proj}_{\mathbf{A}_\perp}([-1,1]^d)) \geq \prod_{i=m+1}^{d} 2\alpha'_{\max,i}.$$

$\square$

### 11.10. Proof of Lemma 4.3

*Proof.* Recollect the definition of the partitioning scheme $\mathcal{T}$ from Definition 2.3. $\mathcal{T}$ represents the equivalent partitioning scheme of $\mathcal{A}$. For the partitioning scheme $\mathcal{T}$, at a specific depth $h$, each cell can be indexed by $i$, i.e, $\mathcal{T}_{h,i}$   $1 \leq i \leq 3^h$, where, $3^h$ represents the total number of cells at depth $h$, and $i$ is the index of a specific cell within that depth. Similarly, For the partitioning scheme $\mathcal{P}$, at a depth $h$, each cell can be indexed by $j$, i.e, $\mathcal{P}_{h,j}$   $1 \leq j \leq 3^h$.

**Definition of the $POpt$ Relation**   Consider the following definition of the $POpt$ relation:

$$POpt = \{(\mathcal{T}_{h,i}, \mathcal{P}_{h,j}) : \mathcal{T}_{h,i} \in \mathcal{N}_\mathcal{T}(\epsilon),\ \mathcal{P}_{h,j} \in \mathcal{N}_\mathcal{P}(\epsilon), \mathcal{T}_{h,i} \cap \mathbf{A}\mathcal{P}_{h,j} \neq \emptyset\}$$

The sets $\mathcal{N}_\mathcal{T}(\epsilon)$ and $\mathcal{N}_\mathcal{P}(\epsilon)$ denote near-optimal cells in the respective partitioning schemes.

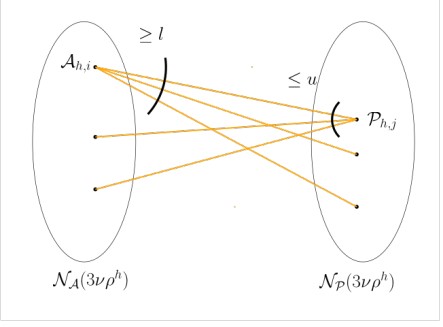

*Figure 6.* Caption

For any $h, i$, consider the cell $\mathcal{T}_{h,i} \in \mathcal{N}_\mathcal{T}(\epsilon)$. Let $l$ be the lower bound on the number of elements in $POpt$ that are of the form $(\mathcal{T}_{h,i}, \cdot)$. Then, we have: $|POpt| \geq |\mathcal{N}_\mathcal{T}(\epsilon)| \cdot l$. Similarly, for any $h, j$, consider the cell $\mathcal{P}_{h,j} \in \mathcal{N}_\mathcal{P}(3\nu\rho^h)$. Let $u$ be the upper bound on the number of elements in $POpt$ that are of the form $(\cdot, \mathcal{P}_{h,j})$. Then, we have: $|\mathcal{N}_\mathcal{P}(\epsilon)| \cdot u \geq |POpt|$. Combining these two inequalities, we get:

$$|\mathcal{N}_\mathcal{T}(\epsilon)| \leq |\mathcal{N}_\mathcal{P}(\epsilon)| \cdot \frac{u}{l} \tag{31}$$

This inequality provides a relationship between the near-optimal cells in the two partitioning schemes, taking into account the bounds on the number of elements in the $POpt$ relation.

### 11.10.1. ESTIMATING UPPER BOUND $u$

If $\mathcal{P}_{h,j}$ is a near-optimal cell, then there is a $\mathbf{x} \in \mathcal{P}_{h,j}$ such that $f(\mathbf{x}) \geq f^* - \epsilon$. For a $\mathbf{y} = \mathbf{A}\mathbf{x}$ we get $g(\mathbf{y}) = f(\mathbf{A}^T\mathbf{y}) = f(\mathbf{x})$. If $\mathbf{y} \in \mathcal{T}_{h,i}$, then $\mathcal{T}_{h,i}$ is a near optimal cell and $y \in \mathcal{T}_{h,i} \cap \mathbf{A}\mathcal{P}_{h,j}$. Thus the pair $(\mathcal{T}_{h,i}, \mathcal{P}_{h,j}) \in POpt$. To obtain the upper bound $u$ on the number of such pairs, suppose that all $\mathbf{x} \in \mathcal{P}_{h,j}$ satisfy $f(\mathbf{x}) \geq f^* - \epsilon$. Then the entire region $\mathbf{A}\mathcal{P}_{h,j}$ is near-optimal, and we calculate the maximum number of distinct $\mathcal{T}_{h,i}$ that can be present in $POpt$.

We use Lemma 4.2 to obtain an upper bound to $u$. To apply the lemma, we first approximate the domain $\mathbf{A}\mathcal{P}_{h,j}$ with a hypercube, which we use as $\kappa$ and we obtain a lower bound on the side-lengths of $\mathcal{T}_{h,i}$ cells which will be used as a $\kappa'$. These approximations allows us to effectively utilize Lemma 4.2 in our analysis.

Using Proposition 9.1, the cell $\mathcal{T}_{h,i}$ and $\mathcal{P}_{h,j}$ can be written as

$$\mathcal{T}_{h,i} = \left\{ \boldsymbol{\alpha} \in \mathbb{R}^m, \left\| \frac{\boldsymbol{\alpha} - \boldsymbol{\alpha}_{h,i}}{\mathbf{s}} \right\|_\infty \leq \alpha \quad \text{with} \quad \mathbf{s} = [3^{-\lfloor \frac{h+m-j}{m} \rfloor}]_{j=1}^m \right\}$$

$$\mathcal{P}_{h,i} = \left\{ \mathbf{x} \in \mathbb{R}^d, \left\| \frac{\mathbf{x} - \mathbf{x}_{h,i}}{\mathbf{c}} \right\|_\infty \leq \kappa \quad \text{with} \quad \mathbf{c} = [3^{-\lfloor \frac{h+d-j}{d} \rfloor}]_{j=1}^d \right\}$$

For all $\mathbf{x}_1, \mathbf{x}_2 \in \mathcal{P}_{h,i}$, consider $\|\mathbf{A}\mathbf{x}_1 - \mathbf{A}\mathbf{x}_2\|_\infty$, we have:

$$\begin{aligned}
&= \|\mathbf{A}\mathbf{x}_1 - \mathbf{A}\mathbf{x}_2\|_\infty \\
&\leq \|\mathbf{A}\|_\infty \|\mathbf{x}_1 - \mathbf{x}_2\|_\infty && \text{(from the Matrix operator norm definition)} \\
&\leq \sqrt{m} \|\mathbf{x}_1 - \mathbf{x}_2\|_\infty && \text{(From Matrix Inequality 13 and } \|\mathbf{A}\|_2 = 1) \\
&= \sqrt{m} \|\mathbf{x}_1 - \mathbf{x}_2 + \mathbf{x}_{h,i} - \mathbf{x}_{h,i}\|_\infty \\
&\leq \sqrt{m} \|\mathbf{x}_1 - \mathbf{x}_{h,i}\|_\infty + \sqrt{m} \|\mathbf{x}_1 - \mathbf{x}_{h,i}\|_\infty \\
&\leq 2\sqrt{m}3^{-k} && \text{(Let } h = kd + i)
\end{aligned}$$

Now, we estimate the side lengths for the $\mathcal{T}_{h,i}$ cell, by computing $\mathbf{s}$

$$\frac{h+m-j}{m} = \frac{kd+m+i-j}{m} \leq \frac{kd+m+d}{m} \leq (k+1)\frac{d}{m} + 1$$

Hence $3^{-\lfloor \frac{h+m-j}{m} \rfloor} \geq 3^{-(1+(k+1)\frac{d}{m})}$.

We set $\kappa = 2\sqrt{m}3^{-k}, \kappa' = \alpha 3^{-(1+(k+1)\frac{d}{m})}$ and invoke Lemma 4.2, to derive the upper bound to $u$. Thus,

$$\begin{aligned}
u &\leq \left( 2 + \frac{2\sqrt{m}3^{-k}}{\alpha 3^{-1}3^{-(k+1)\frac{d}{m}}} \right)^m = \left( 2 + \frac{6\sqrt{m}3^{d/m}3^{k(\frac{d-m}{m})}}{\alpha} \right)^m \\
&\leq \left( 2 + 6\sqrt{m}3^{d/m}3^{k(\frac{d-m}{m})} \right)^m \leq (12\sqrt{m})^m 3^d 3^{k(d-m)}
\end{aligned}$$

### 11.10.2. ESTIMATING LOWER BOUND $l$

Recall the definition of $\mathbf{A}_\perp$ from Definition 9.5. If $\mathcal{A}_{h,i}$ is a near-optimal cell, then there is a $x \in \mathcal{A}_{h,i}$ s.t. $f(x) \geq f^* - 3\nu\rho^h$. Consider $\mathbf{x}' = \mathbf{x} + \mathbf{A}_\perp\mathbf{y}$, where $\mathbf{y} \in \mathbb{R}^{d-m}$. Then, $f(\mathbf{x}') = g(\mathbf{A}\mathbf{x}') = g(\mathbf{A}\mathbf{x}) = f(\mathbf{x})$. Hence the entire domain $\{\mathbf{x} + \mathbf{A}_\perp\mathbf{y} \mid \mathbf{y} \in \mathbb{R}^{d-m}, \ \mathbf{x} + \mathbf{A}_\perp\mathbf{y} \in [-1,1]^d\}$ is a near optimal region. Suppose $\mathcal{P}_{h,j}$ has a not zero intersection with the above domain, then the cell $\mathcal{P}_{h,j}$ is a near-optimal cell. We thus count all these cells to get a lower bound $l$.

$$n_2 \geq \frac{\text{Vol}(\text{Proj}_{\mathbf{A}_\perp}(\mathcal{P}_{0,0}))}{\text{Vol}(\text{Proj}_{\mathbf{A}_\perp}(\mathcal{P}_{h,0}))}$$

$$\geq \frac{\prod_{i=m+1}^d 2(\boldsymbol{\alpha}'_{\max})_i}{\prod_{i=m+1}^d 2\max_{1\leq j\leq 2^d} \mathbf{q}_i^\top \mathbf{x}_j} \qquad \text{(Numerator is through Lemma 11.1)}$$

$$\geq \frac{\prod_{i=m+1}^d (1/\sqrt{d})2\max_{1\leq j\leq 2^d} \mathbf{q}_i^\top \mathbf{c}_j}{\prod_{i=m+1}^d 2\max_{1\leq j\leq 2^d} \mathbf{q}_i^\top \mathbf{x}_j} \qquad \text{(Here } \mathbf{c}_j \text{ are corners of hypercube } [-1,1]^d)$$

$$\geq \frac{(1/\sqrt{d})^{d-m}\prod_{i=m+1}^d \max_{1\leq j\leq 2^d} \mathbf{q}_i^\top \mathbf{c}_j}{\prod_{i=m+1}^d 3^{-k}\max_{1\leq j\leq 2^d} \mathbf{q}_i^\top \mathbf{c}_j}$$

$$\geq \frac{(1/\sqrt{d})^{d-m}}{3^{-k(d-m)}} \geq (1/d)^{d-m}3^{k(d-m)}.$$

Hence, the lower bound $l = (1/d)^{d-m}3^{k(d-m)}$ Thus, using Equation 31, we conclude that

$$|\mathcal{N}_\mathcal{A}(\epsilon)| \leq |\mathcal{N}_\mathcal{T}(3\nu\rho^h)| \cdot 3^d d^{d-m}(12\sqrt{m})^m \tag{32}$$

$\square$

### 11.11. Proof of Lemma 4.4

*Proof.* To start, we recall the relationship between the parameters of the two partitioning schemes as established in Lemma 9.4. Specifically, we have:

$$\rho_\mathcal{A} = \rho^\beta, \nu_\mathcal{A} = \max\{\nu, f^* - \inf_{\mathbf{x}\in\kappa_1\mathbb{H}_1^d} f(\mathbf{x})\}\rho^{(1-\beta)(m-1)-\tilde{h}_1}$$

For brevity, we denote $\nu' = \max\{\nu, f^* - \inf_{\mathbf{x}\in\kappa_1\mathbb{H}_1^d} f(\mathbf{x})\}$.

From the definiton of near-optimality dimension for the $\mathcal{P}$ partioning scheme, we have:

$$\mathcal{N}_\mathcal{P}(3\nu\rho^h) \leq C\rho^{-\eta_\mathcal{P} h} \tag{33}$$

Now, consider $3\nu_\mathcal{A}\rho_\mathcal{A}^h$:

$$= 3\nu'\rho^{\beta h+(1-\beta)(m-1)-\tilde{h}_1}$$

$$= 3\nu\rho^{\beta h+\log_\rho(\nu'\rho^{(1-\beta)(m-1)-\tilde{h}_1})-\log_\rho(\nu)}$$

$$\leq 3\nu\rho^{h+\log_\rho(\nu'\rho^{(1-\beta)(m-1)-\tilde{h}_1})-\log_\rho(\nu)}$$

$$\leq 3\nu\rho^{h+\left\lfloor\log_\rho(\nu'\rho^{(1-\beta)(m-1)-\tilde{h}_1})-\log_\rho(\nu)\right\rfloor}$$

$$= 3\nu\rho^{h-\tilde{h}_3} \qquad \text{(Denote } \tilde{h}_3 = -\left\lfloor\log_\rho(\nu'\rho^{(1-\beta)(m-1)-\tilde{h}_1})-\log_\rho(\nu)\right\rfloor)$$

Now, consider

$$\mathcal{N}_\mathcal{A}(3\nu_\mathcal{A}/\rho_\mathcal{A}^{-h}) \leq \mathcal{N}_\mathcal{A}(3\nu\rho^{h-\tilde{h}_3}) \qquad (3\nu_\mathcal{A}\rho_\mathcal{A}^h \leq 3\nu\rho^{h-\tilde{h}_3})$$

$$\leq 3^d d^{d-m}(12\sqrt{m})^m \mathcal{N}_\mathcal{P}(3\nu\rho^{h-\tilde{h}_3}) \qquad \text{(Using Lemma 4.3)}$$

$$\leq 3^d d^{d-m}(12\sqrt{m})^m C\rho^{-\eta_\mathcal{P}(h-\tilde{h}_3)} \qquad \text{(Using Inequality 33)}$$

$$= 3^d d^{d-m}(12\sqrt{m})^m C\rho^{\eta_\mathcal{P}\tilde{h}_3}\rho^{-\eta_\mathcal{P} h}$$

Therefore, we have:

$$\mathcal{N}_\mathcal{A}(3\nu_\mathcal{A}/\rho_\mathcal{A}^{-h}) \leq 3^d d^{d-m}(12\sqrt{m})^m C\rho^{\eta_\mathcal{P}\tilde{h}_3}\rho^{-\eta_\mathcal{P} h} \quad \forall h \geq \tilde{h}_3$$

For heights $h \in [0 : \tilde{h}_3 - 1]$, since the right-hand side quantity is monotonically increasing, we can use the value of the right-hand side at depth $h = \tilde{h}_3$ in the above expression, which is $3^d d^{d-m} (12\sqrt{m})^m C$.

Therefore, we have

$$\mathcal{N}_{\mathcal{A}}(3\nu_{\mathcal{A}}/\rho_{\mathcal{A}}^{-h}) \le 3^d d^{d-m} (12\sqrt{m})^m C \rho^{-\eta_{\mathcal{P}} \tilde{h}_3} \rho^{-\eta_{\mathcal{P}} h} = 3^d d^{d-m} (12\sqrt{m})^m C \rho^{-\eta_{\mathcal{P}} \tilde{h}_3} \rho_{\mathcal{A}}^{-\frac{\eta_{\mathcal{P}}}{\beta} h}$$

This implies that $\eta_{\mathcal{A}} \le \eta_{\mathcal{P}} / \beta$, and the constant $C_{\mathcal{A}}$ is given by $3^d d^{d-m} (12\sqrt{m})^m C \rho^{-\eta_{\mathcal{P}} \tilde{h}_3}$. $\qquad\square$

### 11.12. Proof of Lemma 4.5

*Proof.* Consider $\mathbf{z}^* = (\mathbf{A}\hat{\mathbf{A}}^\top)^{-1}\mathbf{A}\mathbf{x}^*$, then

$$\begin{aligned} f(\hat{\mathbf{A}}^\top \mathbf{z}^*) = g(\mathbf{A}\hat{\mathbf{A}}^\top \mathbf{z}^*) &= g(\mathbf{A}\hat{\mathbf{A}}^\top (\mathbf{A}\hat{\mathbf{A}}^\top)^{-1}\mathbf{A}\mathbf{x}^*) \\ &= g(\mathbf{A}\mathbf{x}^*) = f(\mathbf{x}^*). \end{aligned}$$

Now, we show that

$$\begin{aligned} \|\mathbf{z}^*\|_\infty = \left\|(\mathbf{A}\hat{\mathbf{A}}^\top)^{-1}\mathbf{A}\mathbf{x}^*\right\|_\infty & \\ \le \left\|(\mathbf{A}\hat{\mathbf{A}}^\top)^{-1}\mathbf{A}\mathbf{x}^*\right\|_2 & \text{(Vector Norm Inequality)} \\ \le \left\|(\mathbf{A}\hat{\mathbf{A}}^\top)^{-1}\right\|_2 \|\mathbf{A}\mathbf{x}^*\|_2 & \text{(Operator Norm Definition)} \\ = \frac{\left\|(\mathbf{A}\hat{\mathbf{A}}^\top)^{-1}\right\|_2 \sigma_{\min}(\mathbf{A}\hat{\mathbf{A}}^\top)}{\sigma_{\min}(\mathbf{A}\hat{\mathbf{A}}^\top)} \|\mathbf{A}\mathbf{x}^*\|_2 & \text{(Since, } \sigma_{\min}(\mathbf{A}\hat{\mathbf{A}}^\top) > 0) \\ \le \frac{\left\|(\mathbf{A}\hat{\mathbf{A}}^\top)^{-1}\mathbf{A}\hat{\mathbf{A}}^\top\right\|_2}{\sigma_{\min}(\mathbf{A}\hat{\mathbf{A}}^\top)} \|\mathbf{A}\mathbf{x}^*\|_2 & \text{(From Matrix Inequality 14)} \\ \le \frac{\|\mathbf{A}\mathbf{x}^*\|_2}{1 - \text{dist}^2(\mathbf{A}, \hat{\mathbf{A}})} & \text{(From the Equation 11 and } \|\mathbf{I}\|_2 = 1) \\ \le \frac{\sqrt{m} \|\mathbf{A}\mathbf{x}^*\|_\infty}{1 - \text{dist}^2(\mathbf{A}, \hat{\mathbf{A}})} & \text{(Vector Norm Inequality)} \\ \le \frac{\sqrt{m}\alpha}{1 - \text{dist}^2(\mathbf{A}, \hat{\mathbf{A}})}. & \end{aligned}$$

$\qquad\square$

For some of the proofs, we use the following equivalence of partitioning schemes.

### 11.13. Equivalence of partitioning schemes

Consider the partitioning schemes $\mathcal{A}$ and $\mathcal{T}$ which are defined in Definition 2.3. For every $\boldsymbol{\alpha} \in \mathcal{T}_{h,i}$, there exists a unique $\mathbf{x} \in \mathcal{A}_{h,i}$ such that $\mathbf{x} = \mathbf{A}^\top \boldsymbol{\alpha}$. This establishes an equivalence:

$$\forall \mathbf{x} \in \mathcal{A}_{h,i}, \quad f(\mathbf{x}) = f(\mathbf{A}^\top \boldsymbol{\alpha}) = g(\mathbf{A}\mathbf{A}^\top \boldsymbol{\alpha}) = g(\boldsymbol{\alpha}) \tag{34}$$

where $f$ is optimized on $\mathcal{A}_{h,i}$ and $g$ on $\mathcal{T}_{h,i}$. Thus, optimizing $f$ over $\mathcal{A}$ is equivalent to optimizing $g$ over $\mathcal{T}$.

Similarly, for the partitioning schemes $\hat{\mathcal{A}}$ and $\hat{\mathcal{T}}$, we have: $\forall \boldsymbol{\alpha} \in \hat{\mathcal{T}}_{h,i}, \exists \mathbf{x} \in \hat{\mathcal{A}}_{h,i}$ such that $\mathbf{x} = \hat{\mathbf{A}}^\top \boldsymbol{\alpha}$. This leads to:

$$\forall \mathbf{x} \in \hat{\mathcal{A}}_{h,i}, \quad f(\mathbf{x}) = f(\hat{\mathbf{A}}^\top \boldsymbol{\alpha}) = g(\mathbf{A}\hat{\mathbf{A}}^\top \boldsymbol{\alpha}) \overset{\text{def}}{=} \hat{g}(\boldsymbol{\alpha}) \tag{35}$$

where $g$ is defined on $\mathcal{T}_{h,i}$ and $\hat{g}$ on $\hat{\mathcal{T}}_{h,i}$. Therefore, optimizing $f$ over $\hat{\mathcal{A}}$ is equivalent to optimizing $\hat{g}$ over $\hat{\mathcal{T}}$.

### 11.14. Proof of Lemma 9.8

*Proof.* From the notation section 7, we have $\mathbf{W}_\perp = \mathbf{W} - \mathbf{W}\mathbf{A}^\top\mathbf{A}$. Next, consider the singular value decomposition (SVD) of $\mathbf{W}$, given by $\mathbf{W} = \mathbf{U}\mathbf{S}\mathbf{V}^\top$, where $\mathbf{U} \in \mathbb{R}^{p \times p}$ and $\mathbf{V} \in \mathbb{R}^{d \times d}$ are orthogonal matrices, satisfying $\mathbf{U}\mathbf{U}^\top = \mathbf{U}^\top\mathbf{U} = \mathbf{I}_p$, $\mathbf{V}\mathbf{V}^\top = \mathbf{V}^\top\mathbf{V} = \mathbf{I}_d$ and $\mathbf{S} \in \mathbb{R}^{p \times d}$ is a diagonal matrix.

We have this relation $\|\mathbf{W}\|_2 = \sigma_1(\mathbf{W}) \geq \cdots \geq \sigma_r(\mathbf{W}) \geq 0, \quad r = \min\{p, d\}$. From the Lemma assumption, we have $\text{rank}(\mathbf{W}) = r \geq m$. Thus, we collect the respective leading $m$ columns of $\mathbf{U}$ and $\mathbf{V}$, denoted as $\mathbf{U}_1 \in \mathbb{R}^{p \times m}$ and $\mathbf{V}_1 \in \mathbb{R}^{d \times m}$, respectively. The remaining columns are denote by $\mathbf{U}_2 \in \mathbb{R}^{p \times p - m}$ and $\mathbf{V}_2 \in \mathbb{R}^{d \times d - m}$. Since, columns of $\mathbf{U}_1$ and $\mathbf{U}_2$ forms a orthonormal basis for $\mathbb{R}^p$, we have:

$$\mathbf{U}_1^\top\mathbf{U}_1 = \mathbf{I}_m \quad \text{and} \quad \mathbf{U}_1^\top\mathbf{U}_2 = \mathbf{0}_{m \times p - m}. \tag{36}$$

Then the SVD of $\mathbf{W}$ can be written as: $\mathbf{W} = \mathbf{U}_1\mathbf{S}_1\mathbf{V}_1^\top + \mathbf{U}_2\mathbf{S}_2\mathbf{V}_2^\top$. and from the lemma hypothesis $\hat{\mathbf{A}} = \mathbf{V}_1^\top$. Now, consider the implications of this decomposition for the proof. Consider,

$$
\begin{aligned}
&= \left\|\mathbf{W}(\mathbf{I} - \mathbf{A}^\top\mathbf{A})\right\|_F \\
&\geq \left\|\mathbf{W}(\mathbf{I} - \mathbf{A}^\top\mathbf{A})\right\|_2 && \text{(From Matrix Norm Inequality 13)} \\
&= \left\|\mathbf{U}_1\mathbf{S}_1\mathbf{V}_1^\top(\mathbf{I} - \mathbf{A}^\top\mathbf{A}) + \mathbf{U}_2\mathbf{S}_2\mathbf{V}_2^\top(\mathbf{I} - \mathbf{A}^\top\mathbf{A})\right\|_2 && \text{(SVD of } \mathbf{W}) \\
&= \left\|\mathbf{U}_1^\top\right\|_2 \left\|\mathbf{U}_1\mathbf{S}_1\mathbf{V}_1^\top(\mathbf{I} - \mathbf{A}^\top\mathbf{A}) + \mathbf{U}_2\mathbf{S}_2\mathbf{V}_2^\top(\mathbf{I} - \mathbf{A}^\top\mathbf{A})\right\|_2 && (\left\|\mathbf{U}_1^\top\right\|_2 = 1) \\
&\geq \left\|\mathbf{U}_1^\top(\mathbf{U}_1\mathbf{S}_1\mathbf{V}_1^\top(\mathbf{I} - \mathbf{A}^\top\mathbf{A}) + \mathbf{U}_2\mathbf{S}_2\mathbf{V}_2^\top(\mathbf{I} - \mathbf{A}^\top\mathbf{A}))\right\|_2 && \text{(From Matrix Norm Inequality 14)} \\
&= \left\|\mathbf{S}_1\mathbf{V}_1^\top(\mathbf{I} - \mathbf{A}^\top\mathbf{A})\right\|_2 && \text{(Using Equation 36)} \\
&\geq \sigma_{min}(\mathbf{S}_1)\left\|\mathbf{V}_1^\top(\mathbf{I} - \mathbf{A}^\top\mathbf{A})\right\|_2 && \text{(From Matrix Norm Inequality 14)} \\
&= \sigma_m \left\|\hat{\mathbf{A}}(\mathbf{I} - \mathbf{A}^\top\mathbf{A})\right\|_2 \\
&= \sigma_m \left\|\hat{\mathbf{A}}\mathbf{A}_\perp^\top\mathbf{A}_\perp\right\|_2 && ([\mathbf{A}^\top, \mathbf{A}_\perp^\top] \text{ is a orthonormal matrix, thus } \mathbf{A}^\top\mathbf{A} + \mathbf{A}_\perp^\top\mathbf{A}_\perp = \mathbf{I}_d) \\
&= \sigma_m \left\|\hat{\mathbf{A}}\mathbf{A}_\perp^\top\mathbf{A}_\perp\right\|_2 \left\|\mathbf{A}_\perp^\top\right\|_2 && (\left\|\mathbf{A}_\perp^\top\right\|_2 = 1) \\
&\geq \sigma_m \left\|\hat{\mathbf{A}}\mathbf{A}_\perp^\top\mathbf{A}_\perp\mathbf{A}_\perp^\top\right\|_2 && \text{(From Matrix Norm Inequality 14)} \\
&= \sigma_m \left\|\hat{\mathbf{A}}\mathbf{A}_\perp^\top\right\|_2 && (\mathbf{A}_\perp \text{ consists of orthonormal rows)} \\
&= \sigma_m \left\|\sin\Theta(\mathbf{A}, \hat{\mathbf{A}})\right\|_2 && \text{(From Definition 9.5)}
\end{aligned}
$$

Since $\sigma_m > 0$, we have

$$\left\|\sin\Theta(\mathbf{A}, \hat{\mathbf{A}})\right\|_2 \leq \frac{\left\|\mathbf{W}(\mathbf{I} - \mathbf{A}^\top\mathbf{A})\right\|_2}{\sigma_m} \leq \frac{\left\|\mathbf{W}(\mathbf{I} - \mathbf{A}^\top\mathbf{A})\right\|_F}{\sigma_m}.$$

$\square$

### 11.15. Proof of Lemma 9.6

*Proof.* Using the partitioning scheme equivalence from Section 11.13, we can work with partitioning schemes $\mathcal{T}$ and $\hat{\mathcal{T}}$ in place of the $\mathcal{A}$ and $\hat{\mathcal{A}}$ partitioning schemes.

Let $\mathbf{z}_h, \hat{\mathbf{z}}_h$ be the representatives of $\mathcal{T}_h^*$ and $\hat{\mathcal{T}}_h^*$ cells respectively. Then

$$\mathcal{T}_h^* = \left\{ \mathbf{z} \in \mathbb{R}^m, \left\| \frac{\mathbf{z} - \mathbf{z}_h}{\mathbf{s}} \right\|_\infty \leq \alpha \quad \text{with} \quad \mathbf{s} = [3^{-\lfloor \frac{h+m-i}{m} \rfloor}]_{i=1}^m \right\}$$

and

$$\hat{\mathcal{T}}_h^* = \left\{ \mathbf{z} \in \mathbb{R}^m, \left\| \frac{\mathbf{z} - \hat{\mathbf{z}}_h}{\mathbf{s}} \right\|_\infty \leq \frac{\sqrt{m}\alpha}{\sqrt{1 - \text{dist}^2(\mathbf{A}, \hat{\mathbf{A}})}} \quad \text{with} \quad \mathbf{s} = [3^{-\lfloor \frac{h+m-i}{m} \rfloor}]_{i=1}^m \right\}$$

For brevity, denote $\kappa_2 = \frac{\sqrt{m}}{\sqrt{1 - \text{dist}^2(\mathbf{A}, \hat{\mathbf{A}})}}$. Let $\mathbf{z}^*$ denote a maximizer of the function $g$, i.e., $\mathbf{z}^* \in \arg\max_{\mathbf{z} \in \alpha \mathbb{H}_1^m} g(\mathbf{z})$. Given that $\hat{g}(\mathbf{z}) = g(\mathbf{A}\hat{\mathbf{A}}^\top \mathbf{z})$ and that $\mathbf{A}\hat{\mathbf{A}}^\top$ an invertible matrix, we can identify a corresponding maximizer $\mathbf{z}_{\hat{g}}^*$ for $\hat{g}$ such that: $\mathbf{z}_g^* = \mathbf{A}\hat{\mathbf{A}}^\top \mathbf{z}_{\hat{g}}^*$. Under this transformation, $\hat{g}(\mathbf{z}_{\hat{g}}^*)$ will be a maximizer of the function $\hat{g}$.

From lemma assumption, $\mathcal{T}$ satisfies Assumption 2.6, therefore we have:

$$\forall h \in \mathbb{N}_0, \sup_{\mathbf{z} \in \mathcal{T}_h^*} (g^* - g(\mathbf{z})) \leq \nu_{\mathcal{A}} \rho_{\mathcal{A}}^h \tag{37}$$

We aim to control

$$\forall h \in \mathbb{N}_0, \sup_{\mathbf{z} \in \hat{\mathcal{T}}_h^*} (g^* - \hat{g}(\mathbf{z})) = \sup_{\mathbf{z} \in \hat{\mathcal{T}}_h^*} (g^* - g(\mathbf{A}\hat{\mathbf{A}}^\top \mathbf{z})) \tag{38}$$

We choose some height $h$ and represent it as $h = km + i$. Let us denote:

$$R_h = \{ \mathbf{A}\hat{\mathbf{A}}^\top \mathbf{z} : \mathbf{z} \in \hat{\mathcal{T}}_h^* \}.$$

Now, we show that:

$$R_{km} \subseteq \mathcal{T}_{km-mk'}^*,$$

where $k' = \left\lceil \log_3 \frac{2\sqrt{m}\kappa_2}{\kappa_2 - 1} \right\rceil$.

Consider a point $\mathbf{A}\hat{\mathbf{A}}^\top \mathbf{z}$ from the set $R_{km}$, where $\mathbf{z} \in \hat{\mathcal{T}}_{km}^*$. By definition, $\mathbf{z}$ satisfies the following condition:

$$\|\mathbf{z} - \hat{\mathbf{z}}_{km}\|_\infty \leq 3^{-k}\kappa_2\alpha. \tag{39}$$

Now, we show that $\mathbf{A}\hat{\mathbf{A}}^\top \mathbf{z} \in \mathcal{T}_{km-mk'}^*$. We begin with the following inequality:

$$= \left\| \mathbf{z}_{(k-k')m} - \mathbf{A}\hat{\mathbf{A}}^\top \mathbf{z} \right\|_\infty$$

$$= \left\| \mathbf{z}_{(k-k')m} - \mathbf{z}_g^* + \mathbf{A}\hat{\mathbf{A}}^\top \mathbf{z}_{\hat{g}}^* - \mathbf{A}\hat{\mathbf{A}}^\top \mathbf{z} \right\|_\infty \qquad \text{(Using } \mathbf{z}_g^* = \mathbf{A}\hat{\mathbf{A}}^\top \mathbf{z}_{\hat{g}}^*\text{)}$$

$$\leq \left\| \mathbf{z}_{(k-k')m} - \mathbf{z}_g^* \right\|_\infty + \left\| \mathbf{A}\hat{\mathbf{A}}^\top (\mathbf{z}_{\hat{g}}^* - \mathbf{z}) \right\|_\infty \qquad \text{(Triangle Inequality of Norms)}$$

$$\leq 3^{-(k-k')}\alpha + \left\| \mathbf{A}\hat{\mathbf{A}}^\top (\mathbf{z}_{\hat{g}}^* - \mathbf{z}) \right\|_\infty \qquad (\mathbf{z}_g^* \in \mathcal{T}_{km}^* \text{ cell. Thus } \|\mathbf{z}_{km} - \mathbf{z}^*\|_\infty \leq 3^{-k}\alpha)$$

$$\leq 3^{-(k-k')}\alpha + \left\| \mathbf{A}\hat{\mathbf{A}}^\top \right\|_\infty \|\mathbf{z}_{\hat{g}}^* - \mathbf{z}\|_\infty \qquad \text{(Operator Norm Definition)}$$

$$\leq 3^{-(k-k')}\alpha + \sqrt{m} \|\mathbf{z}_{\hat{g}}^* - \mathbf{z}\|_\infty \qquad \text{(From Matrix Inequality 13, 14 and } \|\mathbf{A}\|_2 = \left\| \hat{\mathbf{A}} \right\|_2 = 1)$$

$$\leq 3^{-(k-k')}\alpha + \sqrt{m} \|-\hat{\mathbf{z}}_{km} + \mathbf{z}_{\hat{g}}^*\|_\infty + \sqrt{m} \|\hat{\mathbf{z}}_{km} - \mathbf{z}\|_\infty$$

$$\leq 3^{-(k-k')}\alpha + \sqrt{m}3^{-k}2\kappa_2\alpha \qquad \text{(using Inequality 39 and } \mathbf{z}_{\hat{g}}^* \in \hat{\mathcal{T}}_{km}^*)$$

$$\leq 3^{-(k-k')}\alpha + 3^{-(k-k')}\alpha(\kappa_2 - 1) \qquad \text{(Suppose } k' \text{ is chosen such that } 2\sqrt{m}\kappa_2 \leq (\kappa_2 - 1)3^{k'})$$

$$= \alpha\kappa_2 3^{-(k-k')}$$

From the above inequality, we can conclude:

$$R_{km} \subseteq \mathcal{T}^*_{km-m\left\lceil \log_3 \frac{2\sqrt{m}\kappa_2}{\kappa_2-1} \right\rceil}$$

Using the above set containment, for any height $h = km + i$, we have

$$R_{km+i} \subseteq R_{km} \subseteq \mathcal{T}^*_{km-m\left\lceil \log_3 \frac{2\sqrt{m}\kappa_2}{\kappa_2-1} \right\rceil} \subseteq \mathcal{T}^*_{km-m\left\lceil \log_3 \frac{2\sqrt{m}\kappa_2}{\kappa_2-1} \right\rceil + i - m}$$

First and the last set containment are valid from the round robin paritioning scheme. Substituting, $h = km + i$, we get:

$$R_h \subseteq \mathcal{T}^*_{h-m\left\lceil \log_3 \frac{2\sqrt{m}\kappa_2}{\kappa_2-1} \right\rceil - m}$$

Define: $\tilde{h}_2 = m\left\lceil \log_3 \frac{2\sqrt{m}\kappa_2}{\kappa_2-1} \right\rceil + m$. Using the Inequalities 37, 38 and the above set containment, we conclude:

$$\sup_{\mathbf{z} \in \hat{\mathcal{T}}^*_h} (g^* - \hat{g}(\mathbf{z})) \leq \nu_{\mathcal{A}} \rho_{\mathcal{A}}^{h-\tilde{h}_2} \quad \forall h \geq \tilde{h}_2 \tag{40}$$

For height $h \in [0 : \tilde{h}_2 - 1]$, we know that:

$$\inf_{\mathbf{z} \in \hat{\mathcal{T}}^*_h} \hat{g}(\mathbf{z}) \geq \inf_{\mathbf{z} \in \hat{\mathcal{T}}^*_0} \hat{g}(\mathbf{z})$$

which implies:

$$\sup_{\mathbf{z} \in \hat{\mathcal{T}}^*_h} (g^* - \hat{g}(\mathbf{z})) \leq g^* - \inf_{\mathbf{z} \in \hat{\mathcal{T}}^*_0} \hat{g}(\mathbf{z}) \leq (g^* - \inf_{\mathbf{z} \in \hat{\mathcal{T}}^*_0} \hat{g}(\mathbf{z})) \rho_{\mathcal{A}}^{-\tilde{h}_2} \rho_{\mathcal{A}}^h$$

Combining 40 and the above inequality, we conclude that $\hat{\mathcal{T}}$ satisfies Assumption 2.6 with parameters:

$$\left(\rho_{\mathcal{A}}, \max\{\nu_{\mathcal{A}}, g^* - \inf_{\mathbf{z} \in \kappa_2 \alpha \mathbb{H}^m_1} g(\mathbf{A}\hat{\mathbf{A}}\mathbf{z})\} \rho_{\mathcal{A}}^{-\tilde{h}_2}\right)$$

$\square$

### 11.16. Proof of Proposition 8.1

*Proof.* Let $\mathbf{A} = [\mathbf{w}_1, \mathbf{w}_2, \mathbf{w}_3, \ldots, \mathbf{w}_p] \in \mathbb{R}^{d \times p}$, where we assume without loss of generality that the vectors $\{\mathbf{w}_1, \mathbf{w}_2, \cdots, \mathbf{w}_p\}$ are linearly independent. If the vectors were dependent, we could consider only the independent vectors without changing the $\mathrm{Span}\{\mathbf{w}_1, \mathbf{w}_2, \ldots, \mathbf{w}_p\}$. The orthogonal projection matrix $\mathbf{P}$ onto $\mathrm{Span}\{\mathbf{w}_1, \mathbf{w}_2, \ldots, \mathbf{w}_p\}$ is given by $\mathbf{P} = \mathbf{A}(\mathbf{A}^\top \mathbf{A})^{-1} \mathbf{A}^\top$.

For all $\mathbf{x} \in \mathbb{R}^d$, consider $(\mathbf{w}_i^T \mathbf{P})\mathbf{x} = (\mathbf{P}^\top \mathbf{w}_i)^\top \mathbf{x} = (\mathbf{P}\mathbf{w}_i)^\top \mathbf{x} = \mathbf{w}_i^\top \mathbf{x}$, where the equalities hold due to the following: First, $\mathbf{P}^\top = \mathbf{P}$ because $\mathbf{P}$ is a projection matrix and thus symmetric. Second, $\mathbf{P}\mathbf{w}_i = \mathbf{w}_i$ since $\mathbf{w}_i$ is in the column span of $\mathbf{A}$, and $\mathbf{P}$ projects onto this span. Therefore, we can express $f(\mathbf{x})$ as

$$f(\mathbf{x}) = \sum_{i=1}^p v_i \sigma(\mathbf{w}_i^\top \mathbf{x} + b_i) = \sum_{i=1}^p v_i \sigma(\mathbf{w}_i^\top \mathbf{P}\mathbf{x} + b_i). \tag{41}$$

Since $(\mathbf{x} - \mathbf{x}') \perp \mathrm{Span}\{\mathbf{w}_1, \mathbf{w}_2, \ldots, \mathbf{w}_p\}, \mathbf{P}\mathbf{x} = \mathbf{P}\mathbf{x}'$ and therefore using Equation (41), $f(\mathbf{x}) = f(\mathbf{x}')$. $\square$

### 11.17. Proof of Lemma 4.6

*Proof.* Using the partitioning scheme equivalence from Section 11.13, we can work with partitioning schemes $\mathcal{T}$ and $\hat{\mathcal{T}}$ in place of the $\mathcal{A}$ and $\hat{\mathcal{A}}$ partitioning schemes.

An arbitrary cell $\mathcal{T}_{h,i}$ in $\mathcal{T}$ is defined as:

$$\mathcal{T}_{h,i} = \left\{ \mathbf{z} : \mathbf{z} \in \mathbb{R}^m, \left\| \frac{\mathbf{z} - \mathbf{z}_{h,i}}{\mathbf{s}} \right\|_\infty \leq \alpha \quad \text{with} \quad \mathbf{s} = [3^{-\lfloor \frac{h+m-i}{m} \rfloor}]_{i=1}^m \right\}, \tag{42}$$

and an arbitrary cell $\hat{\mathcal{T}}_{h,i}$ in $\hat{\mathcal{T}}$ is defined as:

$$\hat{\mathcal{T}}_{h,i} = \left\{ \mathbf{z} : \mathbf{z} \in \mathbb{R}^m, \left\| \frac{\mathbf{z} - \tilde{\mathbf{z}}_{h,i}}{\mathbf{s}} \right\|_\infty \leq \frac{\alpha\sqrt{m}}{\sqrt{1 - \mathrm{dist}^2(\mathbf{A}, \hat{\mathbf{A}})}} \quad \text{with} \quad \mathbf{s} = [3^{-\lfloor \frac{h+m-i}{m} \rfloor}]_{i=1}^m \right\}. \tag{43}$$

We optimize the function $\hat{g}(\mathbf{z}) = g(\mathbf{A}\hat{\mathbf{A}}^\top \mathbf{z})$ over $\hat{\mathcal{T}}$, while optimizing $g(\mathbf{z})$ over $\mathcal{T}$. For a near-optimal cell $\mathcal{T}_{h,i} \in \mathcal{N}_{\mathcal{T}}(3\nu_{\mathcal{T}}\rho_{\mathcal{T}}^h)$, the following holds:

$$\sup_{\mathbf{z} \in \mathcal{T}_{h,i}} g(\mathbf{z}) = \sup_{\mathbf{z} \in (\mathbf{A}\hat{\mathbf{A}}^\top)^{-1}\mathcal{T}_{h,i}} g(\mathbf{A}\hat{\mathbf{A}}^\top \mathbf{z}) \geq g^* - 3\nu_{\mathcal{T}}\rho_{\mathcal{T}}^h, \tag{44}$$

where the inequality holds because $\mathcal{T}_{h,i}$ is a near-optimal cell by assumption.

We aim to count the number of cells in $\hat{\mathcal{T}}$ that satisfy the following near-optimality cell condition:

$$\sup_{\mathbf{z} \in \hat{\mathcal{T}}_{h,i}} g(\mathbf{A}\hat{\mathbf{A}}^\top \mathbf{z}) \geq g^* - 3\nu_{\hat{\mathcal{T}}}\rho_{\hat{\mathcal{T}}}^h. \tag{45}$$

Since, $\nu_{\hat{\mathcal{T}}}\rho_{\hat{\mathcal{T}}}^h > \nu_{\mathcal{T}}\rho_{\mathcal{T}}^h$, the above condition is satisfied by

$$\sup_{\mathbf{z} \in \hat{\mathcal{T}}_{h,i}} g(\mathbf{A}\hat{\mathbf{A}}^\top \mathbf{z}) \geq g^* - 3\nu_{\mathcal{T}}\rho_{\mathcal{T}}^h. \tag{46}$$

Using relations (44) and (46), we observe that since the function $g(\mathbf{A}\hat{\mathbf{A}}^\top \mathbf{z})$ is identical in both cases, it suffices to work with the domain of the cells. Define:

$$B = \{(\mathbf{A}\hat{\mathbf{A}}^\top)^{-1}\mathbf{z} : \mathbf{z} \in \mathcal{T}_{h,i}\}.$$

To obtain an upper bound to the number of near-optimal cells in the partitioning scheme $\hat{\mathcal{T}}$, we consider every cell in $\hat{\mathcal{T}}$ ($\hat{\mathcal{T}}_{h,i}$) that intersects with $B$ as potentially near-optimal. For simplicity, we enlarge the domain $B$ into a hypercube in $\mathbb{R}^m$.

The set $B$ is the transformation of the cell $\mathcal{T}_{h,i}$ under $(\mathbf{A}\hat{\mathbf{A}}^\top)^{-1}$. Since the transformation is linear and invertible, $B$ remains a bounded region. However, to simplify analysis, we enclose $B$ in a hypercube. By ensuring that the hypercube fully contains $B$, any cell $\hat{\mathcal{T}}_{h,i}$ that intersects this hypercube is considered a candidate near-optimal cell.

$\forall \mathbf{z}_1, \mathbf{z}_2 \in \mathcal{T}_{h,i}$, consider,

$$\left\| (\mathbf{A}\hat{\mathbf{A}}^\top)^{-1}\mathbf{z}_1 - (\mathbf{A}\hat{\mathbf{A}}^\top)^{-1}\mathbf{z}_1 \right\|_\infty$$

$$\leq \left\| (\mathbf{A}\hat{\mathbf{A}}^\top)^{-1} \right\|_\infty \|\mathbf{z}_1 - \mathbf{z}_2\|_\infty \qquad \text{(From the Matrix Operator Norm definition)}$$

$$\leq \sqrt{m} \left\| (\mathbf{A}\hat{\mathbf{A}}^\top)^{-1} \right\|_2 \|\mathbf{z}_1 - \mathbf{z}_2\|_\infty \qquad \text{(From the Matrix Inequality 13)}$$

$$\leq \frac{\sqrt{m} \left\| (\mathbf{A}\hat{\mathbf{A}}^\top)^{-1} \right\|_2 \|\mathbf{z}_1 - \mathbf{z}_2\|_\infty \, \sigma_{\min}(\mathbf{A}\hat{\mathbf{A}}^\top)}{\sigma_{\min}(\mathbf{A}\hat{\mathbf{A}}^\top)} \qquad \text{(Since, } \sigma_{\min}(\mathbf{A}\hat{\mathbf{A}}) > 0)$$

$$\leq \sqrt{m} \frac{\left\| (\mathbf{A}\hat{\mathbf{A}}^\top)^{-1}\mathbf{A}\hat{\mathbf{A}}^\top \right\|_2 \|\mathbf{z}_1 - \mathbf{z}_2\|_\infty}{\sigma_{\min}(\mathbf{A}\hat{\mathbf{A}}^\top)} \qquad \text{(From the Matrix Inequality 14)}$$

$$= \frac{\sqrt{m} \|\mathbf{z}_1 - \mathbf{z}_2\|_\infty}{\sigma_{\min}(\mathbf{A}\hat{\mathbf{A}}^\top)} \qquad (\|\mathbf{I}\|_2 = 1)$$

$$\overset{(1)}{\leq} \frac{\sqrt{m}}{\sigma_{\min}(\mathbf{A}\hat{\mathbf{A}}^\top)} 2\alpha 3^{-k}$$

$$= 2\alpha 3^{-k} \frac{\sqrt{m}}{\sqrt{1 - \text{dist}^2(\mathbf{A}, \hat{\mathbf{A}})}} \qquad \text{(From the Equation 11)}$$

$(1)$ is true from the following inequality,

$$\|\mathbf{z}_1 - \mathbf{z}_2\|_\infty = \left\| \mathbf{s}\left( \frac{\mathbf{z}_1 - \mathbf{z}_{h,i} + \mathbf{z}_{h,i} - \mathbf{z}_2}{\mathbf{s}} \right) \right\|_\infty = \left\| \text{diag}(\mathbf{s})\left( \frac{\mathbf{z}_1 - \mathbf{z}_{h,i} + \mathbf{z}_{h,i} - \mathbf{z}_2}{\mathbf{s}} \right) \right\|_\infty$$

$$\leq \|\text{diag}(\mathbf{s})\|_\infty \left\| \left( \frac{\mathbf{z}_1 - \mathbf{z}_{h,i} + \mathbf{z}_{h,i} - \mathbf{z}_2}{\mathbf{s}} \right) \right\|_\infty \leq 3^{-k}2\alpha$$

Hence:

$$B \subseteq 2\alpha 3^{-k} \frac{\sqrt{m}}{\sqrt{1 - \text{dist}^2(\mathbf{A}, \hat{\mathbf{A}})}} \mathbb{H}_1^m.$$

To establish an upper bound for the near-optimal cells of $\hat{\mathcal{T}}$ partitioning scheme, we consider the hypercube $2\alpha 3^{-k} \frac{\sqrt{m}}{\sqrt{1-\text{dist}^2(\mathbf{A}, \hat{\mathbf{A}})}} \mathbb{H}_1^m$ as potentially near-optimal. We can then count the maximum number of cells in the $\hat{\mathcal{T}}$ partitioning scheme at height $h$ that can intersect with this region.

To invoke the Lemma 4.2, we have $\kappa = 2\alpha 3^{-k} \frac{\sqrt{m}}{\sqrt{1-\text{dist}^2(\mathbf{A}, \hat{\mathbf{A}})}}$ and we need the side-length of the cell $\hat{\mathcal{T}}_{h,i}$ cell or for simplicity, lower-bound to side-length will be $\kappa'$. And the side-length of $\hat{\mathcal{T}}_{h,i}$ cell is greater than $2 \cdot 3^{-(k+1)} \frac{\alpha\sqrt{m}}{\sqrt{1-\text{dist}^2(\mathbf{A}, \hat{\mathbf{A}})}}$. Hence, the maximum number of cells of $\hat{\mathcal{T}}_{h,i}$ that can be tiled inside the hypercube with side length $= 2\alpha 3^{-k} \frac{\sqrt{m}}{\sqrt{1-\text{dist}^2(\mathbf{A}, \hat{\mathbf{A}})}}$ are

$$\left( 1 + \left\lceil \frac{2\alpha 3^{-k} \frac{\sqrt{m}}{\sqrt{1-\text{dist}^2(\mathbf{A}, \hat{\mathbf{A}})}}}{2 \cdot 3^{-(k+1)} \frac{\alpha\sqrt{m}}{\sqrt{1-\text{dist}^2(\mathbf{A}, \hat{\mathbf{A}})}}} \right\rceil \right)^m = 4^m$$

Hence, upper-bound to the near-optimal calls of $\hat{\mathcal{T}}$ partitioning scheme is

$$\forall h \geq 0, \quad \mathcal{N}_{\hat{\mathcal{T}}}(3\nu_{\hat{\mathcal{T}}} \rho_{\hat{\mathcal{T}}}^h) \leq 4^m \mathcal{N}_{\mathcal{T}}(3\nu_{\mathcal{T}} \rho_{\mathcal{T}}^h) \tag{47}$$

$\square$

### 11.18. Proof of Lemma 4.7

*Proof.* From the Lemma 4.6, we have

$$\forall h \geq 0, \quad \mathcal{N}_{\hat{\mathcal{T}}}(3\nu_{\hat{\mathcal{T}}}\rho_{\hat{\mathcal{T}}}^h) \leq 4^m \mathcal{N}_{\mathcal{T}}(3\nu_{\mathcal{T}}\rho_{\mathcal{T}}^h) \tag{48}$$

Using the above relation, we relate the near-optimality dimension. Suppose, $\eta_{\mathcal{T}}(\nu_{\mathcal{T}}, \rho_{\mathcal{T}}, C_{\mathcal{T}})$ is the near optimality dimension of $\mathcal{T}$ then,

$$\forall h \geq 0, \quad \mathcal{N}_{\mathcal{T}}(3\nu_{\mathcal{T}}\rho_{\mathcal{T}}^h) \leq C_{\mathcal{T}}\rho_{\mathcal{T}}^{-\eta_{\mathcal{T}}h} \tag{49}$$

From Lemma 9.6, we have the following SequOOL parameter relations:

$$\nu_{\hat{\mathcal{T}}} = \max\{\nu_{\mathcal{T}}, g^* - \inf_{\mathbf{z} \in \kappa_2 \alpha \mathbb{H}_1^m} g(\mathbf{A}\hat{\mathbf{A}}^\top \mathbf{z})\}/\rho_{\mathcal{T}}^{\tilde{h}_2}, \rho_{\hat{\mathcal{T}}} = \rho_{\mathcal{T}} \text{ where } \tilde{h}_2 = m + m\left\lceil \log_3 \frac{2\sqrt{m}\kappa_2}{\kappa_2 - 1}\right\rceil \text{ and } \kappa_2 = \frac{\sqrt{m}}{\sqrt{1 - \text{dist}^2(\mathbf{A}, \hat{\mathbf{A}})}}.$$

For brevity, denote $\nu'_{\mathcal{T}} = \max\{\nu_{\mathcal{T}}, g^* - \inf_{\mathbf{z} \in \kappa_2 \alpha \mathbb{H}_1^m} g(\mathbf{A}\hat{\mathbf{A}}^\top \mathbf{z})\}$. Now, consider $3\nu_{\hat{\mathcal{T}}}\rho_{\hat{\mathcal{T}}}^h$:

$$
\begin{aligned}
&= 3\nu'_{\mathcal{T}}\rho_{\mathcal{T}}^{h - \tilde{h}_2} \\
&= 3\nu_{\mathcal{T}}\rho_{\mathcal{T}}^{\log_{\rho_{\mathcal{T}}}(\nu'_{\mathcal{T}}\rho_{\mathcal{T}}^{-\tilde{h}_2}) - \log_{\rho_{\mathcal{T}}}(\nu_{\mathcal{T}})}\rho_{\mathcal{T}}^h \\
&\leq 3\nu_{\mathcal{T}}\rho_{\mathcal{T}}^{\left\lfloor \log_{\rho_{\mathcal{T}}}(\nu'_{\mathcal{T}}\rho_{\mathcal{T}}^{-\tilde{h}_2}) - \log_{\rho_{\mathcal{T}}}(\nu_{\mathcal{T}})\right\rfloor}\rho_{\mathcal{T}}^h \\
&= 3\nu_{\mathcal{T}}\rho_{\mathcal{T}}^{-\tilde{h}_4}\rho_{\mathcal{T}}^h && \text{(Denote } \tilde{h}_4 = -\left\lfloor \log_{\rho_{\mathcal{T}}}(\nu'_{\mathcal{T}}\rho_{\mathcal{T}}^{-\tilde{h}_2}) - \log_{\rho_{\mathcal{T}}}(\nu_{\mathcal{T}})\right\rfloor)
\end{aligned}
$$

Next, we examine $\mathcal{N}_{\hat{\mathcal{T}}}(3\nu_{\hat{\mathcal{T}}}\rho_{\hat{\mathcal{T}}}^h)$:

$$
\begin{aligned}
&\leq \mathcal{N}_{\hat{\mathcal{T}}}(3\nu_{\mathcal{T}}\rho_{\mathcal{T}}^{h - \tilde{h}_4}) \\
&\leq 4^m \mathcal{N}_{\mathcal{T}}(3\nu_{\mathcal{T}}\rho_{\mathcal{T}}^{h - \tilde{h}_4}) && \text{(Using Inequality 48 and } \mathcal{N}_{\hat{\mathcal{T}}}(3\nu_{\mathcal{T}}\rho_{\mathcal{T}}^h) \leq \mathcal{N}_{\hat{\mathcal{T}}}(3\nu_{\hat{\mathcal{T}}}\rho_{\hat{\mathcal{T}}}^h)) \\
&\leq 4^m C_{\mathcal{T}}\rho_{\mathcal{T}}^{-\eta_{\mathcal{T}}(h - \tilde{h}_4)} && \text{(Using Inequality 49 and } \forall h \geq \tilde{h}_4)
\end{aligned}
$$

Therefore, we have:

$$\mathcal{N}_{\hat{\mathcal{T}}}(3\nu_{\hat{\mathcal{T}}}\rho_{\hat{\mathcal{T}}}^h) \leq 4^m C_{\mathcal{T}}\rho_{\mathcal{T}}^{\eta_{\mathcal{T}}\tilde{h}_4}\rho_{\mathcal{T}}^{-\eta_{\mathcal{T}}h} \quad \forall h \geq \tilde{h}_4 \tag{50}$$

For heights $h \in [0 : \tilde{h}_4 - 1]$, we can use the value of the right-hand side at depth $h = \tilde{h}_4$, which is $4^m C_{\mathcal{T}}$. Hence, we have:

$$
\begin{aligned}
\mathcal{N}_{\hat{\mathcal{T}}}(3\nu_{\hat{\mathcal{T}}}\rho_{\hat{\mathcal{T}}}^h) &\leq 4^m C_{\mathcal{T}}\rho_{\mathcal{T}}^{\eta_{\mathcal{T}}\tilde{h}_4}\rho_{\mathcal{T}}^{-\eta_{\mathcal{T}}h} \quad \forall h \geq 0 \\
&= 4^m C_{\mathcal{T}}\rho_{\mathcal{T}}^{\eta_{\mathcal{T}}\tilde{h}_4}\rho_{\hat{\mathcal{T}}}^{-\eta_{\mathcal{T}}h} \quad \forall h \geq 0
\end{aligned}
$$

Therefore, we conclude that $\eta_{\hat{\mathcal{T}}} \leq \eta_{\mathcal{T}}$ and $C_{\hat{\mathcal{T}}} = C_{\mathcal{T}}4^m \rho_{\mathcal{T}}^{\eta_{\mathcal{T}}\tilde{h}_4}$. $\square$

### 11.19. Proof of Proposition 3.1

*Proof.* Let $h_{\max} = \left\lfloor \frac{n^2}{n\overline{\log}n + \frac{Tn}{3c}}\right\rfloor$ as given in the lemma statement. SequOOL opens $\left\lfloor \frac{h_{\max}}{h}\right\rfloor$ cells at depth $h$ for all $h \in [1, h_{\max}]$. Additionally, we utilize $T$ samples for every $c$ heights to learn $\hat{f}$. The total number of samples used to learn $\hat{f}$ up to height $h_{\max}$ is thus $\left\lfloor \frac{Th_{\max}}{c}\right\rfloor$. Each cell opening in SequOOL requires 3 samples. Therefore, the total number of openings performed by Algorithm 3 is given by $\sum_{i=1}^{h_{\max}} \left\lfloor \frac{h_{\max}}{i}\right\rfloor + \left\lfloor \frac{Th_{\max}}{3c}\right\rfloor$. According to the proposition, we need to show

that this quantity is $\leq n$. Consider, total number of openings:

$$
\begin{aligned}
&= \sum_{i=1}^{n} \left\lfloor \frac{h_{\max}}{i} \right\rfloor + \left\lfloor \frac{h_{\max} T}{3c} \right\rfloor \\
&\leq \left\lfloor \frac{n^2}{n\overline{\log}n + \frac{Tn}{3c}} \right\rfloor (\sum_{i=1}^{n} \frac{1}{i} + \frac{T}{3c}) && \text{(Definition of } h_{\max}) \\
&\leq \frac{n^2}{n\overline{\log}n + \frac{Tn}{3c}} (\overline{\log}n + \frac{T}{3c}) && \text{(Recall the definition: } \overline{\log}n \triangleq \sum_{t=1}^{n} \frac{1}{t}) \\
&= n
\end{aligned}
$$

Hence, the number of openings made in Algorithm 3 does not exceed $n$.

### 11.20. Proof of Theorem 4.8

We start with the Theorem 5 of (Bartlett et al., 2019). We restate the theorem, adapting it to our notation and incorporating the dependency of the parameters on the partitioning scheme $\mathcal{P}$:

**Theorem 11.2** ((Bartlett et al., 2019), Theorem 5). *Let $W$ be the standard Lambert $W$ function. Suppose $f$ along the partitioning scheme $\mathcal{P}$ satisfies Assumption 2.6 with associated $(\nu_{\mathcal{P}}, \rho_{\mathcal{P}})$, $C_{\mathcal{P}} > 1$, and near-optimality dimension $\eta_{\mathcal{P}} = \eta_{\mathcal{P}}(\nu_{\mathcal{P}}, C_{\mathcal{P}}, \rho_{\mathcal{P}})$ parameters. Then, after $n$ rounds, the simple regret of SequOOL is bounded as follows: For $\eta_{\mathcal{P}} > 0$, we use Corollary 6 of (Bartlett et al., 2019). Let $\tilde{n} = \lfloor n/\overline{\log}n \rfloor \eta_{\mathcal{P}} \log(1/\rho_{\mathcal{P}})/(C_{\mathcal{P}})$.*

- *If $\eta_{\mathcal{P}} = 0, r_n \leq \nu_{\mathcal{P}} \rho_{\mathcal{P}}^{\frac{1}{C_{\mathcal{P}}} \lfloor \frac{n}{\log n} \rfloor}$*
- *If $\eta_{\mathcal{P}} > 0, r_n \leq \nu_{\mathcal{P}} \left( \frac{\tilde{n}}{\log \tilde{n}} \right)^{-\frac{1}{\eta_{\mathcal{P}}}}$*

To invoke this Theorem for our proof, first we apply the theorem for the partitioning scheme $\hat{\mathcal{A}}$. 4.7 shows that $\hat{\mathcal{A}}$ is a valid partioning scheme, i.e., it satisfies 2.6, hence we can invoke Theorem 11.2.

Thus, for our partitioning scheme $\hat{\mathcal{A}}$, denoting $\tilde{n} = \lfloor n/\overline{\log}n \rfloor \eta_{\hat{\mathcal{A}}} \log(1/\rho_{\hat{\mathcal{A}}})/(C_{\hat{\mathcal{A}}})$, the regret is bounded by

- If $\eta_{\hat{\mathcal{A}}} = 0, r_n \leq \nu_{\hat{\mathcal{A}}} \rho_{\hat{\mathcal{A}}}^{\frac{1}{C_{\hat{\mathcal{A}}}} \lfloor \frac{n}{\log n} \rfloor}$
- If $\eta_{\hat{\mathcal{A}}} > 0, r_n \leq \nu_{\hat{\mathcal{A}}} \left( \frac{\tilde{n}}{\log \tilde{n}} \right)^{-\frac{1}{\eta_{\hat{\mathcal{A}}}}}$

Corollary 9.7 relates SequOOL parameters and gives,

$$
\rho_{\hat{\mathcal{A}}} = \rho^{\beta}, \nu_{\hat{\mathcal{A}}} = \max\{\max\{\nu, l_f\}\rho^{(1-\beta)(m-1)-\tilde{h}_1}, l_g\}\rho_{\mathcal{A}}^{-\tilde{h}_2},
$$

$$
\eta_{\hat{\mathcal{A}}}(\nu_{\hat{\mathcal{A}}}, \rho_{\hat{\mathcal{A}}}, C_{\hat{\mathcal{A}}}) \leq \frac{\eta_{\mathcal{P}}(\nu, \rho, C)}{\beta}, C_{\hat{\mathcal{A}}} = 3^d d^{d-m} (12\sqrt{m})^m C \rho^{-\eta_{\mathcal{P}} \tilde{h}_3} 4^m \rho^{\eta_{\mathcal{P}} \tilde{h}_4}.
$$

Now, we substitute these relations in our regret bound to get the upper bound in terms of default partitioning scheme $\mathcal{P}$ parameters.

$$
r_n \leq
\begin{cases}
\max\{\max\{\nu, l_f\}\rho^{(1-\beta)(m-1)-\tilde{h}_1}, l_g\}\rho^{-\beta\tilde{h}_2}\rho^{\frac{\beta}{C_1} \lfloor \frac{n}{\log n} \rfloor} & \text{if } \eta_{\mathcal{P}} = 0, \\
\max\{\max\{\nu, l_f\}\rho^{(1-\beta)(m-1)-\tilde{h}_1}, l_g\}\rho^{-\beta\tilde{h}_2} \left( \frac{\tilde{n}}{\log \tilde{n}} \right)^{-\frac{\beta}{\eta_{\mathcal{P}}}} & \text{if } \eta_{\mathcal{P}} > 0,
\end{cases}
$$

Where $C_1 = 3^d d^{d-m} (12\sqrt{m})^m C \rho^{-\eta_{\mathcal{P}} \tilde{h}_3} 4^m \rho^{\eta_{\mathcal{P}} \tilde{h}_4}$ and $\tilde{n} = \lfloor n/\overline{\log}n \rfloor \eta_{\mathcal{P}} \log(1/\rho)/C_1$

with $\tilde{h}_2 = m + m \left\lceil \log_3 \frac{2\sqrt{m}\kappa_2}{\kappa_2 - 1} \right\rceil$ and $\tilde{h}_1 = d \left\lceil \log_3 3^{\lceil \log_3 \sqrt{m}\alpha \rceil} \right\rceil$ with $\kappa_2 = \frac{\sqrt{m}}{\sqrt{1-\text{dist}^2(\mathbf{A}, \hat{\mathbf{A}})}}$

$$
\tilde{h}_3 = - \left\lfloor \log_\rho(\max\{\nu, l_f\}\rho^{(1-\beta)(m-1)-\tilde{h}_1}) - \log_\rho(\nu) \right\rfloor, \tilde{h}_4 = - \left\lfloor \log_{\rho_{\mathcal{A}}}(\max\{\nu_{\mathcal{A}}, l_g\}\rho_{\mathcal{A}}^{-\tilde{h}_2}) - \log_{\rho_{\mathcal{A}}}(\nu_{\mathcal{A}}) \right\rfloor
$$

$\square$

# 12. Additional Experiment Details & Results

## 12.1. Test Functions Experiments

We implemented SequOOL, SOO, and RESOO ourselves due to the absence of publicly available open-source code for these algorithms. For DiRect and Dual Annealing, we utilized the implementations provided in the SciPy library's optimize module. The CMA-ES algorithm was sourced from its dedicated project repository[1]. REMBO and HesBO implementations were derived from the original HesBO repository[2]. Bayesian Optimization was implemented using the repository [3].

## 12.2. Multi-Index Functions Results

We present additional experimental results to further demonstrate the effectiveness of our approach. Figure 7 showcases the performance of various algorithms on low-dimensional multi-index functions with $d = 5$ and $m = 2$. Our algorithm consistently achieves lower regret across different test functions, including Sphere, Branin, Ellipsoid, and Rastrigin, often reaching zero regret with fewer samples compared to competing methods.

## 12.3. Training Details of LLM Quantization Experiment

We implemented our Large Language Model (LLM) code on hardware equipped with one Quadro RTX 5000 GPU having 16GB VRAM. For comparison, we ran AWQ baselines using the original authors' code, which also served as a foundation for developing our proposed method.

To optimize the neural network used in Algorithm 3 for our LLM Quantization objective function, we employed the Ray package for hyper-parameter tuning [4]. We used Adam optimizer and our search space included hidden layer sizes (500, 1000, 2000, 3000), learning rates (log-uniform from $1 \times 10^{-4}$ to $1 \times 10^{-1}$), weight decay (log-uniform from $1 \times 10^{-2}$ to $1 \times 10^{-1}$), and learning rate Step Decay with gamma values (uniform from 0.9 to 0.99), and step sizes (500, 1000, 2000). We utilized early stopping to prevent overfitting.

The neural network was retrained on SequOOL-collected samples every 5 heights, with the look-ahead strategy applied up to a height of 60 and performed round-robin direction selection after this height.

---

[1] https://github.com/CyberAgentAILab/cmaes
[2] https://github.com/aminnayebi/HesBO
[3] https://github.com/bayesian-optimization/BayesianOptimization
[4] https://github.com/ray-project/ray

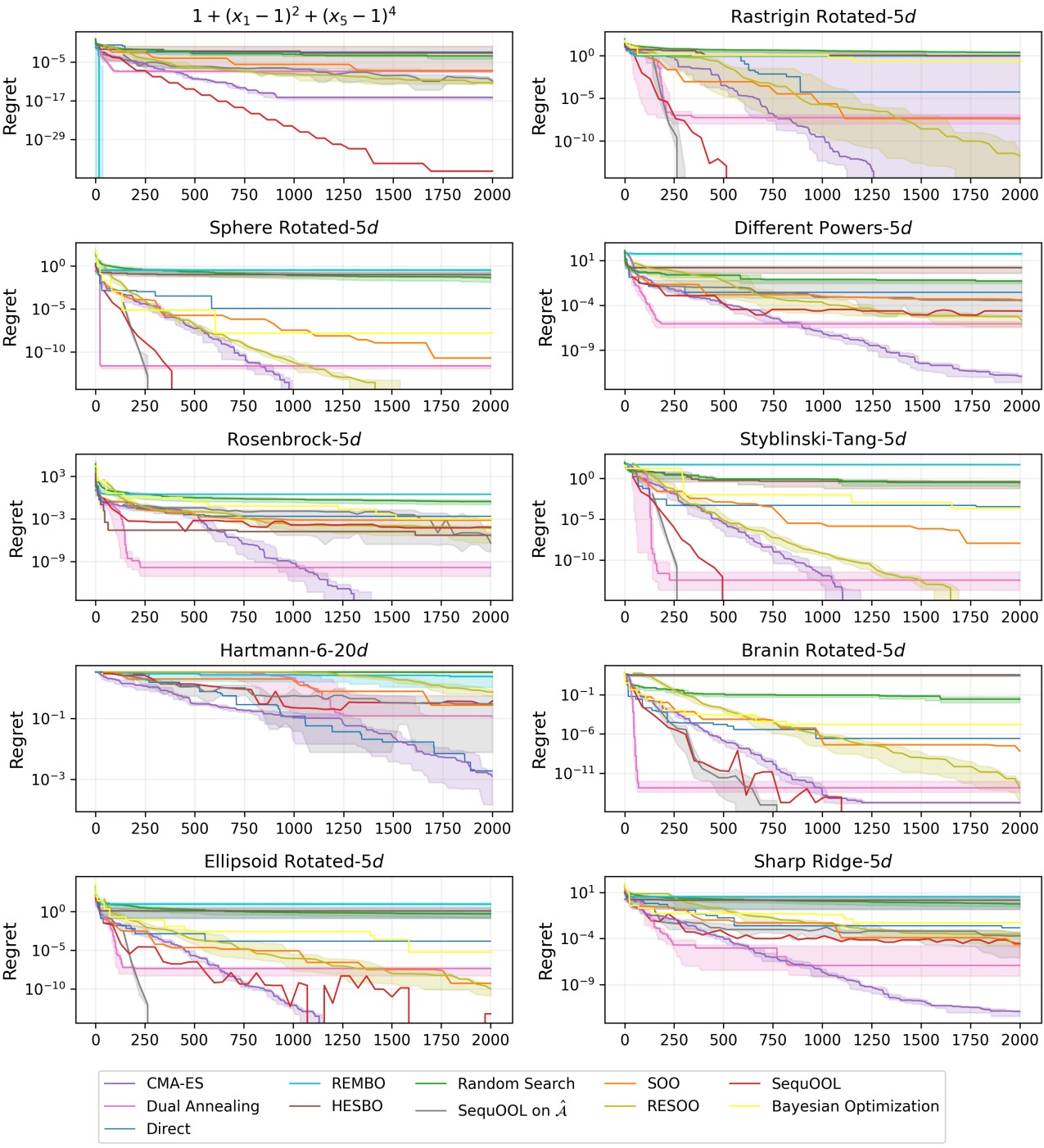

*Figure 7.* Regret Plots: Legend for the plots are arranged in the order of their performance. Algorithm 1 (SequOOL on $\hat{\mathcal{A}}$) uses 100 additional samples to learn the subspace through the Fornasier et al. (2012) approach. Our Algorithm, SOO, RESOO, SequOOL are budget algorithms, so we run these algorithms using 100 equally spaced budget values between 1 and 2000 and plot the regret at the end of each run. For the randomized algorithms, we took 10 trials and plotted the median curve (thick line) and 0 and 95 percentile curves. Random Search is run on $\hat{\mathcal{A}}$.