# OpenReview forum: "Adaptive Partitioning Schemes for Optimistic Optimization"
_ICML.cc/2025/Conference — ICML 2025 poster_

### Official Review · Reviewer_k4R6 · 2025-03-12

**Overall Recommendation:** 2

**Summary:**

This paper proposes an adaptive partitioning schemes which divides the search domain into a hierarchical subspace tree to reduce search space and enhance optimization performance. The subspace matrix is learned and updated as a hidden layer of neural network surrogate model. Experimental results on several synthetic problems and LLM quantization problems show that the proposed methods surpass the compared baselines.

## update after rebuttal

I keep my score. My final assessment of this paper is 2.

**Claims And Evidence:**

The authors demonstrate the effectiveness of the proposed methods through theoretical analysis and comparison experiment on synthetic and LLM quantization problems.

**Essential References Not Discussed:**

I didn’t see any essential references not discussed.

**Experimental Designs Or Analyses:**

1) For each problem instance, the surrogate network should be trained from scratch to obtain the partitioning matrix, however, I did not find the surrogate training settings for synthetc problems? The time efficiency for the surrogate training in synthetic problems is not reported.

2) The number of layers M for the LLMs in the experiments is not reported.

3) The test synthetic problems included in the experiments are relatively simple, which are mostly unimodal or low conditioning.

**Methods And Evaluation Criteria:**

The overall method is to learn a subspace partitioning matrix by fitting a neural network surrogate model and sample solutions from the partitioned subspace which contains the optimal solution. The authors proof the correctness of the proposed method through theoretical analysis.

**Other Comments Or Suggestions:**

None.

**Other Strengths And Weaknesses:**

Weakness: the test synthetic problems are relatively simple. The tested problems are mostly unimodal or low conditioning. For simple or unimodal problems the surrogate model could be fitted more quickly and the subspace containing optimum could also be recognized more easily. Testing the methods on multi-modal and high conditioning problems such as COCO BBOB or CEC series benchmarks might enhance the convincingness of the experiments.

**Questions For Authors:**

1) In Appendix 8, Proposition 8.1, what is the reason of using the top m right singular vectors of the p vectors in the learned weight matrix of the hidden layer as the partitioning scheme instead of setting p=m?

2) The rows in the estimated A may not be orthonormal, will it affect the optimization performance?

3) How will the values of the number of dimensions m and integer c in Algorithm 2 affect the performance? Or how to choose a proper value for m and integer c when optimizing an unseen problem?

**Relation To Broader Scientific Literature:**

1) Black-box Optimization: The proposed methods are developed to solving black-box optimization problems.

2) Surrogate Model: The partitioning matrix is learned by fitting a surrogate model.

3) Large Language Model: In the experiment, the proposed method is applied to solve the LLM quantization problems.

**Theoretical Claims:**

The authors provide the detailed proof of the proposed theorems and lemmas in the paper and Appendix.

---

> ### Author Rebuttal · Authors · 2025-04-01
>
> We respond to the reviewer's questions below.
>
> > For each problem instance, the surrogate network should be trained from scratch to obtain the partitioning matrix
>
> We trained the neural network using the Ray package for hyperparameter tuning. The search space included hidden layer sizes (500, 1000, 2000, 3000), learning rates (log-uniform from $1 \times 10^{-4}$ to $1 \times 10^{-1}$), weight decay (log-uniform from $1 \times 10^{-2}$ to $1 \times 10^{-1}$), and learning rate step decay with gamma values (uniform from 0.9 to 0.99) and step sizes (500, 1000, 2000). Early stopping was used to prevent overfitting. We will add these details in our revised draft.
>
> > The number of layers M for the LLMs in the experiments is not reported.
>
> The number of layers $M$ in our experiment is 24. We will clarify this in the revised draft.
>
> > The test synthetic problems included in the experiments are relatively simple, which are mostly unimodal or low conditioning.
>
> > Weakness: the test synthetic problems are relatively simple. The tested problems are mostly unimodal or low conditioning.
>
> We thank the reviewers for the question! Our rationale in choosing these test functions is from the paper [1]. We choose one test from each of the category to cover different type of optimization challenges. Sphere function belongs to "Separable function class", Ellipsoid function belongs to "Functions with low or moderate conditioning class". Branin function belongs to "highly non-linear class". Rastrigin belongs to "Multi-modal functions with adequate global structure". Sum of Different Powers function belongs to "Functions with high conditioning and unimodal" class. Also, RESOO [2] was evaluated using the Branin, Rosenbrock function, while HesBO [3] was tested on the Branin, Hartmann-6, Rosenbrock, and Styblinski-Tang functions. Therefore, we included the results of all the baselines on these functions.
>
> [1] Nikolaus Hansen, Steffen Finck, Raymond Ros, Anne Auger. Real-Parameter Black-Box Optimization Benchmarking 2009: Noiseless Functions Definitions. [Research Report](https://inria.hal.science/inria-00362633v2) RR-6829, INRIA. 2009.
>
> [2] Qian, Hong, and Yang Yu. "Scaling simultaneous optimistic optimization for high-dimensional non-convex functions with low effective dimensions." Proceedings of the AAAI Conference on Artificial Intelligence. Vol. 30. No. 1. 2016.
>
> [3] Nayebi, A., Munteanu, A. &; Poloczek, M.. (2019). ["A Framework for Bayesian Optimization in Embedded Subspaces."](https://proceedings.mlr.press/v97/nayebi19a.html) Proceedings of the 36th International Conference on Machine Learning, in Proceedings of Machine Learning Research 97:4752-4761.
>
> > In Appendix 8, Proposition 8.1, what is the reason of using the top m right singular vectors of the p vectors in the learned weight matrix
>
> Proposition 8.1 is valid for any $p$ and we could set the $p$ to be $m$.
>
> > The rows in the estimated A may not be orthonormal, will it affect the optimization performance?
>
> A is estimated from the SVD of neural network weights, hence the rows of estimated A are orthonormal.
>
> > How will the values of the number of dimensions m and integer c in Algorithm 2 affect the performance? Or how to choose a proper value for m and integer c
>
> In practice, we can choose the value of m to explain a desired percentage (such as 95\%) of the total variation in the SVD step calculating $\hat{\mathbf{A}}$. $c$ is a hyper parameter which is chosen as 5 in our experiments. Larger er $c$ would mean we use more samples for the optimization and lesser samples for the surrogate training. Smaller $c$ would mean we use more samples for surrogate training and less samples for optimization. For a unseen problem we need to perform hyperparameter tuning to find the optimal value for $c$.

---

### Official Review · Reviewer_4oos · 2025-03-15

**Overall Recommendation:** 3

**Summary:**

The paper proposes an adaptive partitioning scheme for optimistic optimization that extends existing gradient-free algorithms such as SequOOL. The authors consider both a two-stage and an interleaved algorithm. In the context of multi-index functions (defined on a n-dimensional subspace within m dimensions), they prove an improved simple regret bound for their method compared to SequOOL and another baseline. The algorithms are evaluated empirically on a set of synthetic functions as well as a LLM quantization task.

### Update after rebuttal
My overall assessment of the paper remains unchanged.

**Claims And Evidence:**

* The theoretical results are supported by a large set of technical results and proofs (which I have no particular reason to doubt but I also did not double check).
* The empirical results are less convincing: the performance of the proposed methods is mixed, and it is not particularly deeply explored by the authors as to why.

**Essential References Not Discussed:**

I do not have enough familiarity with the specific literature to assess this.

**Experimental Designs Or Analyses:**

* It is not clear what the settings were that were chosen for the baseline methods. Without this (e.g. dimensionality of the embedding dimension for REMBO and HESOB) it's very hard to interpret the results of the experiments.
* The evaluation of the random embedding Bayesian Optimization (REMBO, HESBO) on the low-dimensional problems in the appendix appears rather odd. It doesn't make a lot of sense to generate some lower-dimensional embedding in a 5-dimensional space if the evaluation budget is 2000. The proper comparison here would be to just use standard Bayesian Optimization.
* A more clear ablation of the performance as the ambient dimension increases would have been useful to understand the behavior.
* Why is Algorithm 2 not evaluated on the test functions?
* The results in Figure 2 are quite hard to read. I recommend focusing on showing traces from a smaller number of functions and relegate the rest into the appendix. You can also aggregate the results across functions in a more compact format to keep the key message in the paper.
* Some of the discussions appear to focus on extremely small differences (e.g. on the order of 10^-10 - 10^-12 in Section 10). How relevant are these differences? Can we actually conclude anything from them them given the variance in the results?

**Methods And Evaluation Criteria:**

* The benchmark setup appears generally reasonable, modulo the comments below.
* I would have also liked to see SAASBO (https://proceedings.mlr.press/v161/eriksson21a.html) and TuRBO (https://proceedings.neurips.cc/paper/2019/hash/6c990b7aca7bc7058f5e98ea909e924b-Abstract.html) as additional baselines, but I don't think this is critical.

**Other Comments Or Suggestions:**

I find the significance and quality of this paper quite hard to judge as it's outside my expertise, so this is a low-confidence vote.

**Other Strengths And Weaknesses:**

* The authors never describe the SequOOL algorithm in detail. Given that this is not just a comparison baseline but an integral part of the proposed method, this makes it hard to understand the contribution for the non-expert already familiar with SequOOL.
* Apart from that, despite being quite technical, the paper tries to make itself at least somewhat accessible to the non-expert. For instance, I liked the intuition provided in the paragraph right after Definition 2.7 (though I believe there is a typo: eta -> nu).
* Overall, the empirical results are rather disappointing:
    * Algorithm 1 does well on a couple of the examples (however, the authors don't provide much intuition for why htat is the case) but is otherwise worse than many baselines, especially in the small-sample regime (anytime performance matters!).
    * Algorithm 2 does not appear to be evaluated on the synthetic functions (why?).
    * For the LLM example, while a nice and practical real-world example, it is not clear how meaningful the improvement is compared to the baselines, and the results are from a single replicate - I understand that compute capacity here is a limitation, but it's still just not clear how meaningful the result is.

**Questions For Authors:**

* For the experiments, it appears that you generate a single A once per function and then run multiple optimization runs, rather than each replicate being over a new embedded subspace (i.e. a new A), is that a correct understanding? This seems like it could cause some bias in the evaluation. It may be good to re-randomize A, that would make the comparison against the non-randomized algorithms more interesting as well.
* You mention that one possible direction is to "extend your approach to the case when the function evaluations are noisy" - I'm curious how that would work; one of the core assumptions of the index function setting is that the function has a narrow ridge structure. If evaluations are noisy, then it appears that the direction of this structure would be quite hard to estimate (in other words, the learned partitioning schemes would not align with it). How would the theoretical results you obtained translate to a noisy setting.

**Relation To Broader Scientific Literature:**

The primary contribution of the paper is to propose an adaptive partitioning scheme and proving improved regret bounds for this compared to non-adaptive approaches. This is significant in the sense that - if I parse the paper correctly - this is the first work that considers an adaptive partitioning approach of this kind in the context of optimistic optimization.

**Theoretical Claims:**

* No. This is a highly technical paper in an area that I'm not very familiar with so this was not feasible.

---

> ### Author Rebuttal · Authors · 2025-04-01
>
> We respond to the reviewer's questions below.
>
> > I would have also liked to see SAASBO and TuRBO.
>
> Thank you for sharing these two relevant baselines. Previously, we tried to ran the TuRBO method and after 100 evaluations the computation time is very large, hence we did not include it in our results. We will try to run SAASBO for 2000 evaluations and add it in our revised draft.
>
> > It is not clear what the settings were that were chosen for the baseline methods (e.g. dimensionality of the embedding dimension for REMBO and HESBO).
>
> Thank you for pointing this out. dimensionality of the embedding dimension is choosen to be equal to the value $m$. We will add these details in our revised draft.
>
> > The evaluation of the random embedding Bayesian Optimization (REMBO, HESBO) on the low-dimensional problems in the appendix appears rather odd.
>
> We thank the reviewer for the suggestion!. We will include standard Bayesian Optimization in the updated version.
>
>
> > Why is Algorithm 2 not evaluated on the test functions?
>
> The experiments in Figure 2 were on synthetic multi-index functions with a low-dimensional structure. We ran variant 1 to verify and demonstrate the benefit of learning the low-dimensional subspace. However, as per your comment, we could also run Algorithm 2 on these multi-index functions. We will run it and add those results to our revised draft.
>
> > The results in Figure 2 are quite hard to read.
>
> Thank you for the suggestion. We will incorporate this in our revised draft.
>
> > Some of the discussions appear to focus on extremely small differences (e.g. on the order of 10^-10 - 10^-12 in Section 10).
>
> Thank you for your feedback. Section 10 presents an illustrative experiment to motivate lookahead direction selection. The function $f(x_1, x_2) = 1 - |x_1| - x_2^2$ is evaluated with different parameterized choices of $A$, without randomness. Thus, variance is not a concern, and the goal is to highlight potential benefits of our proposed strategy.
>
>
> > Algorithm 1 does well on a couple of the examples (however, the authors don't provide much intuition for why htat is the case)
>
> Overall, Algorithm 1 is expected to perform well when a low-dimensional ridge structure is present in the objective function. This is because our approach learns the low-dimensional structure and performs optimization on the reduced search space. The effect of the reduced search space can be seen in our regret bounds and is demonstrated in our experiments. Our Algorithm 1 is a two-stage algorithm where the first stage learns an adaptive partitioning scheme and the second stage uses it for optimization. In Figure 2, we used 650 samples for the first learning stage; consequently the anytime regret till 650 samples is high. While standard approaches like the doubling trick can be used to convert a budgeted algorithm to an anytime algorithm, they are not the most effective, and we will develop an effective anytime algorithm in future work.
>
>
> > For the LLM example, while a nice and practical real-world example, it is not clear how meaningful the improvement is compared to the baselines
>
> Thank you for your feedback. Our response is as follows:
>
> 1. Our approach to LLM quantization enables a more faithful implementation of AWQ, leading to improved quantized model accuracy.
> 2. AWQ is a widely used quantization method, as evidenced by several publicly available models:
>
>    - [Huginn-13B-v4-AWQ](https://huggingface.co/TheBloke/Huginn-13B-v4-AWQ?utm_source=chatgpt.com)
>    - [Capybara-Tess-Yi-34B-200K-DARE-Ties-AWQ](https://huggingface.co/TheBloke/Capybara-Tess-Yi-34B-200K-DARE-Ties-AWQ?utm_source=chatgpt.com)
>    - [ChatQA-1.5-8B-AWQ](https://huggingface.co/bartowski/ChatQA-1.5-8B-AWQ?utm_source=chatgpt.com)
>
> 3. Quantization is a one-time process, and prioritizing accuracy at the cost of compute is justified to ensure a high-quality model.
>
>
> > For the experiments, it appears that you generate a single A once per function and then run multiple optimization runs
>
> We do reinitialize the subspace by generating a new $A$ after each trial to ensure unbiased evaluation. We will clarify this in the revised draft.

---

> > ### Comment · Reviewer_4oos · 2025-04-03
> >
> > Thanks for the additional comments on my review.
> >
> > > For the LLM example, [...] Thank you for your feedback. Our response is as follows:
> >
> > I'm not questioning the importance and relevance of quantization and that AWQ is widely used - my question is about whether the improvements demonstrated in Table 2 are meaningful / reproducible. Calibration does not seem to improve relative to SequOOL,, and what's the incremental value of decreasing PPL from 16.83 to 16.68? Also, what variance would we expect from re-running this? Are the results statistically significant in any way?

---

> > > ### Author Response · Authors · 2025-04-08
> > >
> > > We thank the reviewer for raising this question. To better illustrate the contribution of our results, we provide the following analysis:
> > >
> > >  - **Improvement Relative to the Unquantized Model:** The unquantized model has a perplexity of 14.47, and since quantization increases the perplexity, we can think of that value as the minimum possible perplexity of any quantized model. The baseline (AWQ) quantized model achieves a perplexity of 16.92, giving a perplexity difference of 16.92 - 14.47 = 2.45. In contrast, Algorithm 2 achieves a perplexity of 16.68, giving a perplexity difference of 16.68 - 14.47 = 2.21. The relative improvement in perplexity loss is thus (2.45 - 2.21) / 2.45 $\approx$ 10\%.
> > >
> > > - **Objective Function Comparison with SequOOL:** When we apply **SequOOL** to our proposed joint optimization objective (in $3M$ dimensions), it achieves a perplexity of **16.83**. Since SequOOL is deterministic, this demonstrates that our new formulation of the quantization problem is itself beneficial and the improvement is not due to random chance.
> > >
> > >
> > > - **Empirical Evidence of Robustness:** To assess statistical robustness under compute constraints, we performed partial quantization experiments. Specifically, we quantized:
> > >
> > >   1. First 3 and 4 layers
> > >   2. Last 3 and 4 layers
> > >
> > >     In each case, we observed consistent reductions in perplexity, indicating that the improvements from Algorithm 2 are not due to chance, but are reliably obtained across different regions of the model. However, we plan to run more independent trials (5 -10) and report the results in our revised draft.

---

### Official Review · Reviewer_QRGR · 2025-03-16

**Overall Recommendation:** 2

**Summary:**

The authors propose two different versions of learning partitioning ideas for Optimistic Optimization algorithms for black-box optimization. The first, uses a two step approach that first learns the partitioning and then optimizes. The second updates the partitioning while optimizing. The authors support their claims with theoretical and empirical evidence.

**Claims And Evidence:**

Claim 1: We demonstrate the benefit of using a learned partitioning scheme for existing derivative-free optimization algorithms such as SequOOL.

On standard toy optimization problems, the learned partitioning shows improvement over SequOOL. This advantage doesn't show in the real-world problem of LLM quantization. Here, variant 1 performs worse. Variant 2 shows small (not clear whether this is significant as the others don't provide std) improvements, but also has 20% more compute budget. I consider this a mixed data point with clear need for improvement in the latter experiment.


Claim 2: When the function is a low-dimensional multi-index function we theoretically prove improved regret bounds shown in Table 1.

I'm no expert, but looks ok to me.

Claim 3: Empirically, we demonstrate the improvement in optimization error for several benchmark functions including Rastrigin (multi-modal), Branin (multiple minima), and Sharp Ridge (non-differentiable).

This is a strictly stronger claim than Claim 1 and thus those remarks apply here as well. Additionally, we see that RESOO shows similar performance on average compared to the proposed method. Thus, this claim has no strong support.

Claim 4: We pose the quantization of Large Language Model (LLM) as a high-dimensional black-box optimization problem and obtain an improved perplexity value.

The authors show an improvement, but also use 1/3 more search budget. Thus, this is no fair comparison. Improvements seem rather small as well.

**Essential References Not Discussed:**

This might be also interesting work for the authors. Instead of using Bayesian optimization on a random lower dimensional space, there is work learning this space:

Wenlong Lyu, Shoubo Hu, Jie Chuai, Zhitang Chen: Efficient Bayesian Optimization with Deep Kernel Learning and Transformer Pre-trained on Multiple Heterogeneous Datasets

Martin Wistuba, Josif Grabocka: Few-Shot Bayesian Optimization with Deep Kernel Surrogates

**Experimental Designs Or Analyses:**

Yes. Covered in "Methods And Evaluation Criteria"

**Methods And Evaluation Criteria:**

The authors choose high-dimensional problems which makes sense for their problem. Most benchmarks are "toy" functions. I'd like them to tackle real-world problems instead. NAS benchmarks could be one example of high-dimensional benchmarks with stronger connection to practical relevance.

Allowing different compute time for the result in Table 2 makes little sense since we are no longer able to compare the performance of different search methods.

**Other Comments Or Suggestions:**

I found it hard to identify which are the algorithms presented by the authors at times. Maybe give them a name instead of referring to Alg 1/2?

**Other Strengths And Weaknesses:**

N/A

**Questions For Authors:**

Where are Algorithm 2 results for Figure 2?

**Relation To Broader Scientific Literature:**

In my opinion the authors describe it very well. They could have covered some other learning-related work as well. I leave pointers below.

**Theoretical Claims:**

I tried follow the authors reasoning, but I'm a more applied researcher. I leave the evaluation of this part to my fellow reviewers.

---

> ### Author Rebuttal · Authors · 2025-04-01
>
> We respond to the reviewer's questions below.
>
> > Claim 1: We demonstrate the benefit of using a learned partitioning scheme & Claim 4: We pose the quantization of Large Language Model (LLM) as a high-dimensional black-box optimization problem
>
> We thank the reviewers for recognizing our claims. Our approach enables a more faithful application of the AWQ quantization strategy: We quantize the entire LLM end-to-end while the AWQ algorithm had to perform an independent grid search over each successive layer. Since quantization is a "perform once, reuse many times" operation, and small improvements in the perplexity score are often critical, we believe our approach could be fruitfully applied on other larger models. We believe this to be one of our main contributions. Regarding the search budget values in the experiments, grid search had long stopped improving and hence we had paused its execution at 9 hours. We will match the compute time budget in the updated version.
>
> Variant 1 of our algorithm exploits a low-dimensional ridge structure in the objective function. Since the LLM quantization problem did not have such a structure, variant 1 did not provide an advantage. However, as the experimental results show, Variant 2 can provide an advantage even when such a structure is not present.
>
> > Claim 3: Empirically, we demonstrate the improvement in optimization error
>
> The contribution statement is a general claim. Specifically, compared to RESOO, our method achieved zero regret on both the Rastrigin and Styblinski-Tang functions, whereas RESOO had a regret of $5.5 \times 10^{-3}$ on Rastrigin. For Styblinski-Tang, our method required approximately 900 evaluations to reach zero regret, while RESOO needed 2000.
>
> > Where are Algorithm 2 results for Figure 2?
>
> Thank you for pointing this out. The experiments in Figure 2 were on synthetic multi-index functions with a low-dimensional structure. We ran variant 1 to verify and demonstrate the benefit of learning the low-dimensional subspace. However, as per your comment, we could also run Algorithm 2 on these multi-index functions. We will run it and add those results to our revised draft.

---

### Decision · Program_Chairs · 2025-05-01

**Decision:**

Accept (poster)

**Comment:**

The paper introduces two adaptive partitioning algorithms for optimistic optimization, extending gradient-free algorithms such as SequOOL. The paper proposes a novel approach by bringing adaptivity to the optimistic optimization setting. It provides an interesting theoretical proof. However, reviewers pointed out a few weaknesses, such as a lack of important baselines, limited experimentation, and a lack of important reproducibility information. Based on the rebuttal and the author's commitment to update the paper, I would recommend a weak accept.